# Optimal criterion for feature learning of two-layer linear neural network in high dimensional interpolation regime

**Keita Suzuki**
University of Tokyo and Preferred Netowrks, Inc.
keitasuzuki@preferred.jp

**Taiji Suzuki**
University of Tokyo and RIKEN AIP
taiji@mist.i.u-tokyo.ac.jp

## Abstract

Deep neural networks with feature learning have shown surprising generalization performance in high dimensional settings, but it has not been fully understood how and when they enjoy the benefit of feature learning. In this paper, we theoretically analyze the statistical properties of the benefit from feature learning in a two-layer linear neural network with multiple outputs in a high-dimensional setting. For that purpose, we propose a new criterion that allows feature lerning of a two-layer linear neural network in a high-dimensional setting. Interestingly, we can show that models with smaller values of the criterion generalize even in situations where normal ridge regression fails to generalize. This is because the proposed criterion contains a proper regularization for the feature mapping and acts as an upper bound on the predictive risk. As an important characterization of the criterion, the two-layer linear neural network that minimizes this criterion can achieve the optimal Bayes risk that is determined by the distribution of the true signals across the multiple outputs. To the best of our knowledge, this is the first study to specifically identify the conditions under which a model obtained by proper feature learning can outperform normal ridge regression in a high-dimensional multiple-output linear regression problem.

## 1 Introduction

Thanks to the benefit of feature learning, neural networks have demonstrated superior generalization performance across numerous high-dimensional problems, including image classification and natural language processing. This empirical observation is surprising as it conflicts with conventional statistical theory, which claims a model shouldn't be too complex to predict unseen randomly labeled data well. Therefore it has drawn great attention from the community of learning theory (Belkin et al., 2018). To understand the generalization of high-dimensional statistical models, many researchers have studied the generalization mechanism of simpler models such as linear models, which are amenable to theoretical analysis in over-parametrized settings. Thanks to intensive research, we've gained an understanding of benign overfitting phenomenon in many simple models. In this phenomenon, models generalize well to unseen test data even when they interpolate training data with significant label noise, and we can see this in linear regression (Bartlett et al., 2020; Hastie et al., 2022; Muthukumar et al., 2020; Negrea et al., 2020; Xu and Hsu, 2019), ridge regression (Tsigler and Bartlett, 2020; Dobriban and Wager, 2018; Mei and Montanari, 2022), logistic regression (Montanari et al., 2019; Chatterji and Long, 2021), and kernel-based estimators (Liang and Rakhlin, 2020; Belkin et al., 2019; Li et al., 2022). However, in practical use, neural networks frequently surpass these simpler models in high-dimensional problem settings. Hence, understanding the benefits of feature learning and its dominance over simple models without feature learning remains crucial in order to elucidate the generalization of deep learning. For this reason, many researchers have studied feature learning of two-layer neural networks, which is the simplest case of neural networks. However, much of this research is confined to the setting of classification problems (Frei et al., 2022; Cao et al., 2022), with few studies dedicated to regression problems. This is because there is a difficulty in feature learning for over-parameterized regression problems that increasing the number of parameters excessively for feature mapping can lead to overfitting. Accordingly, it's crucial to apply regularization for parameters including feature mapping for appropriate

feature learning. As a regularization for such parameters, a considerable number of researchers are focusing on implicit regularization. This form of regularization is automatically induced by optimization methods and deep learning architectures. Several studies have examined the influence of implicit regularization on the generalization of two-layer neural networks in high-dimensional regression problems (Chatterji et al., 2022; Azulay et al., 2021). However, these analyses on implicit regularization aren't enough to fully understand the benefit of feature learning. This is because such implicit regularization does not always leverage the entire potential of feature learning, that is, it isn't necessarily an optimal regularization in terms of feature learning. In particular, the existing work on implicit regularization as listed above missed important properties of feature learning such as the Bayes risk optimality that will be shown in our study. Therefore, the goal of this work is to answer the following question:

*Can we design an optimal regularization of feature mapping to fully exploit the benefit of feature learning and demonstrate its improvement over simpler models like ridge regression?*

## 1.1 CONTRIBUTIONS

For the goal above, we consider a two-layer linear neural network which is the simplest setting of feature learning in a multi-output linear regression problem setting. We use the ridge estimator as the second-layer parameter and design a proper regularization on the first-layer parameter which enables feature learning. Our contribution can be summarized as follows:

- We propose a criterion that gives an upper bound of the predictive risk and show that the predictive risk for an estimator with smaller values of the criterion converges to zero. This criterion has a proper regularization term on the first-layer parameter.
- An important characterization of the proposed criterion is that the minimizer of the criterion achieves near-optimal Bayes risk under some conditions, which represents the adequacy of the criterion. This is a merit of considering a multi-output setting.
- We characterize when the normal ridge estimator does not generalize and show the two-layer neural network selected by the proposed criterion generalizes even in such situations using examples. Indeed, the predictive risk of such an estimator can converge to zero even when that of the naive ridge regression does not. This result demonstrates the benefit of feature learning and its dominance over simpler models in high-dimensional regression problems.

## 1.2 RELATED WORK

**Benign overfitting of linear models.** One line of research sought a theoretical understanding of the benign overfitting phenomenon in linear models. Hastie et al. (2022) gave an asymptotic evaluation of the bias and variance of linear models and random feature regression. Additionally, they demonstrated that the predictive risks associated with certain models exhibit a double descent curve as the ratio of parameter size to sample size becomes large. Bartlett et al. (2020) derived a tight upper bound for the predictive risk of the minimum-norm interpolator in an over-parametrized setting. They further elucidated the role of the input covariance matrix's spectrum in enabling the model to generalize well. Furthermore, Tsigler and Bartlett (2020) extended this result to ridge regression. Belkin et al. (2019) showed that the Nadaraya-Watson estimator with a singular kernel that perfectly fits training data enjoys the benign overfitting phenomenon, because thanks to singular kernels, this model fits the training data with spike-like shape. Xu and Hsu (2019) gave an evaluation of the predictive risk depending on the number of features in principal component regression and showed that the number of features shouldn't be reduced too much in high dimensional settings. Li et al. (2022) showed that in the framework of (Bartlett et al., 2020), noises added to features play a role of regularization and demonstrated that the double descent phenomenon occurs in this model. However, these methods are not optimal and can sometimes fail to generalize in our settings as this study demonstrates.

**Benign overfitting of two-layer neural network.** Many recent works have characterized the generalization performance of two-layer neural networks. Chatterji et al. (2022) studied the two-layer linear neural network setting we consider here and analyzed the generalization performance of a two-layer linear neural network trained by gradient flow. Frei et al. (2022) studied benign overfitting of a two-layer neural network with nonlinear activation function in classification problems. They showed that if the ground truth is linear, a two-layer neural network trained by gradient descent can achieve min-max optimal test error even if its training error is vanishingly small. Cao et al. (2022)

also studied benign overfitting in classification problem settings. They characterized the benign overfitting phenomenon in terms of signal-to-noise ratio in a convolutional neural network whose activation function is the power of ReLU. Additionally, they demonstrated that if the signal-to-noise ratio satisfies a certain condition, a two-layer CNN trained by gradient descent achieves arbitrary small training and test loss, and otherwise, the CNN can only achieve constant level test loss. Other works (Ba et al., 2022; Damian et al., 2022) showed that the two-layer neural networks with one gradient step on the first-layer parameter can outperform fixed kernel regression. Especially, Ba et al. (2022) highlighted the importance of a large learning rate in the early phase of training and demonstrated that if the learning rate is small, the two-layer neural network can't defeat the best linear model. However, these analyses handled only one-step gradient learning, as they were motivated by optimizability, and it's unknown what kind of statistical optimality the actual optimal solution satisfies in situations where $n \ll d$. Indeed, both were considering settings where $n \geq d$.

## 2 PROBLEM SETTINGS AND NOTATIONS

**Multivariate linear regression.** In this paper, we focus on the multi-output linear regression problem. Suppose that we observe $n$ training examples $D_n = \left\{ \left( x_1, \left( y_1^{(1)}, \ldots, y_1^{(m)} \right) \right), \ldots, \left( x_n, \left( y_n^{(1)}, \ldots, y_n^{(m)} \right) \right) \right\}$, where $x_i \in \mathbb{R}^d$ and $\left( y_i^{(1)}, \ldots, y_i^{(m)} \right) \in \mathbb{R}^m$, generated by

$$y_i^{(j)} = \beta_{*j}^\top x_i + \epsilon_i^{(j)}, \ \ j = 1, \ldots, m,$$

with the unknown signals $\beta_{*j}$ $(j = 1, \ldots, m)$ that we want to learn, where $(x_i)_{i=1}^n$ are i.i.d. sample from some distribution $P_\chi$ with mean 0 and covariance matrix $\Sigma_x$, and $\left( \epsilon_i^{(j)} \right)_{i=1}^n$ are noise independent of $x_i$. This setting is called multivariate linear regression, which has been studied for a long time (Breiman and Friedman, 1997; Bai et al., 1992). Indeed, we encounter a multivariate output setting in several situations. Especially, in standard deep learning applications, we typically deal with multivariate output like multi-label classification problems. Our problem setting could be considered as the simplest problem setting to consider such a problem.

**Assumption.** Throughout the paper, we put the following sub-gaussian assumption on the input random variable. We define $\|Z\|_{\psi_2}$ to be sub-gaussian norm defined by $\|Z\|_{\psi_2} := \inf \left\{ t > 0 : \mathbb{E} \left[ \exp \left( Z^2 / t^2 \right) \right] \leq 2 \right\}$ for any random variable $Z$ taking its value in $\mathbb{R}$, and $\|Z\|_{\psi_2} := \sup_{s \neq 0} \|\langle s, Z \rangle / \|s\| \|_{\psi_2}$ for any random variable vector $Z$ taking its value in $\mathbb{R}^d$. Let design matrix $X := (x_1, \ldots, x_n)^\top \in \mathbb{R}^{n \times d}$, and then we assume the following condition on the distribution of $X$.

**Assumption 1.** *For some positive constants $\sigma_x = O(1)$ and $\sigma = O(1)$, the rows of $X\Sigma_x^{-\frac{1}{2}}$ are sub-gaussian vectors with the sub-gaussian norm at most $\sigma_x$ and components of $\epsilon^{(i)}$ are independent and have sub-gaussian norms bounded by $\sigma$.*

**Two-layer linear neural network.** In this paper, we consider estimating the target linear functions $\beta_{*i}^\top x$ by a two-layer linear neural network which is given by a form of $f_i(x; W, \beta) = \beta_i^\top W x$ where $W \in \mathbb{R}^{d \times d}$ and $\beta_i \in \mathbb{R}^d$ $(i = 1, \ldots, m)$. When the first-layer's parameter $W$ is given, we estimate the second layer's parameter $\beta_i$ as the ridge regression estimator:

$$\hat{\beta}_i(W) = W X^\top \left( X W^\top W X^\top + n\lambda I_n \right)^{-1} y^{(i)}, \ \ i = 1, \ldots, m. \tag{2.1}$$

Note that the first layer's parameter $W$ works as a linear coordinate transform. Accordingly, we estimate the target function by $f_i(x; W, \hat{\beta}_i(W)) = \hat{\beta}_i(W)^\top W x$. Then, our question becomes what kind of $W$ yields better performance.

**Notations.** We write $a \lesssim b$ if there exists an absolute constant $c$ such that $a \leq cb$, and $a \lesssim_{\sigma_x} b$ if there exists an absolute constant $c_x$ that only depends on constant $\sigma_x$ such that $a \leq c_x b$. For positive semidefinite matrix $M$, we define $\| \cdot \|_M := \sqrt{x^\top M x}$. For any matrix $M \in \mathbb{R}^{n \times d}$, we define $M_k \in \mathbb{R}^{n \times k}$ to be the matrix which is composed of the first $k$ columns of $M$, $M_{-k} \in \mathbb{R}^{n \times d-k}$ to be the matrix which is composed of the rest of $d - k$ columns of $M$. Accordingly, for any matrix $M \in \mathbb{R}^{d \times d}$, we define the eigenvalues of $M$ in decreasing order by $\mu_1(M) \geq \cdots \geq \mu_d(M)$, $M_{k,k} \in \mathbb{R}^{k \times k}$ to be the submatrix of $M$ which is composed of the upper left $k \times k$ components

of $M$, and $M_{-k,-k} \in \mathbb{R}^{d-k \times d-k}$ to be the submatrix of $M$ which is composed of the lower right $d - k \times d - k$ components of $M$. For matrix $A, B$, if $B - A$ is positive semidefinite, we write $A \leqslant B$.

# 3 MAIN RESULTS

## 3.1 SELECTING FIRST-LAYER PARAMETER $W$

Given the ridge estimator for the second layer's parameter, our interest is what kind of $W$ yields generalization. A natural strategy is choosing W corresponding to vanishingly small predictive risk which is unknown a priori. For this purpose, we aim to construct an estimator of the predictive risk as a function of $W$. More specifically, we introduce a criterion $R(W)$ as an upper bound of the predictive risk defined as

$$R(W) = \frac{1}{m} \sum_{i=1}^{m} \min_{\beta_i \in \mathbb{R}^d} \frac{1}{n} \left\| y^{(i)} - XW^\top \beta_i \right\|^2 + \lambda \|\beta_i\|^2 + \frac{\sigma'^2}{n} \mathrm{Tr} \left( \hat{\Sigma}_{wx} \left( \hat{\Sigma}_{wx} + \lambda I_d \right)^{-1} \right),$$

where $\hat{\Sigma}_{wx} := \frac{1}{n} WX^\top XW^\top$ and $\sigma'^2$ is a hyper-parameter. Note that the last term plays a role of regularization on feature mapping $W$ as stated in the introduction. For $\Sigma_{wx} := W\Sigma_x W^\top$, we obtain an upper bound of the predictive risk by bias-variance decomposition as follows:

$$\frac{1}{m} \sum_{i=1}^{m} \mathbb{E}_x \left[ \left( x^\top \beta_{*i} - x^\top W^\top \hat{\beta}_i (W) \right)^2 \right] \quad \lesssim \quad B + V,$$

where,

$$B := \frac{1}{m} \sum_{i=1}^{m} \left\| \left( I_d - WX^\top \left( XW^\top WX^\top + n\lambda I_n \right)^{-1} XW^\top \right) W^{-\top} \beta_{*i} \right\|_{\Sigma_{wx}}^2,$$

$$V := \frac{1}{m} \sum_{i=1}^{m} \left\| WX^\top \left( XW^\top WX^\top + n\lambda I_n \right)^{-1} \epsilon^{(i)} \right\|_{\Sigma_{wx}}^2.$$

$B$ and $V$ correspond to bias and variance terms respectively. $R(W)$ is essentially important because it indeed yields an upper bound of the bias and variance as stated in the following theorem.

**Theorem 1.** *There exist (large) constants $c_1$, $c_2$ and $c_3$ which only depend on $\sigma_x$ such that if $t$ satisfies $t \in \left( 1, \min \left\{ n/c_1, n/4c_1^2 \right\} \right)$, and $\sigma'^2$ satisfies $t\sigma^2 \lesssim \sigma'^2$, then it holds that with probability at least $1 - 24e^{-t/c_1} - 8e^{-c_3 n} - \frac{(\log 2)(1+\sigma_x^2)\sigma^2 \max\|\beta_{*i}\|_{\Sigma_x}^2}{n\delta^2}$, for all $W$ which satisfies $R(W) - \sigma^2 \leq \frac{\sigma'^2}{\max\{8c_1^2, c_2\}}$,*

$$B + V \lesssim_{\sigma_x} \max\{R(W) - \sigma^2, \delta\}.$$

Theorem 1 asserts that if we take $\sigma'^2$ moderately large, $R(W)$ plays a role of a criterion to select first-layer representation $W$ because $R(W) - \sigma^2$ can be regarded as an estimator for the predictive risk. Therefore, a small function value of $R(W)$ directly leads to small predictive risk, and if we obtain $W$ which makes $R(W) - \sigma^2$ close to zero, the two-layer neural network with such $W$ generalizes in the sense of predictive risk. The regularization term $\sigma'^2 \mathrm{Tr} \left( \hat{\Sigma}_{wx} \left( \hat{\Sigma}_{wx} + \lambda I_d \right)^{-1} \right)$ is called "degrees of freedom", which we can see in Widely Applicable Information Criterion (WAIC) (Watanabe, 2010) or Mallows' $C_p$ (Mallows, 1973) of ridge regression defined as $\hat{L}(\lambda) := \left\| y - X(X^\top X + n\lambda I_d)^{-1} X^\top y \right\|^2 + 2\hat{\sigma}^2 \mathrm{Tr} \left( \hat{\Sigma}_x \left( \hat{\Sigma}_x + \lambda I_d \right)^{-1} \right)$. This criterion is used to select optimal $\lambda$ in $d \ll n$ settings. Our proposed function can be regarded as a kind of extension in the sense that we can select optimal basis $W$ in $d \geq n$ settings. Note that, the theoretical properties of Mallows' Cp and WAIC are valid only in $d \ll n$ settings, and so a theoretical guarantee of the degrees of freedom as a penalty term in high-dimensional settings is a novelty of our work. Typically, in a high dimensional setting where the eigenvalues of $\Sigma_x$ have slow decay, the ridge regression with positive $\lambda$ is not necessarily optimal (Wu and Xu) and this kind of classic degrees of freedom may not work because the sum of eigenvalues diverges (Bartlett et al., 2020; Tsigler and Bartlett,

2020). Indeed, diverging eigenvalue-sum is essentially important for obtaining benign overfitting. However, in a two-layer neural network setting, *the coordinate transformation by $W$ can change the problem into like a kernel regime*. The $W$ chosen by $R(W)$ yields an optimal kernel basis in terms of predictive risk, enabling the model to generalize even in situations where it could not generalize without the transformation as we will see later. This can happen if the $\beta_{*j}$ are well correlated to each other. Such a condition will be distilled to Assumption 3, under which we can show the ridge regression with the optimal $W$ minimizing $R(W)$ gives the Bayes optimal risk.

## 3.2 EVALUATION OF THE MINIMUM RISK ESTIMATE $R(W) - \sigma^2$

Based on Theorem 1, we may have an upper bound of the predictive risk by evaluating how small $R(W) - \sigma^2$ can be. Here, we give an explicit form of the upper bound by using the eigenvalues of $\Sigma_x$ and the covariance matrix of $(\beta_{*j})_{j=1}^m$. We let $\Sigma_\beta := \frac{1}{m} \sum_{i=1}^m \beta_{*i} \beta_{*i}^\top$, and define $\sigma_1^2 \geq \cdots \geq \sigma_d^2$ and $v_1, \ldots, v_d \in \mathbb{R}^d$ as eigenvalues of $\Sigma_\beta$ and corresponding eigenvectors respectively. Let $V := (v_1, \ldots, v_d)$. Then, we can obtain an upper bound of the infimum of the objective $R(W) - \sigma^2$ as stated in the theorem below.

**Theorem 2.** *Suppose there exists $k \leq n$ such that $\mu_{k+1} \left( \Sigma_\beta^{\frac{1}{2}} \Sigma_x \Sigma_\beta^{\frac{1}{2}} \right) \leq \frac{\sigma'^2}{n} \leq \mu_k \left( \Sigma_\beta^{\frac{1}{2}} \Sigma_x \Sigma_\beta^{\frac{1}{2}} \right)$. For $W = \mathrm{diag}\left( w_1, \ldots, w_d \right) V^\top$ such that $w_i^2 = \frac{n\lambda}{\sigma'^2} \sigma_i^2$, there exists a constant $c$ which only depends on $\sigma_x$ such that, with probability at least $1 - 4e^{-cn}$,*

$$R(W) - \sigma^2 \lesssim_{\sigma_x} \sum_{i=1}^d \min \left\{ \frac{\sigma'^2}{n}, \mu_i \left( \Sigma_\beta^{\frac{1}{2}} \Sigma_x \Sigma_\beta^{\frac{1}{2}} \right) \right\} := U_{\mathrm{NN}}. \tag{3.1}$$

Therefore, as long as the right-hand side of (3.1) is close to 0, $W$ in Theorem 2 can make the predictive risk itself close to 0. This means that the two-layer neural network selected by $R(W)$ can generalize. Qualitatively, this result is due to "good" regularization via feature learning using $R(W)$ on the second-layer parameter. Normal $l_2$ regularization is often not enough because this regularization can be too strong even in a direction where signals have large contribution, which leads to large bias. In an ideal regularization, directions in which the signals have a large contribution should be weakly regularized, thereby resulting in small bias. Conversely, directions where signals have a small contribution ought to be strongly regularized to facilitate small variance. For this ideal regularization, information about the distribution of signals is essential which is usually not accessible. Thanks to degrees of freedom as a regularization on $W$, $R(W)$ enables us to make use of the information on the distribution of signals, and this is one of the biggest merits of selecting $W$ by $R(W)$.

## 3.3 BAYES RISK OPTIMALITY

So far, we have derived an upper bound of the predictive risk. In this section, we show that the derived upper bound can achieve a lower bound under some conditions using the optimal Bayes risk. As stated in the introduction, this is the first attempt to show that an optimized two-layer neural network can achieve Bayes risk optimality. We suppose $\Sigma_\beta$ is positive definite. To construct a lower bound of the predictive risk of the two-layer linear neural network we consider here, we introduce the optimal Bayes risk $R_{\mathrm{opt}}(X, \sigma) := \inf_{\hat\beta:\text{estimator}} \mathbb{E}_{\beta_* \sim \mathcal{N}(0, \Sigma_\beta)} \left[ \mathbb{E}_{Y \sim N(X\beta_*, \sigma^2 I)} \left[ \| \beta_* - \hat\beta \|_{\Sigma_x}^2 \right] \right]$ where $X = (x_1, \ldots, x_n)$ is a given training input data, $Y = X\beta_* + \epsilon$ with $\epsilon_i \sim N(0, \sigma^2)$, and the infimum is taken over all estimators depending on $(X, Y)$. The Bayes risk coincides with the predictive risk of the multi-output linear regression averaged over the multiple-output since the Bayes risk is an average of risks over the choice of $\beta_*$. Therefore, it is natural to expect that attaining the Bayes optimal risk directly leads to minimizing the predictive risk of the multiple-output regression. The Bayes estimator $\hat\beta_{\mathrm{Bayes}}$ is defined as the minimizer that attains the infimum.

**Proposition 1.** *For positive definite matrices $\Sigma_x, \Sigma_\beta$, we suppose $x \sim \mathcal{N}(0, \Sigma_x)$, $\beta_* \sim \mathcal{N}(0, \Sigma_\beta)$, $\epsilon_i \sim \mathcal{N}(0, \sigma^2)$ and training data $(x_i, y_i)_{i=1}^n \in \mathbb{R}^d \times \mathbb{R}$ is generated by $y_i = \beta^\top x_i + \epsilon_i$. For $X = (x_1, \ldots, x_n)$, $\epsilon = (\epsilon_1, \ldots, \epsilon_d)^\top$ and $y(\beta_*, \epsilon) = (y_1, \ldots, y_n)^\top$, the Bayes estimator $\hat\beta_{\mathrm{Bayes}}$ can be explicitly given by*

$$\hat\beta_{\mathrm{Bayes}}(\beta_*, \epsilon) = \left( X^\top X + \sigma^2 \Sigma_\beta^{-1} \right)^{-1} X^\top y(\beta_*, \epsilon).$$

This regularization term $\Sigma_\beta^{-1}$ enables ideal regularization discussed above. Furthermore, for $W_B = \frac{\sqrt{n\lambda}}{\tilde{\sigma}} \Sigma_\beta^{1/2}$, we can write this optimal Bayes estimator as $\hat{\beta}_{\text{Bayes}} = W_B^\top \left( W_B X^\top X W_B^\top + n I_d \right)^{-1} W_B X^\top y$. Therefore, we can represent this Bayes estimator with a two-layer linear neural network and this $W_B$ coincides with $W$ in Theorem 2 except for constant factor. As we will see later, minimum norm interpolator or more generally, normal ridge regression, does not achieve Bayes risk optimality. Asymptotically, this result coincides with the result of (Wu and Xu) when $\Sigma_x$ and $\Sigma_\beta$ have the same eigenvectors. To construct a lower bound of this optimal Bayes risk, for $\tilde{\sigma}^2 = \text{Var}\left[\epsilon_i^{(j)}\right]$ we make the following assumptions:

**Assumption 2.** *We assume that the variance of label noise is not too large, i.e., it holds $2\tilde{\sigma}^2 \leq 1$.*

**Assumption 3.** *We assume that the contribution to the output of the components corresponding to small eigenvalues of $\Sigma_x$ is not too large, that is, there exists $k \leq n$, such that $\sum_{i=k+1}^{d} \mu_i \left( \Sigma_\beta^{\frac{1}{2}} \Sigma_x \Sigma_\beta^{\frac{1}{2}} \right) \leq \tilde{\sigma}^2$.*

This assumption ensures that $\Sigma_\beta^{\frac{1}{2}} \Sigma_x \Sigma_\beta^{\frac{1}{2}}$ has a rapidly decreasing eigenvalues, which indicates that alignment between $\Sigma_\beta$ and $\Sigma_x$ has anisotropic property and then we may have better performance by finding the informative direction in the first layer. Under these assumptions, we can construct a lower bound of the predictive risk in the regime of (Tsigler and Bartlett, 2020) as follows:

**Theorem 3.** *Write eigenvectors of $\Sigma_\beta^{\frac{1}{2}} \Sigma_x \Sigma_\beta^{\frac{1}{2}}$ as $u_{B,1}, \ldots, u_{B,d} \in \mathbb{R}^d$ and $U_B := (u_{B,1}, \ldots, u_{B,d}) \in \mathbb{R}^{d \times d}$. Suppose that components of the rows of $X W_B^\top U_B$ are independent. Then under Assumption 2, 3, for $k$ in Assumption 3 there exists some absolute constant $c$ such that for any $t$ which satisfies $c < t < \min\left\{n, \frac{n}{2\sigma_x^2}\right\}$ and $k + 2\sigma^2 + \sqrt{kt}\sigma^2 < n/2$, it holds that with probability at least $1 - 24e^{-t/c}$,*

$$\min_W \frac{1}{m} \sum_{i=1}^{m} \mathbb{E}_{x,\epsilon^{(i)}}\left[ \left( x^\top \beta_{*i} - x^\top W^\top \hat{\beta}_i(W) \right)^2 \right] \geq R_{\text{opt}}(X, \tilde{\sigma}) \gtrsim \sum_{i=1}^{d} \min\left\{ \frac{\tilde{\sigma}^2}{n}, \mu_i\left( \Sigma_\beta^{\frac{1}{2}} \Sigma_x \Sigma_\beta^{\frac{1}{2}} \right) \right\}.$$

Recalling that the upper bound is $U_{\text{NN}} = \sum_{i=1}^{d} \min\left\{ \frac{\sigma'^2}{n}, \mu_i\left( \Sigma_\beta^{\frac{1}{2}} \Sigma_x \Sigma_\beta^{\frac{1}{2}} \right) \right\}$, Theorem 3 claims that the upper bound of the bias and variance of the optimized two-layer neural network (3.1) achieves the lower bound of the optimal Bayes risk as long as $\sigma'^2 = \Omega\left(\tilde{\sigma}^2\right)$. This result characterizes the effectiveness of feature learning using the proposed criterion $R(W)$. In other words, minimizing $R(W)$ can properly find the informative direction, which can be realized thanks to the multiple-output setting. This result indicates that the degrees of freedom is the optimal regularization of feature mapping $W$ in terms of feature learning in the multivariate linear regression problem. Even though $R(W)$ isn't employed in real-world applications, we can establish that in practical terms, neural networks utilize the full potential of feature learning by identifying the implicit regularization that lowers the value of this criterion.

## 4 COMPARISON WITH NORMAL RIDGE REGRESSION

In this section, we discuss the generalization performance of the two-layer linear neural network whose first-layer representation $W$ is chosen based on $R(W)$, and compare this to a normal ridge estimator without feature learning. Here, we compare the predictive risk of the selected two-layer linear neural network with that of the normal ridge regression (i.e., $W = I$). For that purpose, we derive an upper bound of the predictive risk of the normal ridge regression. This is the first attempt to clarify when a two-layer linear neural network surpasses normal ridge regression in the setting of high-dimensional, multivariate linear regression problems. To construct an upper bound, first, we define eigenvalues of $\Sigma_x$ and their corresponding eigenvectors as $\lambda_1 \geq \cdots \geq \lambda_d$ and $u_1, \ldots, u_d \in \mathbb{R}^d$. Accordingly, we define $U := (u_1, \ldots, u_d) \in \mathbb{R}^{d \times d}$ and $\tilde{\sigma}_i^2 := u_i^\top \Sigma_\beta u_i$. In the same way as (Tsigler and Bartlett, 2020), for any $k < n$, we define $\rho_k := \frac{1}{n\lambda_{k+1}} \left( n\lambda + \sum_{i>k} \lambda_i \right)$ and $A_k(XU, \lambda) := (XU)_{-k} (XU)_{-k}^\top + n\lambda I_n$. Then, we can evaluate the predictive risk of normal ridge regression as stated below.

**Corollary 1.** *Suppose it is known that for some $\delta < 1 - 4^{-n/c_x^2}$, with probability at least $1 - \delta$, the condition number of $A_k(XU, \lambda)$ is at most $L$. Then there exists a constant $c_x$, which only depends*

*on $\sigma_x$, such that for any $\lambda \in \mathbb{R}$, $k \le n/c_x$ , and $t \in (1, n/c_x)$, if $A_k(XU, \lambda)$ is positive definite and $\rho_k = O\left(\frac{1}{L}\right)$, then it holds that with probability at least $1 - \delta - 20e^{-t/c_x}$,*

$$\frac{1}{m} \sum_{i=1}^{m} \mathbb{E}_x \left[ \left( x^\top \beta_{*i} - x^\top \hat{\beta}_i(I_d) \right)^2 \right] \lesssim B_{\text{Norm}} + V_{\text{Norm}},$$

$$\frac{B_{\text{Norm}}}{L^3 \max \left\{ L, \frac{\sum_{i>k} \lambda_i}{n\lambda + \sum_{i>k} \lambda_i} \right\}} = \sum_{i=1}^{k} \lambda_i \tilde{\sigma}_i^2 \frac{\rho_k^2 \lambda_{k+1}^2}{\lambda_i^2} + \sum_{i=k+1}^{d} \lambda_i \tilde{\sigma}_i^2, \qquad (4.1)$$

$$\frac{V_{\text{Norm}}}{L^2} = \frac{kt\sigma^2}{n} + \frac{t\sigma^2}{n} \sum_{i=k+1}^{d} \frac{\lambda_i^2}{\rho_k^2 \lambda_{k+1}^2}, \qquad (4.2)$$

*where $B_{\text{Norm}}$ corresponds to the bias term and $V_{\text{Norm}}$ corresponds to the variance term.*

This result is an extension of the result of Tsigler and Bartlett (2020) for multi-output linear regression. $k$ is called effective dimensionality (Bartlett et al., 2020; Tsigler and Bartlett, 2020), which stands for the effective complexity of the ridge regression estimator in high-dimensional settings. Notice that $\lambda$ in Corollary 1 can be negative. In fact, some negative $\lambda$ can show higher generalization performance than any $\lambda > 0$ under some conditions (Kobak et al., 2020; Wu and Xu). In particular, when $\lambda = 0$ this ridge estimator coincides with minimum norm interpolator (Bartlett et al., 2020; Liang and Rakhlin, 2020). In a similar way as Tsigler and Bartlett (2020), we can construct lower bounds of the bias $B_{\text{Norm}}$ and the variance $V_{\text{Norm}}$ (See Lemma 6 and Lemma 7 in Appendix C), and we can also show that the upper bounds and the lower bounds match under some conditions (See Appendix 3). Furthermore, we can demonstrate that the upper bound $U_{\text{NN}}$ of the selected model and that of normal ridge regression coincide when $\Sigma_\beta$ is isotropic (See Appendix G). This result is natural because when $\Sigma_\beta$ is isotropic, the ideal regularization is equivalent to normal $l_2$ regularization, that is, $W$ selected by the criterion $R(W)$ also becomes close to being isotropic, which indicates that the selected two-layer neural network comes close to normal ridge estimator. In the following part of this section, we discuss the condition that the selected two-layer linear neural network outperforms normal ridge regression comparing $U_{\text{NN}}$ and $B_{\text{Norm}} + V_{\text{Norm}}$. Note that since $B_{\text{Norm}}$ and $V_{\text{Norm}}$ achieve their lower bounds, this comparison is reasonable. For simplicity, we assume that $\Sigma_x$ and $\Sigma_\beta$ have the same eigenvectors.

## 4.1 WHEN $\lambda_i$ DECREASE SLOWLY

In this section, we show that there appears separation between feature learning by $R(W)$ and the naive ridge regression when the eigenvalues $\lambda_i$ decrease slowly. The situation where the decay of $\lambda_i$ is slow appears in the benign overfitting regime (Bartlett et al., 2020; Xu and Hsu, 2019). The following argument shows selecting $W$ by $R(W)$ is effective in such a setting. Indeed, for the case of normal ridge regression, observing Corollary 1, if $\lambda_i$ is large in comparison with $\rho_k \lambda_{k+1}$, $\frac{\rho_k^2 \lambda_{k+1}^2}{\lambda_i^2}$ in the first term of the bias $B_{\text{Norm}}$ can be seen as a weight that controls the impact of $\lambda_i \sigma_i^2$ to the bias. However, if the decay of $\lambda_i$ is gradual, the weight $\frac{\rho_k^2 \lambda_{k+1}^2}{\lambda_i^2}$ can be close to 1, thereby enhancing the effect of the bias. On the other hand, in the context of the two-layer linear neural network which is chosen based on $R(W)$, the upper bound of Theorem 2 is written as $U_{\text{NN}} = \sum_{i=1}^{d} \min \left\{ \frac{\sigma'^2}{n}, \lambda_i \sigma_i^2 \right\}$.

This bound implies that the effect of bias $\lambda_i \sigma_i^2$ is no larger than $\frac{\sigma'^2}{n}$ and this prevents the bias term from deteriorating the predictive risk of the model. The example below reflects the difference in the generalization performance of both models discussed above.

**Example 1.** *We suppose $\sqrt{n} \in \mathbb{N}$ and eigenvalues of $\Sigma_x$ decay slowly, that is, we consider the situation as follows:*

$$\lambda_i = \begin{cases} 1 & i \le n \\ \frac{1}{i+1-n} & i \ge n \end{cases}, \ \sigma_i = \frac{n}{\log n} e^{-i}, \ \sigma^2 = 1, \ \sigma'^2 = 1, d \ge 2n.$$

*Then for any $k < n$ in Corollary 1, it holds that*

$$B_{\text{Norm}} \gtrsim 1, \quad U_{\text{NN}} \lesssim \frac{\log n}{n}.$$

Example 1 claims that the two-layer linear neural network selected by $R(W)$ can generalize even when the decay of $\Sigma_x$ is slow and normal ridge regression can't generalize. This result reflects the

effect of the good regularization discussed above, using $W$ obtained by feature learning via $R(W)$. This allows for weak regularization in directions where signals make significant contributions, and strong regularization where the signals' contributions are minor.

## 4.2 WHEN $x$ AND $\beta_{*i}$ ARE MISALIGNED

In this section, we compare the predictive performance of the selected two-layer neural network and normal ridge regression when $x$ and $\beta_{*i}$ are misaligned. First, we introduce the notion of alignment and misalignment. We say that $x$ and $\beta_{*i}$ are aligned when the directions of large variance for $x$ and that of $\beta_{*i}$ closely match. In other words, an aligned situation is that the eigenvectors of $\Sigma_x$ corresponding to large eigenvalues and those of $\Sigma_\beta$ are in close directions. On the other hand, if the direction in which $x$ has large variance and that of $\beta_{*i}$ are different, that is, if eigenvectors of $\Sigma_x$ corresponding to large eigenvalues and those of $\Sigma_\beta$ are orthogonal, we refer to $x$ and $\beta_{*i}$ as being misaligned. We may define the alignment in a more rigorous way by considering the situation where $\Sigma_x$ and $\Sigma_\beta$ have the same eigenvectors. In this setting, for $\lambda_1 \geq \cdots \geq \lambda_d$ if $\sigma_1 \geq \cdots \geq \sigma_d$ then we say $x$ and $\beta_{*i}$ are aligned, and on the contrary, if $\sigma_1^2 \leq \cdots \leq \sigma_d^2$, then we say $x$ and $\beta_{*i}$ are misaligned. Here, we consider a misaligned situation, that is, $\sigma_1^2 \leq \cdots \leq \sigma_d^2$. For the case of normal ridge regression, for any $k < n$, we can write the bias term as $B_{\text{Norm}} = \sum_{i=1}^{k} \lambda_i \sigma_i^2 \frac{\rho_k^2 \lambda_{k+1}^2}{\lambda_i^2} + \sum_{i=k+1}^{d} \lambda_i \sigma_i^2$. Since $\sigma_1^2 \leq \cdots \leq \sigma_d^2$, $\sigma_{k+1}^2, \ldots, \sigma_d^2$ can be large and this leads to large bias. On the other hand, the two-layer neural network selected by $R(W)$ can decrease both bias and variance thanks to good regularization via learning with $R(W)$ as the example below demonstrates.

**Example 2.** *We suppose $x$ and $\beta_{*i}$ are misaligned, that is, we consider the situation as follows:*
$$\lambda_i = i^{-1}, \ \sigma_i^2 = ie^{i-d}, \ \sigma^2 = 1, \ \sigma'^2 = 1.$$

*Then for any $k < n$ in Corollary 1, it holds that*
$$B_{\text{Norm}} \gtrsim 1, \quad U_{\text{NN}} \lesssim \frac{\log n}{n}.$$

This result indicates that the estimator selected by $R(W)$ outperforms the normal ridge estimator in misaligned situations.

## 4.3 WHEN TAIL EIGENVALUES OF $\Sigma_x$ ARE LARGE

In this section, we show that the selected two-layer linear neural network outperforms normal ridge regression when tail eigenvalues of $\Sigma_x$ are large. In normal ridge regression, for any $k = o(n)$, the variance term is written as $\frac{V_{\text{Norm}}}{L^2 t \sigma^2} = \frac{k}{n} + \frac{1}{n \rho_k^2 \lambda_{k+1}^2} \sum_{i=k+1}^{d} \lambda_i^2$. We can see that if tail eigenvalues of $\Sigma_x$ are large, that is, if $\sum_{i>k} \lambda_i^2$ is large, then the second term of $V_{\text{Norm}}$ can also be large. This indicates that normal ridge regression doesn't generalize. On the other hand, because of feature learning via $R(W)$, the selected two-layer linear neural network can generalize even when the normal ridge regression can't generalize as the following example shows.

**Example 3.** *Suppose that tail eigenvalues of $\Sigma_x$ are large, that is, we consider the situation as follows:*
$$\lambda_i = \begin{cases} i^{-1} & i \leq n \\ \frac{1}{n} & i > n \end{cases}, \ \sigma_i^2 = i^{-2}, \ \sigma^2 = 1, \ \sigma'^2 = 1, \ d = n^3.$$

*Then for any $k < n$ in Corollary 1, it holds that*
$$B_{\text{Norm}} + V_{\text{Norm}} \gtrsim 1, \quad U_{\text{NN}} \lesssim n^{-\frac{2}{3}}.$$

Qualitatively, this result also reflects the good regularization obtained by $R(W)$. $W$ selected by $R(W)$ enables strong regularization in directions where signals don't contribute significantly, leading to a decrease in variance without a drastic increase in bias.

## 4.4 WHEN $y^{(i)}$ IS LARGE

In this section, we demonstrate that the selected two-layer linear neural network surpasses normal ridge regression when the norm of $y^{(i)}$ is notably large. In normal ridge regression, the bias can be written as $B_{\text{Norm}} = \sum_{i=1}^{k} \lambda_i \sigma_i^2 \frac{\rho_k^2 \lambda_{k+1}^2}{\lambda_i^2} + \sum_{i=k+1}^{d} \lambda_i \sigma_i^2$ and the weights $\frac{\rho_k^2 \lambda_{k+1}^2}{\lambda_i^2}$ in the first term contribute to reducing the impact on the bias term. However, if $y^{(i)}$ is large, that is, $\lambda_i \sigma_i^2$ is large, then

the bias term doesn't vanish even if the weight is small. This means that an isotropic regularization is not enough to effectively decrease the effect of bias in such situations. On the contrary, the selected model with good regularization via feature learning by $R(W)$ is capable of bringing both the bias and variance close to zero. The following example demonstrates the difference between the selected two-layer neural network and normal ridge regression.

**Example 4.** *We suppose $n^{\frac{2}{3}} \in \mathbb{N}$ and norm of $y^{(i)}$ is large, that is, we consider the situation as follows:*

$$\lambda_i = i^{-1}, \ \sigma_i^2 = \left\{ \begin{array}{ll} n^{\frac{3}{2}} & i \leq n^{\frac{2}{3}} \\ i^{-1} & i > n^{\frac{2}{3}} \end{array} \right. , \ \sigma^2 = 1, \ \sigma'^2 = 1.$$

*Then for any $k < n$ in Corollary 1, it holds that*

$$B_{\text{Norm}} + V_{Norm} \gtrsim 1, \quad U_{\text{NN}} \lesssim n^{-\frac{1}{3}}.$$

## 5 NUMERICAL EXPERIMENTS

In this section, we conduct numerical experiments to justify our theoretical results. We consider the setting that $(d, n, m) = (1,000, 500, 1,000)$, $\Sigma_x = \text{diag}\left(1, 2^{-1}, \ldots, 1,000^{-1}\right)$, $x \sim \mathcal{N}(0, \Sigma_x)$, $\Sigma_\beta = \text{diag}\left(1, 2^{-a}, \ldots, 1,000^{-a}\right)$ ($a = 0, 0.5, 1, \ldots, 10$), $(\sigma^2, \sigma'^2) = (1, 2)$ and $\epsilon_i^{(j)} \sim_{\text{i.i.d.}} \mathcal{N}(0, \sigma^2)$. Since the criterion $R(W)$ is non-convex, we optimized this function with gradient Langevin dynamics, which is a noisy gradient descent method aiming to escape from sharp local minima. For a random variable matrix $\xi_t$ which satisfies $(\xi_t)_{i,j} \sim_{\text{i.i.d.}} \mathcal{N}(0, 1)$, learning rate $\eta$ and inverse temperature parameter $\beta$, the update of the gradient Langevin dynamics can be written as $W^{(t+1)} = W^{(t)} - \eta \nabla R\left(W^{(t)}\right) + \sqrt{\frac{2\eta}{\beta}} \xi_t$. In this experiment, we set $(\eta, \beta) = (0.1, 100, 000)$ and iterated the update 500 times and selected $W$ corresponding to the smallest $R(W)$. Figure 1 shows the comparison of the predictive risks with a base-10 logarithm between the obtained two-layer linear neural network, normal ridge regression, and Bayes-optimal estimator. The regularization parameters for ridge regression were chosen from $1.0, 0.5, 0.1, 0.01, 0.001, 0.0001$ to minimize the predictive risk calculated using additional data. When $a$ is small and the eigenvalues of $\Sigma_\beta$ decay slowly, the proposed method has equivalent performance to or falls below the ridge regression. However, as $a$ becomes larger and the eigenvalues of $\Sigma_\beta$ decay more quickly, it can be seen that the proposed method begins to outperform ridge regression. This reflects the fact that the more isotropic $\Sigma_\beta$ is, the more isotropic the optimal parameter of the first layer becomes, hence the benefit of finding informative directions by feature learning is diminished in such cases. 2 shows the minimum values of the proposed criterion $R(W) - \sigma^2$ with a base-10 logarithm obtained by gradient Langevin dynamics and predictive risks with a base-10 logarithm of the two-layer linear neural network with corresponding $W$. We can see that the behavior of the function values of $R(W) - \sigma^2$ captures that of predictive risk. This result shows that the proposed criterion $R(W) - \sigma^2$ can play a role of an estimator for an upper bound of the predictive risk.

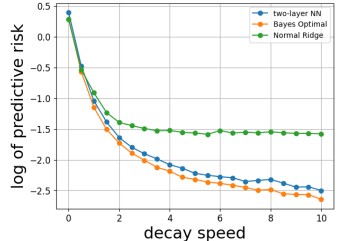
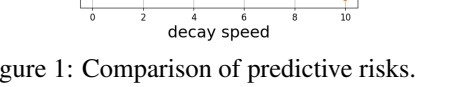
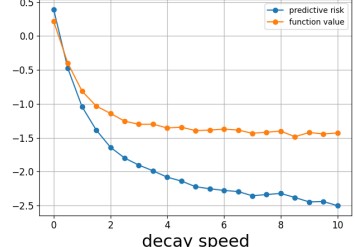

Figure 1: Comparison of predictive risks.    Figure 2: $R(W) - \sigma^2$ and predictive risk.

## 6 CONCLUSION AND FUTURE WORK

In this paper, we have considered the feature learning of a multi-output linear regression problem using a two-layer linear neural network. We have proposed a criterion with penalty term on feature mapping $W$ like Mallows' $C_p$ and have shown that an estimator selected by the criterion leads to making its predictive risk close to zero. Then, we have shown the optimized model by the proposed criterion can achieve nearly optimal Bayes risk. Furthermore, we have characterized when normal

ridge regression doesn't generalize and shown that the selected two-layer linear neural network generalizes even in such situations. We state the future directions of this research. On the optimization aspect, we have to guarantee that we can minimize the proposed criterion well. It is not obvious whether we can obtain the optimal solution by standard optimization techniques such as stochastic gradient descent because the objective is non-convex. Another future direction is replacing the activation function. Although we considered identity in this paper, we usually use nonlinear activation functions such as ReLU in real applications.

## 7 ACKNOWLEDGEMENT

KS was partially supported by JST CREST (JPMJCR2015). TS was partially supported by JSPS KAKENHI (20H00576) and JST CREST (JPMJCR2115).

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
