## A    PROOF OF THEOREM 1

*Proof.* We define $X_w = XW^\top$, $\Sigma_{wx} = W\Sigma_x W^\top$, $\hat{\Sigma}_{wx} = W\hat{\Sigma}_x W^\top$ and $\Sigma_{w\beta} = W^{-\top}\Sigma_\beta W^{-1}$. Accordingly, we introduce the following notations:

- Let $\hat{\lambda}_1 \geq \cdots \geq \hat{\lambda}_d$ be eigenvalues of $\hat{\Sigma}_{wx}$ and $\hat{u}_1, \ldots, \hat{u}_d$ be corresponding eigenvectors. Then, we define $\hat{U} = (\hat{u}_1, \ldots, \hat{u}_d)$.

- Let $\hat{\Sigma}_{\hat{u}x} := \hat{U}^\top \hat{\Sigma}_{wx} \hat{U} = \text{diag}\left(\hat{\lambda}_1, \ldots, \hat{\lambda}_d\right)$.

- Let $\Sigma_{\hat{u}x} := \hat{U}^\top \Sigma_{wx} \hat{U}$.

- Let $\beta_{wi} = W^{-\top}\beta_{*i}$ and $\Sigma_{w\beta} = W^{-\top}\Sigma_\beta W^{-1}$.

- Let $\hat{\sigma}_i^2 := \hat{u}_i^\top \Sigma_{w\beta} \hat{u}_i$, and $\Sigma_{\hat{u}\beta} := \hat{U}^\top \Sigma_{w\beta} \hat{U}$.

- Let $A_k(X_w\hat{U}, \lambda) = \left(X_w\hat{U}\right)_{-k}\left(X_w\hat{U}\right)_{-k}^\top + n\lambda I_n$

From Proposition 2, there exists a constant $c_x$ which only depends on $\sigma_x$ such that for any $t$ and $k$ which satisfies $t \in (1, n/c_x)$ and $\sqrt{k} + \sqrt{t} \leq \sqrt{n}/c_x$, if $\text{cond}\left(A_k(X_w\hat{U}, \lambda)\right) \leq L$, then it holds that with probability at least $1 - 20e^{-t/c_x}$,

$$\frac{1}{m}\sum_{i=1}^{m}\mathbb{E}_x\left[\left(x^\top \beta_{*i} - x^\top W^\top \hat{\beta}_i(W, \lambda)\right)^2\right] \lesssim_{\sigma_x} B + V,$$

$$\frac{B}{L^4} = \text{Tr}\left((\Sigma_{\hat{u}x})_{k,k}^{-1}(\Sigma_{\hat{u}\beta})_{-k,-k}\right)\left(\lambda + \frac{1}{n}\text{Tr}\left((\Sigma_{\hat{u}x})_{-k,-k}\right)\right)^2$$
$$+ \text{Tr}\left((\Sigma_{\hat{u}x})_{-k,-k}(\Sigma_{\hat{u}\beta})_{-k,-k}\right), \tag{A.1}$$

$$\frac{V}{L^2t\sigma^2} = \frac{k}{n} + \frac{1}{n}\frac{\text{Tr}\left((\Sigma_{\hat{u}x})_{-k,-k}^2\right)}{\left(\lambda + \frac{1}{n}\text{Tr}\left((\Sigma_{\hat{u}x})_{-k,-k}\right)\right)^2}. \tag{A.2}$$

From Lemma 2, there exist constant $c_b, C_b$ which only depend on $\sigma_x$ such that it holds with probability at least $1 - 2e^{-c_b n}$ for any $k' < n$,

$$\left(\frac{3}{2} + C_b\sqrt{\frac{k'}{n}}\right)^{-1}\left(\text{Tr}\left(\left((\Sigma_{\hat{u}x})_{k',k'} + \lambda I_k'\right)^{-1}(\Sigma_{\hat{u}x})_{k',k'}(\Sigma_{\hat{u}\beta})_{k',k'}\right)\right)$$

$$\leq \text{Tr}\left(\left(\left(\hat{\Sigma}_{\hat{u}x}\right)_{k',k'} + \lambda I_k'\right)^{-1}\left(\hat{\Sigma}_{\hat{u}x}\right)_{k',k'}(\Sigma_{\hat{u}\beta})_{k',k'}\right)$$

$$\leq \left(\frac{3}{2} + C_b\sqrt{\frac{k'}{n}}\right)\left(\text{Tr}\left(\left((\Sigma_{\hat{u}x})_{k',k'} + \lambda I_k'\right)^{-1}(\Sigma_{\hat{u}x})_{k',k'}(\Sigma_{\hat{u}\beta})_{k',k'}\right)\right). \tag{A.3}$$

For these $c_x$ and $C_b$, we assume that it holds $R(W) - \sigma^2 \leq \frac{\sigma'^2}{\max\{8c_x^2, 8C_b^2\}}$. Then, we define $k_w$ such that $\mu_{k_w}\left(\hat{\Sigma}_{wx}\right) \leq \lambda \leq \mu_{k_w+1}\left(\hat{\Sigma}_{wx}\right)$. Suppose $k_w > \frac{n}{\max\{4c_x^2, 4C_b^2\}}$, then

$$R(W) - \sigma^2 \geq \frac{\sigma'^2}{n}\text{Tr}\left(\hat{\Sigma}_{wx}\left(\hat{\Sigma}_{wx} + \lambda I_d\right)^{-1}\right)$$

$$= \frac{\sigma'^2}{n}\sum_{i=1}^{n}\frac{\mu_i\left(\hat{\Sigma}_{wx}\right)}{\lambda + \mu_i\left(\hat{\Sigma}_{wx}\right)}$$

$$\geq \frac{\sigma'^2}{n}\sum_{i=1}^{k_w}\frac{\mu_i\left(\hat{\Sigma}_{wx}\right)}{\lambda + \mu_i\left(\hat{\Sigma}_{wx}\right)}$$

$$\geq \frac{\sigma'^2}{2n}k_w$$

$$> \frac{\sigma'^2}{\max\{8c_x^2, 8C_b^2\}}.$$

This contradicts with the assumption above, and therefore it holds $k_w \leq \frac{n}{\max\{4c_x^2, 4C_b^2\}}$. We set $k = k_w$ and $k' = k_w$. Since $k_w \leq \frac{n}{4c_x^2}$, if $t$ satisfies $\sqrt{n}/2c_x + \sqrt{t} \leq \sqrt{n}/c_x$, then $t$ also satisfies $\sqrt{k_w} + \sqrt{t} \leq \sqrt{n}/c_x$. Combining with $t \in (1, n/c_x)$, the condition on $t$ can be written as $t \in \left(1, \min\{n/c_x, n/4c_x^2\}\right)$. Furthermore, since $k_w \leq \frac{n}{4C_b^2}$, we can also evaluate as

$$\frac{1}{2}\left(\text{Tr}\left(\left((\Sigma_{\hat{u}x})_{k_w,k_w} + \lambda I_k'\right)^{-1}(\Sigma_{\hat{u}x})_{k_w,k_w}(\Sigma_{\hat{u}\beta})_{k_w,k_w}\right)\right)$$

$$\leq \text{Tr}\left(\left(\left(\hat{\Sigma}_{\hat{u}x}\right)_{k_w,k_w} + \lambda I_k'\right)^{-1}\left(\hat{\Sigma}_{\hat{u}x}\right)_{k_w,k_w}(\Sigma_{\hat{u}\beta})_{k_w,k_w}\right)$$

$$\leq \quad 2\left(\mathrm{Tr}\left(\left((\Sigma_{\hat{u}x})_{k_w,k_w} + \lambda I'_k\right)^{-1}(\Sigma_{\hat{u}x})_{k_w,k_w}(\Sigma_{\hat{u}\beta})_{k_w,k_w}\right)\right).$$

From Lemma 3, there exists constant $c_1$ such that with probability at least $1 - 4e^{-c_1 n}$, it holds

$$\lambda + \frac{1}{n}\mathrm{Tr}\left((\Sigma_{\hat{u}x})_{-k_w,-k_w}\right) \lesssim \lambda \lesssim \mu_k\left((\Sigma_{\hat{u}x})_{k_w,k_w}\right).$$

Therefore, we can evaluate the first term of A.1 as

$$\mathrm{Tr}\left((\Sigma_{\hat{u}x})_{k_w,k_w}^{-1}(\Sigma_{\hat{u}\beta})_{k_w,k_w}\right)\left(\lambda + \frac{1}{n}\mathrm{Tr}\left((\Sigma_{\hat{u}x})_{-k_w,-k_w}\right)\right)^2$$

$$\lesssim \quad \lambda^2\mathrm{Tr}\left((\Sigma_{\hat{u}x})_{k_w,k_w}^{-1}(\Sigma_{\hat{u}\beta})_{k_w,k_w}\right)$$

$$= \quad \sum_{i=1}^{k_w}\mu_i\left((\Sigma_{\hat{u}x})_{k_w,k_w}\right)\hat{\sigma}_{k_w,i}^2\frac{\lambda^2}{\mu_i\left((\Sigma_{\hat{u}x})_{k_w}\right)^2}$$

$$\lesssim \quad \sum_{i=1}^{k_w}\mu_i\left((\Sigma_{\hat{u}x})_{k_w,k_w}\right)\hat{\sigma}_{k_w,i}^2\frac{\lambda^2}{\left(\mu_i\left((\Sigma_{\hat{u}x})_{k_w,k_w}\right) + \lambda\right)^2}$$

$$\leq \quad \sum_{i=1}^{k_w}\mu_i\left((\Sigma_{\hat{u}x})_{k_w,k_w}\right)\hat{\sigma}_{k_w,i}^2\frac{\lambda}{\mu_i\left((\Sigma_{\hat{u}x})_{k_w,k_w}\right) + \lambda}$$

$$= \quad \lambda\mathrm{Tr}\left(\left((\Sigma_{\hat{u}x})_{k_w,k_w} + \lambda I_{k_w}\right)^{-1}(\Sigma_{\hat{u}x})_{k_w,k_w}(\Sigma_{\hat{u}\beta})_{k_w,k_w}\right).$$

Therefore, we can evaluate (A.1) as

$$\frac{B}{L^4} \quad \lesssim \quad \lambda\mathrm{Tr}\left(\left((\Sigma_{\hat{u}x})_{k_w,k_w} + \lambda I_{k_w}\right)^{-1}(\Sigma_{\hat{u}x})_{k_w,k_w}(\Sigma_{\hat{u}\beta})_{k_w,k_w}\right)$$

$$+ \quad \mathrm{Tr}\left((\Sigma_{\hat{u}x})_{-k_w,-k_w}(\Sigma_{\hat{u}\beta})_{-k_w,-k_w}\right).$$

By Lemma 11, it holds that with probability at least $1 - 2e^{-c_2 t}$,

$$\mathrm{Tr}\left(\left(\hat{\Sigma}_{\hat{u}x}\right)_{-k_w,-k_w}(\Sigma_{\hat{u}\beta})_{-k_w,-k_w}\right)$$

$$\geq \quad \left(1 - \frac{t}{n}\sigma_x^2\right)\mathrm{Tr}\left((\Sigma_{\hat{u}x})_{-k_w,-k_w}(\Sigma_{\hat{u}\beta})_{-k_w,-k_w}\right).$$

Combining (A) and these two inequalities, we have

$$\frac{B}{L^4} \quad \lesssim_{\sigma_x} \quad \mathrm{Tr}\left(\left(\left(\hat{\Sigma}_{\hat{u}x}\right)_{k_w,k_w} + \lambda I_{k_w}\right)^{-1}\left(\hat{\Sigma}_{\hat{u}x}\right)_{k_w,k_w}(\Sigma_{\hat{u}\beta})_{k_w,k_w}\right)$$

$$+ \mathrm{Tr}\left(\left(\hat{\Sigma}_{\hat{u}x}\right)_{-k_w,-k_w}(\Sigma_{\hat{u}\beta})_{-k_w,-k_w}\right).$$

By the way, since $\hat{\lambda}_{k_w+1} \leq \lambda \leq \hat{\lambda}_{k_w}$ by the definition of $k_w$, we can evaluate as follows:

$$\frac{\lambda}{m}\sum_{i=1}^{m}\beta_{wi}^{\top}X_w^{\top}\left(X_wX_w^{\top} + n\lambda I_n\right)^{-1}X_w\beta_{wi}$$

$$= \quad \lambda\mathrm{Tr}\left(\left(\hat{\Sigma}_{wx} + \lambda I_d\right)^{-1}\hat{\Sigma}_{wx}\Sigma_{w\beta}\right)$$

$$= \quad \sum_{i=1}^{d}\frac{\lambda}{\lambda + \hat{\lambda}_i}\hat{\lambda}_i\hat{\sigma}_i^2$$

$$\gtrsim \quad \sum_{i=1}^{k_w}\frac{\lambda}{\lambda + \hat{\lambda}_i}\hat{\lambda}_i\hat{\sigma}_i^2 + \sum_{i=k_w+1}^{d}\hat{\lambda}_i\hat{\sigma}_i^2$$

$$= \lambda \text{Tr} \left( \left( \left( \hat{\Sigma}_{\hat{u}x} \right)_{k_w,k_w} + \lambda I_{k_w} \right)^{-1} \left( \hat{\Sigma}_{\hat{u}x} \right)_{k_w,k_w} \left( \Sigma_{\hat{u}\beta} \right)_{k_w,k_w} \right)$$

$$+ \text{Tr} \left( \left( \hat{\Sigma}_{\hat{u}x} \right)_{-k_w,-k_w} \left( \Sigma_{\hat{u}\beta} \right)_{-k_w,-k_w} \right).$$

Combining these two inequalities, we have

$$\frac{B}{L^4} \lesssim_{\sigma_x} \frac{\lambda}{m} \sum_{i=1}^{m} \beta_{wi}^\top X_w^\top \left( X_w X_w^\top + n\lambda I_n \right)^{-1} X_w \beta_{wi}. \tag{A.4}$$

For the variance term (A.2), since $\lambda > 0$ and $\hat{\lambda}_{k_w+1} \leq \lambda$, we can evaluate as

$$\frac{V}{L^2 t \sigma^2} = \frac{k_w}{n} + \frac{1}{n} \frac{\text{Tr} \left( (\Sigma_{\hat{u}x})_{-k_w,-k_w} \left( \hat{\Sigma}_{\hat{u}x} \right)_{-k_w,-k_w} \right)}{\left( \lambda + \frac{1}{n} \text{Tr} \left( (\Sigma_{\hat{u}x})_{-k_w,-k_w} \right) \right)^2}$$

$$\leq \frac{k_w}{n} + \frac{\lambda}{n\lambda^2} \text{Tr} \left( (\Sigma_{\hat{u}x})_{-k_w,-k_w} \right)$$

$$= \frac{k_w}{n} + \frac{1}{n\lambda} \text{Tr} \left( (\Sigma_{\hat{u}x})_{-k_w,-k_w} \right).$$

In the same way as the bias, we can evaluate the variance as follows:

$$\frac{\sigma'^2}{n} \text{Tr} \left( \hat{\Sigma}_{wx} \left( \hat{\Sigma}_{wx} + \lambda I_d \right)^{-1} \right) = \frac{\sigma'^2}{n} \sum_{i=1}^{d} \frac{\hat{\lambda}_i}{\hat{\lambda}_i + \lambda}$$

$$\gtrsim \frac{k_w \sigma'^2}{n} + \frac{\sigma'^2}{n} \sum_{i=k_w+1}^{d} \frac{\hat{\lambda}_i}{\hat{\lambda}_i + \lambda}$$

$$\geq \frac{k_w \sigma'^2}{n} + \frac{\sigma'^2}{n\lambda} \sum_{i=k_w+1}^{d} \hat{\lambda}_i$$

$$= \frac{k_w \sigma'^2}{n} + \frac{\sigma'^2}{n\lambda} \text{Tr} \left( \left( \hat{\Sigma}_{\hat{u}x} \right)_{-k_w,-k_w} \right).$$

By Lemma 11, it holds that with probability at least $1 - 2e^{-c_3 t}$,

$$\text{Tr} \left( \left( \hat{\Sigma}_{\hat{u}x} \right)_{-k_w,-k_w} \right) \geq \left( 1 - \frac{t}{n} \sigma_x^2 \right) \text{Tr} \left( (\Sigma_{\hat{u}x})_{-k_w,-k_w} \right).$$

Combining these three inequalities, if $t\sigma^2 \lesssim \sigma'^2$, we have

$$\frac{V}{L^2 t \sigma^2} \lesssim_{\sigma_x} \frac{\sigma'^2}{n} \text{Tr} \left( \hat{\Sigma}_{wx} \left( \hat{\Sigma}_{wx} + \lambda I_d \right)^{-1} \right). \tag{A.5}$$

Let $\bar{U} \in \mathbb{R}^{n \times n}$ be eigenvectors of $X_w X_w^\top$. From Lemma 1 and Lemma 13, for any $t \in (0, n)$ it holds that with probability at least $1 - \frac{(\log 2)(1 + \sigma_x^2)\sigma^2 \max\|\beta_{*i}\|_{\Sigma_x}^2}{n\delta^2} - 2e^{-c_4 t} - 2e^{-c_5 n}$,

$$R(W)$$

$$= \frac{1}{m} \sum_{i=1}^{m} \min_{\beta \in \mathbb{R}^l} \frac{1}{n} \left\| \mathbf{y}^{(i)} - X_w \beta \right\|^2 + \lambda \|\beta\|^2 + \frac{\sigma'^2}{n} \text{Tr} \left( \hat{\Sigma}_{wx} \left( \hat{\Sigma}_{wx} + \lambda I_d \right)^{-1} \right)$$

$$= \frac{\lambda}{m} \sum_{i=1}^{m} \mathbf{y}^{(i)\top} \left( X_w X_w^\top + n\lambda I_n \right)^{-1} \mathbf{y}^{(i)} + \frac{\sigma'^2}{n} \text{Tr} \left( \hat{\Sigma}_{wx} \left( \hat{\Sigma}_{wx} + \lambda I_d \right)^{-1} \right)$$

$$= \frac{\lambda}{m} \sum_{i=1}^{m} \beta_{*i}^\top X^\top \left( X_w X_w^\top + n\lambda I_n \right)^{-1} X \beta_{*i} + \frac{2\lambda}{m} \sum_{i=1}^{m} \epsilon^{(i)\top} \left( X_w X_w^\top + n\lambda I_n \right)^{-1} X \beta_{*i}$$

$$+\frac{\lambda}{m}\sum_{i=1}^{m}\epsilon^{(i)\top}\left(X_wX_w^\top+n\lambda I_n\right)^{-1}\epsilon^{(i)}+\frac{\sigma'^2}{n}\mathrm{Tr}\left(\hat{\Sigma}_{wx}\left(\hat{\Sigma}_{wx}+\lambda I_d\right)^{-1}\right)$$

$$=\quad\frac{\lambda}{m}\sum_{i=1}^{m}\beta_{*i}^\top X^\top\left(X_wX_w^\top+n\lambda I_n\right)^{-1}X\beta_{*i}+\frac{2\lambda}{m}\sum_{i=1}^{m}\epsilon^{(i)\top}\left(X_wX_w^\top+n\lambda I_n\right)^{-1}X\beta_{*i}$$

$$+\frac{\lambda}{nm}\sum_{i=1}^{m}\epsilon^{(i)\top}\left(\hat{\Sigma}_{wx}+\lambda I_n\right)^{-1}\epsilon^{(i)}+\frac{\sigma'^2}{n}\mathrm{Tr}\left(\hat{\Sigma}_{wx}\left(\hat{\Sigma}_{wx}+\lambda I_d\right)^{-1}\right)$$

$$=\quad\frac{\lambda}{m}\sum_{i=1}^{m}\beta_{*i}^\top X^\top\left(X_wX_w^\top+n\lambda I_n\right)^{-1}X\beta_{*i}+\frac{2\lambda}{m}\sum_{i=1}^{m}\epsilon^{(i)\top}\left(X_wX_w^\top+n\lambda I_n\right)^{-1}X\beta_{*i}$$

$$+\frac{1}{nm}\sum_{i=1}^{m}\sum_{j=1}^{d}\frac{\lambda}{\lambda+\hat{\lambda}_j}\epsilon^{(i)\top}\bar{U}_j^\top\bar{U}_j\epsilon^{(i)}+\frac{\sigma'^2}{n}\mathrm{Tr}\left(\hat{\Sigma}_{wx}\left(\hat{\Sigma}_{wx}+\lambda I_d\right)^{-1}\right)$$

$$\geq\quad\frac{\lambda}{m}\sum_{i=1}^{m}\beta_{*i}^\top X^\top\left(X_wX_w^\top+n\lambda I_n\right)^{-1}X\beta_{*i}+\frac{2\lambda}{m}\sum_{i=1}^{m}\epsilon^{(i)\top}\left(X_wX_w^\top+n\lambda I_n\right)^{-1}X\beta_{*i}$$

$$+\frac{1}{nm}\sum_{i=1}^{m}\sum_{j=k_w+1}^{d}\frac{\lambda}{\lambda+\hat{\lambda}_j}\epsilon^{(i)\top}\bar{U}_j^\top\bar{U}_j\epsilon^{(i)}+\frac{\sigma'^2}{n}\mathrm{Tr}\left(\hat{\Sigma}_{wx}\left(\hat{\Sigma}_{wx}+\lambda I_d\right)^{-1}\right)$$

$$\gtrsim\quad\frac{\lambda}{m}\sum_{i=1}^{m}\beta_{*i}^\top X^\top\left(X_wX_w^\top+n\lambda I_n\right)^{-1}X\beta_{*i}+\frac{2\lambda}{m}\sum_{i=1}^{m}\epsilon^{(i)\top}\left(X_wX_w^\top+n\lambda I_n\right)^{-1}X\beta_{*i}$$

$$+\frac{1}{nm}\sum_{i=1}^{m}\sum_{j=k_w+1}^{d}\epsilon^{(i)\top}\bar{U}_j^\top\bar{U}_j\epsilon^{(i)}+\frac{\sigma'^2}{n}\mathrm{Tr}\left(\hat{\Sigma}_{wx}\left(\hat{\Sigma}_{wx}+\lambda I_d\right)^{-1}\right)$$

$$\gtrsim\quad\frac{\lambda}{m}\sum_{i=1}^{m}\beta_{*i}^\top X^\top\left(X_wX_w^\top+n\lambda I_n\right)^{-1}X\beta_{*i}+\frac{2\lambda}{m}\sum_{i=1}^{m}\epsilon^{(i)\top}\left(X_wX_w^\top+n\lambda I_n\right)^{-1}X\beta_{*i}$$

$$+\sigma^2+\frac{\sigma'^2}{n}\mathrm{Tr}\left(\hat{\Sigma}_{wx}\left(\hat{\Sigma}_{wx}+\lambda I_d\right)^{-1}\right)$$

$$\geq\quad\frac{\lambda}{m}\sum_{i=1}^{m}\beta_{*i}^\top X^\top\left(X_wX_w^\top+n\lambda I_n\right)^{-1}X\beta_{*i}+\frac{\sigma'^2}{n}\mathrm{Tr}\left(\hat{\Sigma}_{wx}\left(\hat{\Sigma}_{wx}+\lambda I_d\right)^{-1}\right)$$

$$+\sigma^2-2\delta.$$

Therefore, if $t\sigma^2\lesssim\sigma'^2$, combining with A.4 and A.5 we have

$$\frac{B}{L^4}+\frac{V}{L^2}\lesssim_{\sigma_x}\max\{R(W)-\sigma^2,\delta\}.$$

Finally, we need to check $\mathrm{cond}\left(A_{k_w}\left(XW^\top\hat{U},\lambda\right)\right)=\Theta(1)$. Since

$$\frac{1}{n}\mu_{\max}\left(X_w\hat{U}_{-k_w}^\top\hat{U}_{-k_w}X_w^\top\right)=\mu_{\max}\left(\hat{\Sigma}_{\hat{u}x}\right)=\frac{1}{n}S_{\max}\left(X_w\hat{U}_{-k_w}^\top\right)^2\leq\lambda,$$

we have

$$\mathrm{cond}\left(A_{k_w}\left(XW^\top\hat{U},\lambda\right)\right)\leq\frac{n\lambda+n\mu_{\max}\left(X_w\hat{U}_{-k_w}^\top\hat{U}_{-k_w}X_w^\top\right)}{n\lambda}\leq\frac{n\lambda+n\lambda}{n\lambda}\leq2\,(=L).$$

Therefore, it holds

$$B+V\lesssim_{\sigma_x}\max\{R(W)-\sigma^2,\delta\}.$$

$\square$

In the later part of this section, we show some lemmas needed to prove Theorem 1.

**Lemma 1.** *In the setting of Theorem 1, there exists a constant $c > 0$ which only depends on $\sigma_x$ such that with probability at least $1 - \frac{(\log 2)(1+\sigma_x^2)\sigma^2 \max\|\beta_{*i}\|_{\Sigma_x}^2}{n\delta^2} - 2e^{-cn}$,*

$$\frac{\lambda}{m}\sum_{i=1}^{m} \epsilon^{(i)\top}\left(X_w X_w^\top + n\lambda I_n\right)^{-1} X\beta_{*i} < \delta.$$

*Proof.* For any $i$, $\epsilon^{(i)\top}\left(X_w X_w^\top + n\lambda I_n\right)^{-1} X\beta_{*i}$ is mean 0 and variance $\mathrm{Var}\left[\epsilon_1^{(1)}\right] \left\|\left(X_w X_w^\top + n\lambda I_n\right)^{-1} X\beta_{*i}\right\|^2$ random variable. We define index $I(W, \lambda)$ as follows:

$$I(W, \lambda) = \operatorname*{argmin}_{i} \left|\epsilon^{(i)\top}\left(X_w X_w^\top + n\lambda I_n\right)^{-1} X\beta_{*i}\right|.$$

Since $\epsilon_j^{(i)}$ are sub-gaussian with the sub-gaussian norm at most $\sigma$, $\mathrm{Var}\left[\epsilon_j^{(i)}\right] \leq (\log 2)\sigma^2$. By Chebyshev's inequality, we obtain

$$\mathrm{Pr}\left(\left|\left(\epsilon^{I(W,\lambda)}\right)^\top \left(X_w X_w^\top + n\lambda I_n\right)^{-1} X\beta_{*I(W,\lambda)}\right| < \delta\right)$$

$$> \quad 1 - \frac{\mathrm{Var}\left[\epsilon_1^{(1)}\right]\left\|\left(X_w X_w^\top + n\lambda I_n\right)^{-1} X\beta_{*I(W,\lambda)}\right\|^2}{\delta^2}$$

$$> \quad 1 - \frac{\mathrm{Var}\left[\epsilon_1^{(1)}\right]\left\|X\beta_{*I(W,\lambda)}\right\|^2}{n^2\lambda^2\delta^2}$$

$$> \quad 1 - \frac{(\log 2)\sigma^2\left\|X\beta_{*I(W,\lambda)}\right\|^2}{n^2\lambda^2\delta^2}.$$

From Lemma 11, there exists a constant $c$ such that it holds with probability at least $1 - 2e^{-cn}$,

$$\mathrm{Pr}\left(\frac{1}{n}\|X\beta_{*I(W,\lambda)}\|^2 < \left(1 + \sigma_x^2\right)\max_i\|\beta_{*i}\|_{\Sigma_x}^2\right)$$

$$> \quad \mathrm{Pr}\left(\frac{1}{n}\|X\beta_{*I(W,\lambda)}\|^2 < \left(1 + \frac{t}{n}\sigma_x^2\right)\|\beta_{*I(W,\lambda)}\|_{\Sigma_x}^2\right)$$

$$\geq \quad 1 - 2e^{-ct}.$$

Combining these inequalities, it holds that with probability at least $1 - 2e^{-cn}$,

$$\mathrm{Pr}\left(\left|\left(\epsilon^{I(W,\lambda)}\right)^\top \left(X_w X_w^\top + n\lambda I_n\right)^{-1} X\beta_{*I(W,\lambda)}\right| < \delta\right)$$

$$> \quad 1 - \left(1 + \sigma_x^2\right)\frac{(\log 2)\sigma^2\max_i\|\beta_{*i}\|_{\Sigma_x}^2}{n\lambda^2\delta^2}.$$

Notice that the right hand side doesn't depend on the choice of $i$ and $W$. Therefore, it holds that with probability at least $1 - \left(1 + \sigma_x^2\right)\frac{(\log 2)\sigma^2\max_i\|\beta_{*i}\|_{\Sigma_x}^2}{n\lambda^2\delta^2} - 2e^{-ct}$,

$$\frac{\lambda}{m}\sum_{i=1}^{m} \epsilon^{(i)\top}\left(X_w X_w^\top + n\lambda I_n\right)^{-1} X\beta_{*i} < \lambda\left(\epsilon^{I(W,\lambda)}\right)^\top\left(X_w X_w^\top + n\lambda I_n\right)^{-1} X\beta_{*I(W,\lambda)} < \lambda\delta.$$

Finally, changing $\delta \to \lambda\delta$ gives the result. $\qquad\square$

**Lemma 2.** *Let $x_1, \ldots, x_n$ be i.i.d. centered random vectors in $\mathbb{R}^l$ with covariance matrix $\Sigma$, $X = (x_1, \ldots, x_n) \in \mathbb{R}^{n\times l}$ and $\hat{\Sigma} = \frac{1}{n}X^\top X$. If $x_i$ is sub-Gaussian with $\|x_i\|_{\psi_2} \leq \sigma_x$, then for any $\beta_1, \ldots, \beta_m \in \mathbb{R}^l$ and $\lambda > 0$, there exist constants $c, C$ which only depend on $\sigma_x$ such that it holds with probability at least $1 - 2e^{-cn}$,*

$$\left(\frac{3}{2} + C\sqrt{\frac{l}{n}}\right)^{-1}\frac{\lambda}{m}\sum_{i=1}^{m}\beta_{*i}^\top\left(\Sigma + \lambda I_d\right)^{-1}\Sigma\beta_{*i}$$

$$\leq \quad \frac{\lambda}{m} \sum_{i=1}^{m} \beta_{*i}^{\top} \left(\hat{\Sigma} + \lambda I_d\right)^{-1} \hat{\Sigma} \beta_{*i}$$

$$\leq \quad \left(\frac{3}{2} + C\sqrt{\frac{l}{n}}\right) \frac{\lambda}{m} \sum_{i=1}^{m} \beta_{*i}^{\top} \left(\Sigma + \lambda I_d\right)^{-1} \Sigma \beta_{*i}$$

*Proof.* For a positive semidefinite matrix $A \in \mathbb{R}^{l \times l}$, we define

$$
\begin{aligned}
R_i(\beta, A) &:= \|\beta_i - \beta\|_A^2 + \lambda \|\beta\|^2, \\
\hat{\beta}_i(A) &:= (A + \lambda I_l)^{-1} A \beta_i.
\end{aligned}
$$

By Lemma 12, there exist constants $c, C$ which only depends on $\sigma_x$ such that with probability at least $1 - 2e^{-cn}$ for any $i = 1, \dots, m$, it holds

$$
\begin{aligned}
& \beta_{*i}^{\top} \left(\Sigma + \lambda I_d\right)^{-1} \Sigma \beta_{*i} \\
=\ & \min_{\beta \in \mathbb{R}^l} R_i(\beta, \Sigma) \\
=\ & R_i\left(\hat{\beta}_i(\Sigma), \Sigma\right) \\
\leq\ & R_i\left(\hat{\beta}_i(\hat{\Sigma}), \Sigma\right) \\
=\ & \lambda \beta_i^{\top} \left(\hat{\Sigma} + \lambda I_l\right)^{-1} \hat{\Sigma} \beta_i + \lambda^2 \beta_i^{\top} \left(\hat{\Sigma} + \lambda I_l\right)^{-1} \Sigma \left(\hat{\Sigma} + \lambda I_l\right)^{-1} \beta_i \\
& - \lambda^2 \beta_i^{\top} \left(\hat{\Sigma} + \lambda I_l\right)^{-2} \hat{\Sigma} \beta_i \\
=\ & R_i\left(\hat{\beta}_i(\hat{\Sigma}), \hat{\Sigma}\right) + \lambda^2 \beta_i^{\top} \left(\hat{\Sigma} + \lambda I_l\right)^{-1} \left(\Sigma - \hat{\Sigma}\right) \left(\hat{\Sigma} + \lambda I_l\right)^{-1} \beta_i \\
=\ & \min_{\beta \in \mathbb{R}^l} R_i(\beta, \hat{\Sigma}) + \lambda^2 \beta_i^{\top} \left(\hat{\Sigma} + \lambda I_l\right)^{-1} \left(\Sigma - \hat{\Sigma}\right) \left(\hat{\Sigma} + \lambda I_l\right)^{-1} \beta_i \\
\leq\ & \min_{\beta \in \mathbb{R}^l} R_i(\beta, \hat{\Sigma}) + \left(\frac{1}{2} + C\sqrt{\frac{l}{n}}\right) \lambda^2 \beta_i^{\top} \left(\hat{\Sigma} + \lambda I_l\right)^{-1} \hat{\Sigma} \left(\hat{\Sigma} + \lambda I_l\right)^{-1} \beta_i.
\end{aligned}
$$

Applying the Sherman-Morrison-Woodbury formula yields

$$
\begin{aligned}
\left(\hat{\Sigma} + \lambda I_l\right)^{-1} \hat{\Sigma}^{\frac{1}{2}} &= \lambda^{-1} \hat{\Sigma}^{\frac{1}{2}} - \lambda^{-2} \hat{\Sigma}^{\frac{1}{2}} \left(\lambda^{-1} \hat{\Sigma} + I_l\right)^{-1} \hat{\Sigma} \\
&= \lambda^{-1} \hat{\Sigma}^{\frac{1}{2}} - \lambda^{-1} \hat{\Sigma}^{\frac{1}{2}} \left(\lambda^{-1} \hat{\Sigma} + I_l\right)^{-1} \left(\lambda^{-1} \hat{\Sigma} + I_l - I_l\right) \\
&= \hat{\Sigma}^{\frac{1}{2}} \left(\hat{\Sigma} + \lambda I_l\right)^{-1}.
\end{aligned}
$$

Therefore, it holds

$$
\begin{aligned}
\min_{\beta \in \mathbb{R}^l} R_i(\beta, \Sigma) &\leq \min_{\beta \in \mathbb{R}^l} R_i(\beta, \hat{\Sigma}) + \left(\frac{1}{2} + C\sqrt{\frac{l}{n}}\right) \lambda^2 \beta_i^{\top} \left(\hat{\Sigma} + \lambda I_l\right)^{-1} \hat{\Sigma} \left(\hat{\Sigma} + \lambda I_l\right)^{-1} \beta_i \\
&= \min_{\beta \in \mathbb{R}^l} R_i(\beta, \hat{\Sigma}) + \left(\frac{1}{2} + C\sqrt{\frac{l}{n}}\right) \lambda^2 \beta_i^{\top} \hat{\Sigma}^{\frac{1}{2}} \left(\hat{\Sigma} + \lambda I_l\right)^{-2} \hat{\Sigma}^{\frac{1}{2}} \beta_i \\
&\leq \min_{\beta \in \mathbb{R}^l} R_i(\beta, \hat{\Sigma}) + \left(\frac{1}{2} + C\sqrt{\frac{l}{n}}\right) \lambda \beta_i^{\top} \hat{\Sigma}^{\frac{1}{2}} \left(\hat{\Sigma} + \lambda I_l\right)^{-1} \hat{\Sigma}^{\frac{1}{2}} \beta_i \\
&= \left(\frac{3}{2} + C\sqrt{\frac{l}{n}}\right) \min_{\beta \in \mathbb{R}^l} R_i(\beta, \hat{\Sigma}).
\end{aligned}
$$

Taking the average over $i = 1, \dots, m$ gives

$$
\frac{1}{m} \sum_{i=1}^{m} \min_{\beta \in \mathbb{R}^l} \|\beta_i - \beta\|_{\Sigma}^2 + \lambda \|\beta\|^2 \leq \left(\frac{3}{2} + C\sqrt{\frac{l}{n}}\right) \frac{1}{m} \sum_{i=1}^{m} \min_{\beta \in \mathbb{R}^l} \|\beta_i - \beta\|_{\hat{\Sigma}}^2 + \lambda \|\beta\|^2.
$$

In the same way as above discussion, we obtain

$$\frac{1}{m}\sum_{i=1}^{m}\min_{\beta\in\mathbb{R}^l}\|\beta_i-\beta\|_{\hat{\Sigma}}^2+\lambda\|\beta\|^2\leq\left(\frac{3}{2}+C\sqrt{\frac{l}{n}}\right)\frac{1}{m}\sum_{i=1}^{m}\min_{\beta\in\mathbb{R}^l}\|\beta_i-\beta\|_{\Sigma}^2+\lambda\|\beta\|^2.$$

Combining these two inequalities gives the result. $\qquad\square$

**Lemma 3.** *Let $x_1,\ldots,x_n$ be i.i.d. centered random vectors in $\mathbb{R}^d$ with covariance matrix $\Sigma$, $X=(x_1,\ldots,x_n)\in\mathbb{R}^{n\times m}$ and $\hat{\Sigma}=\frac{1}{n}X^\top X$. If $x_i$ is sub-Gaussian with $\|x_i\|_{\psi_2}\leq\sigma_x$, $\mu_k\left(\hat{\Sigma}_{k,k}\right)\geq\lambda$ and $\mu_1\left(\hat{\Sigma}_{-k,-k}\right)\leq\lambda$, then there exists constant $c$ which only depend on $\sigma_x$ such that it holds with probability at least $1-4e^{-cn}$,*

$$\lambda+\frac{1}{n}\mathrm{Tr}\left(\Sigma_{-k,-k}\right)\lesssim\lambda\lesssim\mu_k\left(\Sigma_{k,k}\right).$$

*Proof.* Since $\mu_1\left(\hat{\Sigma}_{-k,-k}\right)\lesssim\lambda$, from Lemma 11 there exists constant $c_1$ which only depend on $\sigma_x$, such that it holds with probability at least $1-2e^{-c_1n}$,

$$\begin{aligned}\lambda+\frac{1}{n}\mathrm{Tr}\left(\Sigma_{-k,-k}\right)&\leq&\lambda+\frac{1}{n}\left(1+\sigma_x^2\right)\mathrm{Tr}\left(\hat{\Sigma}_{-k,-k}\right)\\&\leq&\lambda+\frac{1}{n}(n-k)\left(1+\sigma_x^2\right)\lambda\\&\lesssim&\lambda.\end{aligned}$$

Since $\mu_k\left(\hat{\Sigma}_{k,k}\right)\geq\lambda$, by Lemma 12 there exists constants $c_2,C$ which only depend on $\sigma_x$, such that it holds that with probability at least $1-2e^{-c_2n}$,

$$\mu_k\left(\Sigma_{k,k}\right)\gtrsim\left(\frac{3}{2}+C\sqrt{\frac{k}{n}}\right)\mu_k\left(\Sigma_{k,k}\right)\geq\mu_k\left(\Sigma_{k,k}\right)\geq\lambda.$$

Combining these two inequalities, we obtain

$$\lambda+\frac{1}{n}\mathrm{Tr}\left(\Sigma_{-k,-k}\right)\lesssim\lambda\lesssim\mu_k\left(\Sigma_{k,k}\right).$$

$\qquad\square$

## B  PROOF OF THEOREM 2

*Proof.* From Lemma 4, for $W=\mathrm{diag}\left(\mathbf{w}\right)V^\top$ ($V$ is eigenvectors of $\Sigma_\beta$) such that $(\mathbf{w})_i=\frac{\sqrt{n\lambda}\sigma_i}{\sigma'}$, there exits a constant $c$ which only depends on $\sigma_x$ such that, with probability at least $1-2e^{-cn}$ it holds,

$$\begin{aligned}&\frac{1}{m}\sum_{i=1}^{m}\min_{\beta\in\mathbb{R}^l}\frac{1}{n}\left\|\mathbf{y}^{(i)}-XW^\top\beta\right\|^2+\lambda\|\beta\|^2+\frac{\sigma'^2}{n}\mathrm{Tr}\left(\hat{\Sigma}_{wx}\left(\hat{\Sigma}_{wx}+\lambda I_d\right)^{-1}\right)\\=&\frac{\lambda}{m}\sum_{i=1}^{m}\mathbf{y}^{(i)\top}\left(X_wX_w^\top+n\lambda I_n\right)^{-1}\mathbf{y}^{(i)}+\frac{\sigma'^2}{n}\mathrm{Tr}\left(\hat{\Sigma}_{wx}\left(\hat{\Sigma}_{wx}+\lambda I_d\right)^{-1}\right)\\=&\frac{\lambda}{m}\sum_{i=1}^{m}\beta_{*i}^\top X^\top\left(X_wX_w^\top+n\lambda I_n\right)^{-1}X\beta_{*i}+\frac{2\lambda}{m}\sum_{i=1}^{m}\epsilon^{(i)\top}\left(X_wX_w^\top+n\lambda I_n\right)^{-1}X\beta_{*i}\\&+\frac{\lambda}{m}\sum_{i=1}^{m}\epsilon^{(i)\top}\left(X_wX_w^\top+n\lambda I_n\right)^{-1}\epsilon^{(i)}+\frac{\sigma'^2}{n}\mathrm{Tr}\left(X_w\left(X_wX_w^\top+\lambda I_n\right)^{-1}X_w^\top\right)\\\leq&\frac{\lambda}{m}\sum_{i=1}^{m}\beta_{*i}^\top X^\top\left(X_wX_w^\top+n\lambda I_n\right)^{-1}X\beta_{*i}+\frac{2\lambda}{m}\sum_{i=1}^{m}\epsilon^{(i)\top}\left(X_wX_w^\top+n\lambda I_n\right)^{-1}X\beta_{*i}\end{aligned}$$

$$+\frac{1}{nm}\sum_{i=1}^{m}\epsilon^{(i)\top}\epsilon^{(i)}+\frac{\sigma'^2}{n}\mathrm{Tr}\left(X_w\left(X_wX_w^\top+\lambda I_n\right)^{-1}X_w^\top\right)$$

$$\lesssim\quad\frac{\lambda}{m}\sum_{i=1}^{m}\beta_{wi}^\top X_w^\top\left(X_wX_w^\top+n\lambda I_n\right)^{-1}X_w\beta_{wi}+\frac{\sigma'^2}{n}\mathrm{Tr}\left(X_w^\top\left(X_wX_w^\top+\lambda I_n\right)^{-1}X_w\right)+\sigma^2$$

$$=\quad\frac{2\sigma'^2}{n}\mathrm{Tr}\left(X_w^\top\left(X_wX_w^\top+n\lambda I_n\right)^{-1}X_w\right)+\sigma^2. \tag{B.1}$$

We write the eigenvectors of $W\Sigma_x W^\top$ corresponding to $\mu_i\left(W\Sigma_x W^\top\right)$ as $u_{\beta,i}$ and define $U_\beta :=$ $(u_{\beta,1},\ldots,u_{\beta,d})$ and $\Lambda_{x\beta} := \mathrm{diag}\left(\mu_1\left(W\Sigma_x W^\top\right),\ldots,\mu_d\left(W\Sigma_x W^\top\right)\right)$. Then, we can evaluate (B.1) as follows:

$$\mathrm{Tr}\left(X_w^\top\left(X_wX_w^\top+n\lambda I_n\right)^{-1}X_w\right)$$

$$=\quad\mathrm{Tr}\left((X_wU_\beta)^\top\left(X_wU_\beta\left(X_wU_\beta\right)^\top+n\lambda I_n\right)^{-1}X_wU_\beta\right)$$

$$\lesssim\quad\mathrm{Tr}\left((X_wU_\beta)_k^\top\left(X_wU_\beta\left(X_wU_\beta\right)^\top+n\lambda I_n\right)^{-1}(X_wU_\beta)_k\right)$$

$$+\mathrm{Tr}\left((X_wU_\beta)_{-k}^\top\left(X_wU_\beta\left(X_wU_\beta\right)^\top+n\lambda I_n\right)^{-1}(X_wU_\beta)_{-k}\right).$$

For the first term, by Lemma 5,

$$\mathrm{Tr}\left((X_wU_\beta)_k^\top\left(X_wU_\beta\left(X_wU_\beta\right)^\top+n\lambda I_n\right)^{-1}(X_wU_\beta)_k\right)$$

$$\leq\quad 2\mathrm{Tr}\left((X_wU_\beta)_k^\top\left((X_wU_\beta)_k(X_wU_\beta)_k^\top+n\lambda I_n\right)^{-1}(X_wU_\beta)_k\right).$$

For the second term, we have

$$\mathrm{Tr}\left((X_wU_\beta)_{-k}^\top\left(X_wU_\beta\left(X_wU_\beta\right)^\top+n\lambda I_n\right)^{-1}(X_wU_\beta)_{-k}\right)$$

$$\leq\quad\frac{1}{n\lambda}\mathrm{Tr}\left((X_wU_\beta)_{-k}^\top(X_wU_\beta)_{-k}\right).$$

From Lemma 11 it holds that with probability at least $1-2e^{-c_2 n}$,

$$\mathrm{Tr}\left((X_wU_\beta)_{-k}(X_wU_\beta)_{-k}^\top\right)\leq\left(1+\sigma_x^2\right)\frac{n\lambda}{\sigma'^2}\sum_{i=k+1}^{d}\mu_i\left(\Sigma_\beta^{\frac{1}{2}}\Sigma_x\Sigma_\beta^{\frac{1}{2}}\right).$$

Furthermore,

$$\mathrm{Tr}\left((X_wU_\beta)_k^\top\left((X_wU_\beta)_k(X_wU_\beta)_k^\top+n\lambda I_n\right)^{-1}(X_wU_\beta)_k\right)\quad\leq\quad k.$$

Combining these bounds yields

$$\frac{2\sigma'^2}{n}\mathrm{Tr}\left(X_w^\top\left(X_wX_w^\top+n\lambda I_n\right)^{-1}X_w\right)$$

$$\lesssim\quad\frac{4k\sigma'^2}{n}+2\left(1+\sigma_x^2\right)\sum_{i=k+1}^{d}\mu_i\left(\Sigma_\beta^{\frac{1}{2}}\Sigma_x\Sigma_\beta^{\frac{1}{2}}\right)$$

$$\lesssim\quad\frac{k\sigma'^2}{n}+\sum_{i=k+1}^{d}\mu_i\left(\Sigma_\beta^{\frac{1}{2}}\Sigma_x\Sigma_\beta^{\frac{1}{2}}\right). \tag{B.2}$$

Selecting $k$ in (B.2) such that $\mu_{k+1}\left(\Sigma_\beta^{\frac{1}{2}}\Sigma_x\Sigma_\beta^{\frac{1}{2}}\right)\leq\frac{\sigma'^2}{n}\leq\mu_k\left(\Sigma_\beta^{\frac{1}{2}}\Sigma_x\Sigma_\beta^{\frac{1}{2}}\right)$ gives the result. $\square$

In the later part of this section, we show lemmas required for Theorem 2.

**Lemma 4.** *Suppose $\epsilon^{(i)}$ are centered vectors whose components are independent and have sub-Gaussian norm at most $\sigma$. Then there exists a constant $c$ which only depends on $\sigma$ such that it holds with probability at least $1 - 2e^{-ct}$*

$$\frac{1}{nm} \sum_{i=1}^{m} \left\| \epsilon^{(i)} \right\|^2 \leq \left(1 + \frac{t}{n}\sigma^2\right) (\log 2)\sigma^2.$$

*Proof.* Let define $I$ as $I := \underset{i}{\operatorname{argmax}} \frac{1}{n} \left\| \epsilon^{(i)} \right\|^2$. From Lemma 11, there exists a constant $c$ which only depends on $\sigma$ such that it holds with probability at least $1 - 2e^{-ct}$

$$\frac{1}{nm} \sum_{i=1}^{m} \left\| \epsilon^{(i)} \right\|^2 \leq \frac{1}{n} \left\| \epsilon^{(I)} \right\|^2 \leq \left(1 + \frac{t}{n}\sigma^2\right) \operatorname{Var}\left[\epsilon^{(I)}\right] \leq \left(1 + \frac{t}{n}\sigma^2\right) (\log 2)\sigma^2.$$

$\square$

**Lemma 5.** *For any matrix $X \in \mathbb{R}^{n \times d}$, it holds that*
$$\operatorname{Tr}\left(X_k \left(XX^\top + n\lambda I_n\right) X_k^\top\right) \leq 2\operatorname{Tr}\left(X_k \left(X_k X_k^\top + n\lambda I_n\right) X_k^\top\right).$$

*Proof.* We define $\overline{A}_k(\lambda, X) := X_k X_k^\top + n\lambda I_n$. First we show the identity below:
$$X_k^\top \overline{A}_d(\lambda, X)^{-1} + X_k^\top \overline{A}_k(\lambda, X)^{-1} X_{-k} X_{-k}^\top \overline{A}_d(\lambda, X)^{-1} = X_k^\top \overline{A}_k(\lambda, X)^{-1}.$$
Indeed,
$$X_k^\top \overline{A}_d(\lambda, X)^{-1} + X_k^\top \overline{A}_k(\lambda, X)^{-1} X_{-k} X_{-k}^\top \overline{A}_d(\lambda, X)^{-1}$$
$$= X_k^\top \left(X_{-k} X_{-k}^\top + \overline{A}_k(\lambda, X)\right)^{-1} + X_k \top \overline{A}_k(\lambda, X)^{-1} X_{-k} X_{-k}^\top \left(X_{-k} X_{-k}^\top + \overline{A}_k(\lambda, X)\right)^{-1}$$
$$= X_k^\top \overline{A}_k(\lambda, X)^{-1} \left(X_{-k} X_{-k}^\top + \overline{A}_k(\lambda, X)\right) \left(X_{-k} X_{-k}^\top + \overline{A}_k(\lambda, X)\right)^{-1}$$
$$= X_k^\top \overline{A}_k(\lambda, X)^{-1}.$$
Thus, multiplying the identity by $X_k$ from the right, we obtain
$$\operatorname{Tr}\left(X_k^\top \overline{A}_k(\lambda, X)^{-1} X_k\right) = \operatorname{Tr}\left(X_k^\top \overline{A}_d(\lambda, X)^{-1} X_k\right)$$
$$+ \operatorname{Tr}\left(X_k^\top \overline{A}_k(\lambda, X)^{-1} X_{-k} X_{-k}^\top \overline{A}_d(\lambda, X)^{-1} X_k\right).$$
Since, $X_k X_k^\top \overline{A}_k(\lambda, X)^{-1}$ and $XX^\top \overline{A}_d(\lambda, X)^{-1}$ are positive semidefinite, we have
$$\operatorname{Tr}\left(X_k^\top \overline{A}_k(\lambda, X)^{-1} X_{-k} X_{-k}^\top \overline{A}_d(\lambda, X)^{-1} X_k\right)$$
$$= \operatorname{Tr}\left(X_k^\top \overline{A}_k(\lambda, X)^{-1} XX^\top \overline{A}_d(\lambda, X)^{-1} X_k\right)$$
$$- \operatorname{Tr}\left(X_k^\top \overline{A}_k(\lambda, X)^{-1} X_k X_k^\top \overline{A}_d(\lambda, X)^{-1} X_k\right)$$
$$\geq -\operatorname{Tr}\left(X_k^\top \overline{A}_k(\lambda, X)^{-1} X_k X_k^\top \overline{A}_d(\lambda, X)^{-1} X_k\right).$$
$X_k^\top \overline{A}_k(\lambda, X)^{-1} X_k$ and $X_k^\top \overline{A}_d(\lambda, X)^{-1} X_k$ are positive semidefinite. Therefore, it holds
$$-\operatorname{Tr}\left(X_k^\top \overline{A}_k(\lambda, X)^{-1} X_k X_k^\top \overline{A}_0(\lambda, X)^{-1} X_k\right)$$
$$\geq -\mu_1\left(X_k^\top \overline{A}_d(\lambda, X)^{-1} X_k\right) \operatorname{Tr}\left(X_k^\top \overline{A}_k(\lambda, X)^{-1} X_k\right).$$
Since $\overline{A}_0(\lambda, X)^{-\frac{1}{2}} X_{-k} X_{-k}^\top \overline{A}_0(\lambda, X)^{-\frac{1}{2}}$ is positive semidefinite, we obtain
$$1 \geq \mu_1\left(\overline{A}_d(\lambda, X)^{-\frac{1}{2}} XX^\top \overline{A}_d(\lambda, X)^{-\frac{1}{2}}\right)$$
$$= \mu_1\left(\overline{A}_d(\lambda, X)^{-\frac{1}{2}} X_k X_k^\top \overline{A}_d(\lambda, X)^{-\frac{1}{2}} + \overline{A}_d(\lambda, X)^{-\frac{1}{2}} X_{-k} X_{-k}^\top \overline{A}_d(\lambda, X)^{-\frac{1}{2}}\right)$$
$$\geq \mu_1\left(\overline{A}_d(\lambda, X)^{-\frac{1}{2}} X_k X_k^\top \overline{A}_d(\lambda, X)^{-\frac{1}{2}}\right)$$
$$= \mu_1\left(X_k^\top \overline{A}_d(\lambda, X)^{-1} X_k\right).$$
Combining these gives the result. $\square$

## C   PROOF OF PROPOSITION 1

*Proof.* From Bayes' theorem, we obtain

$$\mathbb{E}_{\beta_* \sim \mathcal{N}(0, \Sigma_\beta)} \left[ \mathbb{E}_{Y \sim N(X\beta_*, \sigma^2 I)} \left[ \|\beta_* - \hat{\beta}\|_{\Sigma_x}^2 \right] \right]$$

$$= \int \|\beta_* - \hat{\beta}\|_{\Sigma_x}^2 p(\beta_*) p(y|\beta_*, X) d\beta_* dy$$

$$= \int \|\beta_* - \hat{\beta}\|_{\Sigma_x}^2 p(\beta_*|X, y) p(y|X) d\beta_* dy.$$

Therefore, it holds

$$\operatorname*{argmin}_{\beta: \text{estimator}} \mathbb{E}_{\beta_* \sim \mathcal{N}(0, \Sigma_\beta)} \left[ \mathbb{E}_{Y \sim N(X\beta_*, \sigma^2 I)} \left[ \|\beta_* - \hat{\beta}\|_{\Sigma_x}^2 \right] \right]$$

$$= \operatorname*{argmin}_{\beta: \text{estimator}} \int \|\beta - \beta_*\|_{\Sigma_x}^2 \, p(\beta_*|X, y) d\beta_*$$

$$= \operatorname*{argmin}_{\beta: \text{estimator}} \int \left( \beta^\top \Sigma_x \beta - 2\beta^\top \Sigma_x \beta_* + \beta_*^\top \Sigma_x \beta_* \right) p(\beta_*|X, y) d\beta_*$$

$$= \operatorname*{argmin}_{\beta: \text{estimator}} \left( \beta - \int \beta_* p(\beta_*|X, y) d\beta_* \right)^\top \Sigma_x \left( \beta - \int \beta_* p(\beta_*|X, y) d\beta_* \right)$$

$$= \int \beta_* p(\beta_*|X, y) d\beta_*.$$

Since $\beta_* \sim \mathcal{N}(0, \Sigma_\beta)$, $\epsilon \sim \mathcal{N}(0, \sigma^2)$ by assumption, it holds

$$p(\beta_*|x, y)$$

$$\propto \quad \exp\left( -\frac{1}{2\sigma^2} \|y - X\beta_*\|^2 \right) \exp\left( -\frac{1}{2} \beta_*^\top \Sigma_\beta^{-1} \beta_* \right)$$

$$= \quad \exp\left( -\frac{1}{2\sigma^2} \left( \|y\|^2 - 2y^\top X\beta_* + \beta_*^\top \left( X^\top X + \sigma^2 \Sigma_\beta^{-1} \right) \beta_* \right) \right)$$

$$\propto \quad \exp\left( \frac{1}{2\tilde{\sigma}^2} \left\| \beta_* - \left( X^\top X + \sigma^2 \Sigma_\beta^{-1} \right)^{-1} X^\top y \right\|_{X^\top X + \sigma^2 \Sigma_\beta^{-1}}^2 \right).$$

In conclusion, we obtain

$$\int \beta_* p(\beta_*|x, y) d\beta_* = \left( X^\top X + \sigma^2 \Sigma_\beta^{-1} \right)^{-1} X^\top y.$$

$\square$

## D   PROOF OF THEOREM 3

*Proof.* For training data $(x_i, y_i)_{i=1}^n \in \mathbb{R}^d \times \mathbb{R}$, $X = (x_1, \ldots, x_n)^\top$ and $y(\beta_*, \epsilon) = (y_1, \ldots, y_n)^\top$ such that $y(\beta_*, \epsilon) = X\beta_* + \epsilon$, we define $\hat{\beta}(W, \beta_*, \epsilon) = W^\top \left( W X^\top X W^\top + n\lambda I_d \right)^{-1} W X^\top y(\beta_*, \epsilon)$. It holds that

$$\frac{1}{m} \sum_{i=1}^m \mathbb{E}_{x, \epsilon_i} \left[ \left( x^\top \beta_{*i} - x^\top W^\top \hat{\beta}_i(W) \right)^2 \right]$$

$$= \quad \mathbb{E}_{x \sim \mathcal{N}(0, \Sigma_x), \epsilon \sim \mathcal{N}(0, \tilde{\sigma}^2), \beta_* \sim \mathcal{N}(0, \Sigma_\beta)} \left[ \left( x^\top \beta_* - x^\top W^\top \hat{\beta}(W, \beta_*, \epsilon) \right)^2 \right]$$

$$\geq \quad \min_{\beta: \text{estimator}} \mathbb{E}_{x \sim \mathcal{N}(0, \Sigma_x), \epsilon \sim \mathcal{N}(0, \tilde{\sigma}^2), \beta_* \sim \mathcal{N}(0, \Sigma_\beta)} \left[ \left( x^\top \beta_* - x^\top \beta \right)^2 \right]$$

$$= \quad R_{\text{opt}}(X, \tilde{\sigma}). \tag{D.1}$$

From Proposition 1, for $W_B = \frac{\sqrt{n\lambda}}{\tilde{\sigma}} \Sigma_\beta^{\frac{1}{2}}$, we can write Bayes estimator as

$$
\begin{aligned}
\beta_{\text{Bayes}}(\beta_*, \epsilon) &= \left( X^\top X + \tilde{\sigma}^2 \Sigma_\beta^{-1} \right)^{-1} X^\top y(\beta_*, \epsilon) \\
&= W_B^\top \left( W_B X^\top X W_B^\top + n\lambda I_d \right)^{-1} W_B X^\top y(\beta_*, \epsilon).
\end{aligned}
$$

Therefore for $\Sigma_{x,B} = \frac{n\lambda}{\tilde{\sigma}^2} \Sigma_\beta^{\frac{1}{2}} \Sigma_x \Sigma_\beta^{\frac{1}{2}}$ and $\hat{\Sigma}_{x,B} = \frac{n\lambda}{\tilde{\sigma}^2} \Sigma_\beta^{\frac{1}{2}} \hat{\Sigma}_x \Sigma_\beta^{\frac{1}{2}}$, we can write the right hand side of (D.1) as

$$
\begin{aligned}
R_{\text{opt}}(X, \tilde{\sigma}) &= \mathbb{E}_{x \sim \mathcal{N}(0, \Sigma_x), \epsilon \sim \mathcal{N}(0, \tilde{\sigma}^2), \beta_* \sim \mathcal{N}(0, \Sigma_\beta)} \left[ \left( x^\top \beta_* - x^\top \beta_{\text{Bayes}}(\beta_*, \epsilon) \right)^2 \right] \\
&= B_{\text{Bayes}} + V_{\text{Bayes}},
\end{aligned}
$$

where,

$$
B_{\text{Bayes}} := \text{Tr} \left( \left( \left( n\hat{\Sigma}_{x,B} + n\lambda I_d \right)^{-1} n\hat{\Sigma}_{x,B} - I_d \right)^2 \Sigma_{x,B} \right),
$$

$$
V_{\text{Bayes}} = \mathbb{E}_\epsilon \left[ \left\| W_B X^\top \left( X W_B^\top W_B X^\top + n\lambda I_n \right)^{-1} \epsilon \right\|_{\Sigma_{x,B}}^2 \right].
$$

For $k$ in Assumption 3, we define $\rho_k$ such that $\frac{n\lambda}{\sigma^2} \mu_{k+1} \left( \Sigma_\beta^{\frac{1}{2}} \Sigma_x \Sigma_\beta^{\frac{1}{2}} \right) \rho_k = \left( \lambda + \frac{\lambda}{\sigma^2} \sum_{i=k+1}^d \mu_i \left( \Sigma_\beta^{\frac{1}{2}} \Sigma_x \Sigma_\beta^{\frac{1}{2}} \right) \right)$. For the variance term, from Assumption 2 and Lemma 7, there exists some absolute constant $c$ such that for any $t$ which satisfies $c < t < n$ and $k + 2\sigma^2 + \sqrt{kt}\sigma^2 < n/2$, it holds that with probability at least $1 - 20e^{-t/c}$,

$$
\begin{aligned}
V_{\text{Bayes}} &= \mathbb{E}_\epsilon \left[ \left\| W_B X^\top \left( X W_B^\top W_B X^\top + n\lambda I_n \right)^{-1} \epsilon \right\|_{\Sigma_{x,B}}^2 \right] \\
&\geq \frac{\tilde{\sigma}^2}{cn} \sum_{i=1}^d \min \left\{ 1, \frac{\frac{n^2\lambda^2}{\tilde{\sigma}^4} \mu_i \left( \Sigma_\beta^{\frac{1}{2}} \Sigma_x \Sigma_\beta^{\frac{1}{2}} \right)^2}{\sigma_x^4 \frac{n^2\lambda^2}{\tilde{\sigma}^4} \mu_{k+1} \left( \Sigma_\beta^{\frac{1}{2}} \Sigma_x \Sigma_\beta^{\frac{1}{2}} \right)^2 (\rho_k + 2)^2} \right\} \\
&\gtrsim \frac{\tilde{\sigma}^2}{cn} \sum_{i=1}^d \min \left\{ 1, \frac{\frac{n^2\lambda^2}{\tilde{\sigma}^4} \mu_i \left( \Sigma_\beta^{\frac{1}{2}} \Sigma_x \Sigma_\beta^{\frac{1}{2}} \right)^2}{\sigma_x^4 \frac{n^2\lambda^2}{\tilde{\sigma}^4} \mu_{k+1} \left( \Sigma_\beta^{\frac{1}{2}} \Sigma_x \Sigma_\beta^{\frac{1}{2}} \right)^2 \rho_k^2} \right\} \\
&= \frac{\tilde{\sigma}^2}{cn} \sum_{i=1}^d \min \left\{ 1, \frac{\mu_i \left( \Sigma_\beta^{\frac{1}{2}} \Sigma_x \Sigma_\beta^{\frac{1}{2}} \right)^2}{\sigma_x^4 \mu_{k+1} \left( \Sigma_\beta^{\frac{1}{2}} \Sigma_x \Sigma_\beta^{\frac{1}{2}} \right)^2 \rho_k^2} \right\} \\
&\geq \frac{\tilde{\sigma}^2}{cn} \sum_{i=1}^d \frac{\mu_i \left( \Sigma_\beta^{\frac{1}{2}} \Sigma_x \Sigma_\beta^{\frac{1}{2}} \right)^2}{\mu_i \left( \Sigma_\beta^{\frac{1}{2}} \Sigma_x \Sigma_\beta^{\frac{1}{2}} \right)^2 + \sigma_x^4 \mu_{k+1} \left( \Sigma_\beta^{\frac{1}{2}} \Sigma_x \Sigma_\beta^{\frac{1}{2}} \right)^2 \rho_k^2} \\
&\gtrsim \frac{\tilde{\sigma}^2}{cn} \sum_{i=1}^d \frac{\mu_i \left( \Sigma_\beta^{\frac{1}{2}} \Sigma_x \Sigma_\beta^{\frac{1}{2}} \right)^2}{\left( \mu_i \left( \Sigma_\beta^{\frac{1}{2}} \Sigma_x \Sigma_\beta^{\frac{1}{2}} \right) + \sigma_x^2 \mu_{k+1} \left( \Sigma_\beta^{\frac{1}{2}} \Sigma_x \Sigma_\beta^{\frac{1}{2}} \right) \rho_k \right)^2} \\
&\gtrsim \frac{\tilde{\sigma}^2}{cn} \sum_{i=1}^d \frac{\mu_i \left( \Sigma_\beta^{\frac{1}{2}} \Sigma_x \Sigma_\beta^{\frac{1}{2}} \right)^2}{\left( \mu_i \left( \Sigma_\beta^{\frac{1}{2}} \Sigma_x \Sigma_\beta^{\frac{1}{2}} \right) + \mu_{k+1} \left( \Sigma_\beta^{\frac{1}{2}} \Sigma_x \Sigma_\beta^{\frac{1}{2}} \right) \rho_k \right)^2}.
\end{aligned}
$$

In the last inequality, we use Assumption 2. For the bias term, we define eigenvectors $u_{B,1}, \ldots, u_{B,d}$ corresponding to $\mu_1 \left( W_B \Sigma_x W_B^\top \right), \ldots, \mu_1 \left( W_B \Sigma_x W_B^\top \right)$ and $U_B = (u_{B,1}, \ldots, u_{B,d})$. First, for any $j > K$, we need to evaluate the smallest eigenvalue of $A_{-j} \left( X W_B^\top U_B, \lambda \right) =$

$(XW_B^\top U_B)_{j-1}(XW_B^\top U_B)_{j-1}^\top + (XW_B^\top U_B)_{-j}(XW_B^\top U_B)_{-j}^\top + n\lambda I_n$ in order to apply Proposition 6. From Assumption 3, we have

$$n\lambda + \frac{n\lambda}{\tilde{\sigma}^2} \sum_{i=k+1}^{d} \mu_i \left( \Sigma_\beta^{\frac{1}{2}} \Sigma_x \Sigma_\beta^{\frac{1}{2}} \right) \leq 2n\lambda \leq 2\mu_n \left( A_{-j} \left( XW_B^\top U_B, \lambda \right) \right).$$

Therefore, since $W_B^{-\top} \Sigma_\beta W_B^{-1} = \frac{\tilde{\sigma}^2}{n\lambda} I_d$, from Lemma 6 there exists some absolute constant $c'$ such that for any $t$ which satisfies $t < \frac{n}{2\sigma_x^2}$, it holds that with probability at least $1 - 4e^{-t/c'}$,

$$
\begin{aligned}
B_{\text{Bayes}} &:= \text{Tr} \left( \left( \left( n\hat{\Sigma}_{x,B} + n\lambda I_d \right)^{-1} n\hat{\Sigma}_{x,B} - I_d \right)^2 \Sigma_{x,B} \right) \\
&\geq \frac{1}{2} \sum_{i=1}^{d} \frac{\mu_i \left( \Sigma_\beta^{\frac{1}{2}} \Sigma_x \Sigma_\beta^{\frac{1}{2}} \right)}{\left( 1 + \frac{\mu_i \left( \Sigma_\beta^{\frac{1}{2}} \Sigma_x \Sigma_\beta^{\frac{1}{2}} \right)}{2\mu_{k+1} \left( \Sigma_\beta^{\frac{1}{2}} \Sigma_x \Sigma_\beta^{\frac{1}{2}} \right) \rho_k} \right)^2} \\
&\gtrsim \sum_{i=1}^{d} \frac{\left( \mu_{k+1} \left( \Sigma_\beta^{\frac{1}{2}} \Sigma_x \Sigma_\beta^{\frac{1}{2}} \right) \rho_k \right)^2 \mu_i \left( \Sigma_\beta^{\frac{1}{2}} \Sigma_x \Sigma_\beta^{\frac{1}{2}} \right)}{\left( \mu_{k+1} \left( \Sigma_\beta^{\frac{1}{2}} \Sigma_x \Sigma_\beta^{\frac{1}{2}} \right) \rho_k + \mu_i \left( \Sigma_\beta^{\frac{1}{2}} \Sigma_x \Sigma_\beta^{\frac{1}{2}} \right) \right)^2} \\
&\gtrsim \sum_{i=1}^{d} \frac{\tilde{\sigma}^2 \left( \mu_{k+1} \left( \Sigma_\beta^{\frac{1}{2}} \Sigma_x \Sigma_\beta^{\frac{1}{2}} \right) \rho_k \right)^2 \mu_i \left( \Sigma_\beta^{\frac{1}{2}} \Sigma_x \Sigma_\beta^{\frac{1}{2}} \right)}{\left( \mu_{k+1} \left( \Sigma_\beta^{\frac{1}{2}} \Sigma_x \Sigma_\beta^{\frac{1}{2}} \right) \rho_k + \mu_i \left( \Sigma_\beta^{\frac{1}{2}} \Sigma_x \Sigma_\beta^{\frac{1}{2}} \right) \right)^2}.
\end{aligned}
$$

In the last inequality, we use Assumption 2. Combining these two bounds yields

$$
\begin{aligned}
&B_{\text{Bayes}} + V_{\text{Bayes}} \\
&\gtrsim \sum_{i=1}^{d} \frac{\mu_i \left( \Sigma_\beta^{\frac{1}{2}} \Sigma_x \Sigma_\beta^{\frac{1}{2}} \right) \tilde{\sigma}^2}{n} \frac{n \left( \mu_{k+1} \left( \Sigma_\beta^{\frac{1}{2}} \Sigma_x \Sigma_\beta^{\frac{1}{2}} \right) \rho_k \right)^2 + \mu_i \left( \Sigma_\beta^{\frac{1}{2}} \Sigma_x \Sigma_\beta^{\frac{1}{2}} \right)}{\left( \mu_{k+1} \left( \Sigma_\beta^{\frac{1}{2}} \Sigma_x \Sigma_\beta^{\frac{1}{2}} \right) \rho_k + \mu_i \left( \Sigma_\beta^{\frac{1}{2}} \Sigma_x \Sigma_\beta^{\frac{1}{2}} \right) \right)^2} \\
&= \sum_{i=1}^{d} \mu_i \left( \Sigma_\beta^{\frac{1}{2}} \Sigma_x \Sigma_\beta^{\frac{1}{2}} \right) \tilde{\sigma}^2 \frac{\left( n\mu_{k+1} \left( \Sigma_\beta^{\frac{1}{2}} \Sigma_x \Sigma_\beta^{\frac{1}{2}} \right) \rho_k \right)^2 + n\mu_i \left( \Sigma_\beta^{\frac{1}{2}} \Sigma_x \Sigma_\beta^{\frac{1}{2}} \right)}{\left( n\mu_{k+1} \left( \Sigma_\beta^{\frac{1}{2}} \Sigma_x \Sigma_\beta^{\frac{1}{2}} \right) \rho_k + n\mu_i \left( \Sigma_\beta^{\frac{1}{2}} \Sigma_x \Sigma_\beta^{\frac{1}{2}} \right) \right)^2}.
\end{aligned}
\tag{D.2}
$$

By assumption 2 and 3, $n\mu_{k+1} \left( \Sigma_\beta^{\frac{1}{2}} \Sigma_x \Sigma_\beta^{\frac{1}{2}} \right) \rho_k = \tilde{\sigma}^2 + \sum_{i=k+1}^{d} \mu_i \left( \Sigma_\beta^{\frac{1}{2}} \Sigma_x \Sigma_\beta^{\frac{1}{2}} \right) \leq 2\tilde{\sigma}^2$, and (D.2) is a decreasing function as long as $n\mu_{k+1} \left( \Sigma_\beta^{\frac{1}{2}} \Sigma_x \Sigma_\beta^{\frac{1}{2}} \right) \rho_k \leq 2\tilde{\sigma}^2 \leq 1$. Therefore, we have

$$
\begin{aligned}
B_{Bayes} + V_{Bayes} &\gtrsim \sum_{i=1}^{d} \mu_i \left( \Sigma_\beta^{\frac{1}{2}} \Sigma_x \Sigma_\beta^{\frac{1}{2}} \right) \tilde{\sigma}^2 \frac{\left( n\mu_{k+1} \left( \Sigma_\beta^{\frac{1}{2}} \Sigma_x \Sigma_\beta^{\frac{1}{2}} \right) \rho_k \right)^2 + n\mu_i \left( \Sigma_\beta^{\frac{1}{2}} \Sigma_x \Sigma_\beta^{\frac{1}{2}} \right)}{\left( n\mu_{k+1} \left( \Sigma_\beta^{\frac{1}{2}} \Sigma_x \Sigma_\beta^{\frac{1}{2}} \right) \rho_k + n\mu_i \left( \Sigma_\beta^{\frac{1}{2}} \Sigma_x \Sigma_\beta^{\frac{1}{2}} \right) \right)^2} \\
&\geq \sum_{i=1}^{d} \mu_i \left( \Sigma_\beta^{\frac{1}{2}} \Sigma_x \Sigma_\beta^{\frac{1}{2}} \right) \tilde{\sigma}^2 \frac{2\tilde{\sigma}^2 + n\mu_i \left( \Sigma_\beta^{\frac{1}{2}} \Sigma_x \Sigma_\beta^{\frac{1}{2}} \right)}{\left( 2\tilde{\sigma}^2 + n\mu_i \left( \Sigma_\beta^{\frac{1}{2}} \Sigma_x \Sigma_\beta^{\frac{1}{2}} \right) \right)^2} \\
&= \sum_{i=1}^{d} \frac{\mu_i \left( \Sigma_\beta^{\frac{1}{2}} \Sigma_x \Sigma_\beta^{\frac{1}{2}} \right) \tilde{\sigma}^2}{2\tilde{\sigma}^2 + n\mu_i \left( \Sigma_\beta^{\frac{1}{2}} \Sigma_x \Sigma_\beta^{\frac{1}{2}} \right)} \\
&\gtrsim \sum_{i=1}^{d} \min \left\{ \frac{\tilde{\sigma}^2}{n}, \mu_i \left( \Sigma_\beta^{\frac{1}{2}} \Sigma_x \Sigma_\beta^{\frac{1}{2}} \right) \right\}.
\end{aligned}
$$

$\square$

In the later part of the section we show lemmas needed to prove Theorem 3.

**Lemma 6** (Lower bound of bias term). *Suppose that $\Sigma_x$ and $\Sigma_\beta$ have the same eigenvectors, i.e., $U = V$ and that it is known for some $k, \delta, L$ that for any $j > k$ with probability at least $1 - \delta$, $\mu_n (A_{-j} (X, \lambda)) \geq \frac{1}{L} \left( n\lambda + \sum_{i>k} \lambda_i \right)$. Then for some absolute constant $c$ for any non-negative $t < \frac{n}{2\sigma_x^2}$ it holds that with probability at least $1 - 2\delta - 4e^{-t/c}$,*

$$
\begin{aligned}
\mathcal{B} \; := \; & \mathrm{Tr}\left( \Sigma_\beta^{\frac{1}{2}} \left( \left(X^\top X + n\lambda I_d\right)^{-1} X^\top X - I_d \right) \Sigma_x \left( \left(X^\top X + n\lambda I_d\right)^{-1} X^\top X - I_d \right) \Sigma_\beta^{\frac{1}{2}} \right) \\
\geq \; & \frac{1}{2} \sum_{i=1}^{d} \frac{\lambda_i \sigma_i^2}{\left(1 + \frac{\lambda_i}{2L\lambda_{k+1}\rho_k}\right)^2}.
\end{aligned}
$$

*Proof.* Let $X_u := XU$, $\Lambda := \mathbb{E}\left[ U^\top x x^\top U \right] = U^\top \Sigma_x U = \mathrm{diag}\left(\lambda_1, \ldots, \lambda_d\right)$, $\Sigma = U^\top \Sigma_\beta U = \mathrm{diag}\left(\sigma_1^2, \ldots, \sigma_d^2\right)$. In the same way as Lemma 15 in (Tsigler and Bartlett, 2020) we can write the bias term as

$$
\begin{aligned}
\mathcal{B} \; = \; & \mathrm{Tr}\left( \Sigma_\beta^{\frac{1}{2}} \left( I_d - X^\top \left(XX^\top + n\lambda I_n\right)^{-1} X \right) \Sigma_x \left( I_d - X^\top \left(XX^\top + n\lambda I_n\right)^{-1} X \right) \Sigma_\beta^{\frac{1}{2}} \right) \\
= \; & \mathrm{Tr}\left( \Sigma^{\frac{1}{2}} \left( I_d - X_u^\top \left(X_u X_u^\top + n\lambda I_n\right)^{-1} X_u \right) \Lambda \left( I_d - X_u^\top \left(X_u X_u^\top + n\lambda I_n\right)^{-1} X_u \right) \Sigma^{\frac{1}{2}} \right). \\
= \; & \sum_{i=1}^{d} \left( \left( I_d - X_u^\top \left(X_u X_u^\top + n\lambda I_n\right)^{-1} X_u \right) \Lambda \left( I_d - X_u^\top \left(X_u X_u^\top + n\lambda I_n\right)^{-1} X_u \right) \right)_{i,i} \sigma_i^2.
\end{aligned}
$$

$\Lambda$ is a diagonal matrix and so the same discussion as Lemma 15 in (Tsigler and Bartlett, 2020) gives the result. $\qquad\square$

**Lemma 7** (Lower bound of variance term). *Suppose that components the rows of $XU$ are independent and $\mathrm{Var}\left[\epsilon^{(i)}\right] = \sigma^2$. Then for some absolute constant $c$, for any $t, k$ such that $t > c$ and $k + 2\sigma_x^2 + \sqrt{kt}\sigma_x^2 < n/2$ it holds that with probability at least $1 - 20e^{-t/c}$,*

$$
\mathcal{V} := \frac{1}{m} \sum_{i=1}^{m} \mathbb{E}_{\epsilon^{(i)}} \left[ \left\| X^\top \left(XX^\top + \lambda I_n\right)^{-1} \epsilon^{(i)} \right\|_{\Sigma_x}^2 \right] \geq \frac{\sigma^2}{cn} \sum_{i=1}^{d} \min\left\{ 1, \frac{\lambda_i^2}{\sigma_x^4 \lambda_{k+1}^2 (\rho_k + 2)^2} \right\}.
$$

*Proof.* The variance term can be written as

$$
\begin{aligned}
\mathcal{V} \; = \; & \frac{1}{m} \sum_{i=1}^{m} \mathbb{E}_{\epsilon^{(i)}} \left[ \left\| X^\top \left(XX^\top + \lambda I_n\right)^{-1} \epsilon^{(i)} \right\|_{\Sigma_x}^2 \right] \\
= \; & \mathbb{E}_{\epsilon^{(1)}} \left[ \left\| X^\top \left(XX^\top + \lambda I_n\right)^{-1} \epsilon^{(1)} \right\|_{\Sigma_x}^2 \right] \\
= \; & \mathbb{E}_{\epsilon^{(1)}} \left[ \left\| (XU)^\top \left((XU)(XU)^\top + \lambda I_n\right)^{-1} \epsilon^{(1)} \right\|_{U^\top \Sigma_x U}^2 \right].
\end{aligned}
$$

Since $\mathbb{E}\left[ U^\top x x^\top U \right] = U^\top \Sigma_x U = \mathrm{diag}\left(\lambda_1, \ldots, \lambda_d\right)$, the same discussion as Lemma 14 in (Tsigler and Bartlett, 2020) gives the result. $\qquad\square$

# E  EVALUATION OF BIAS AND VARIANCE OF RIDGE REGRESSION IN MULTIVARIATE LINEAR REGRESSION

In this section, for $\lambda \in \mathbb{R}$ and $\eta \in \mathbb{R}^n$, we write $\hat{\beta}(\lambda, \eta) := X^\top \left(X^\top + n\lambda I_n\right)^{-1} \eta$.

**Proposition 2.** *Suppose the condition number of $A_k (X, \lambda)$ is at most $L$, then there exists (large) constants $c_x$, which only depends on $\sigma_x$ such that for all $t$ which satisfies*

1. $t \in (1, n/c_x)$,

2. $\sqrt{k} + \sqrt{t} \leq \sqrt{n}/c_x$,

*if $k$ and $\lambda$ satisfies $\frac{1}{\mu_1(\Sigma_{-k,-k})}\left(\lambda + \frac{1}{n}\mathrm{Tr}\left(\Sigma_{-k,-k}\right)\right) = \Omega\left(\frac{1}{L}\right)$, then it holds that with probability at least $1 - 20e^{-t/c_x}$,*

$$\frac{1}{m}\sum_{i=1}^{m}\mathbb{E}_x\left[\left(x^\top\beta_{*i} - x^\top\hat{\beta}(\lambda, y_i)\right)^2\right] \lesssim B + V,$$

$$\frac{B}{L^3\max\left\{L, \frac{\mathrm{Tr}(\Sigma_{-k,-k})}{n\lambda+\mathrm{Tr}(\Sigma_{-k,-k})}\right\}} = \mathrm{Tr}\left((\Sigma_x)_{k,k}^{-1}(\Sigma_\beta)_{-k,-k}\right)\left(\lambda + \frac{1}{n}\mathrm{Tr}\left((\Sigma_x)_{-k,-k}\right)\right)^2$$
$$+ \mathrm{Tr}\left((\Sigma_x)_{-k,-k}(\Sigma_\beta)_{-k,-k}\right),$$

$$\frac{V}{t\sigma^2 L^2} = \frac{k}{n} + \frac{1}{n}\frac{\mathrm{Tr}\left((\Sigma_x)^2_{-k,-k}\right)}{\left(\lambda + \frac{1}{n}\mathrm{Tr}\left((\Sigma_x)_{-k,-k}\right)\right)^2}.$$

*Alternatively, if $\lambda > 0$ and it is also known that for some $\delta < 1 - 4^{-n/c_x^2}$ with probability at least $1 - \delta$, the condition number of $A_k(X, \lambda)$ is at most $L$, then it holds that with probability at least $1 - 20e^{-t/c_x} - \delta$,*

$$\frac{B}{L^4} = \mathrm{Tr}\left((\Sigma_x)_{k,k}^{-1}(\Sigma_\beta)_{-k,-k}\right)\left(\lambda + \frac{1}{n}\mathrm{Tr}\left((\Sigma_x)_{-k,-k}\right)\right)^2 + \mathrm{Tr}\left((\Sigma_x)_{-k,-k}(\Sigma_\beta)_{-k,-k}\right),$$

$$\frac{V}{t\sigma^2 L^2} = \frac{k}{n} + \frac{1}{n}\frac{\mathrm{Tr}\left((\Sigma_x)^2_{-k,-k}\right)}{\left(\lambda + \frac{1}{n}\mathrm{Tr}\left((\Sigma_x)_{-k,-k}\right)\right)^2},$$

$$\frac{n\lambda + \mathrm{Tr}\left((\Sigma_x)_{-k,-k}\right)}{n\mu_1\left((\Sigma_x)_{-k,-k}\right)} \geq \frac{1}{c_x L}.$$

*Proof.*

$$\frac{1}{m}\sum_{i=1}^{m}\mathbb{E}_x\left[\left(x^\top\beta_{*i} - x^\top X^\top\left(XX^\top + n\lambda I_n\right)^{-1}y_i\right)^2\right]$$
$$\leq \frac{2}{m}\sum_{i=1}^{m}\left\|\beta_{*i} - \hat{\beta}(\lambda, X\beta_{*i})\right\|_{\Sigma_x}^2 + \frac{2}{m}\sum_{i=1}^{m}\left\|X^\top\left(XX^\top + n\lambda I_n\right)^{-1}\epsilon^{(i)}\right\|_{\Sigma_x}^2.$$

The first term corresponds to the bias and the second term corresponds to that of the variance. From Lemma 8 and Lemma 9, for all $t > 1$ it holds that with probability at least $1 - 4e^{-c_1 t}$,

$$\frac{1}{m}\sum_{i=1}^{m}\left\|\beta_{*i} - \hat{\beta}(\lambda, y_i)\right\|_{\Sigma_x}^2$$

$$\lesssim \frac{\mu_1\left(A_k(X, \lambda)^{-1}\right)^2}{\mu_n\left(A_k(X, \lambda)^{-1}\right)^2}\frac{\mu_1\left((\Sigma_x)_{k,k}^{-1/2}X_k^\top X_k(\Sigma_x)_{k,k}^{-1/2}\right)}{\mu_k\left((\Sigma_x)_{k,k}^{-1/2}X_k^\top X_k(\Sigma_x)_{k,k}^{-1/2}\right)^2}\left(\frac{1}{m}\sum_{i=1}^{m}\|X_{-k}(\beta_{*i})_{-k}\|^2\right)$$

$$+ \frac{\mathrm{Tr}\left((\Sigma_x)_{k,k}^{-1}(\Sigma_\beta)_{k,k}\right)}{\mu_n\left(A_k(X, \lambda)^{-1}\right)^2\mu_k\left((\Sigma_x)_{k,k}^{-1/2}X_k^\top X_k(\Sigma_x)_{k,k}^{-1/2}\right)^2}$$

$$+ \left\|X_{-k}(\Sigma_x)_{-k,-k}X_{-k}^\top\right\|\left(\mu_1\left(A_1(X, \lambda)^{-1}\right)^2\left(\frac{1}{m}\sum_{i=1}^{m}\|X_{-k}(\beta_{*i})_{-k}\|^2\right)\right.$$

$$+ \frac{\mu_1\left(A_k(X, \lambda)^{-1}\right)^2}{\mu_n\left(A_k(X, \lambda)^{-1}\right)^2}\frac{\mu_1\left((\Sigma_x)_{k,k}^{-1/2}X_k^\top X_k(\Sigma_x)_{k,k}^{-1/2}\right)}{\mu_k\left((\Sigma_x)_{k,k}^{-1/2}X_k^\top X_k(\Sigma_x)_{k,k}^{-1/2}\right)^2}\mathrm{Tr}\left((\Sigma_x)_{k,k}^{-1}(\Sigma_\beta)_{k,k}\right)\right)$$

$$+\mathrm{Tr}\left((\Sigma_x)_{-k,-k}(\Sigma_\beta)_{-k,-k}\right)+C_1\sigma^2 t\frac{\mu_1\left(A_k\left(X,\lambda\right)^{-1}\right)^2}{\mu_n\left(A_k\left(X,\lambda\right)^{-1}\right)^2}\frac{\mathrm{Tr}\left(X_k(\Sigma_x)_k^{-1}X_k^\top\right)}{\mu_k\left((\Sigma_x)_{k,k}^{-\frac{1}{2}}X_k^\top X_k(\Sigma_x)_{k,k}^{-\frac{1}{2}}\right)^2}$$

$$+C_1\sigma^2 t\mu_1\left(A_k\left(X,\lambda\right)^{-1}\right)^2\mathrm{Tr}\left(X_{-k}(\Sigma_x)_{-k,-k}X_{-k}^\top\right).$$

The matrix $X_k(\Sigma_x)_{k,k}^{-1/2}\in\mathbb{R}^{k\times n}$ has $n$ i.i.d. columns with isotropic sub-gaussian distribution in $\mathbb{R}^k$. By theorem 5.39 in (Vershynin, 2012) there exist constants $c_x', C_x'$ (which only depends on $\sigma_x$) such that for every $t>0$ s.t. $\sqrt{n}-C_x'\sqrt{k}-\sqrt{t}>0$, it holds that with probability at least $1-2e^{-c_x't}$,

$$\mu_k\left((\Sigma_x)_{k,k}^{-1/2}X_k^\top X_k(\Sigma_x)_{k,k}^{-1/2}\right)\geq\left(\sqrt{n}-C_x'\sqrt{k}-\sqrt{t}\right)^2,$$

$$\mu_1\left((\Sigma_x)_{k,k}^{-1/2}X_k^\top X_k(\Sigma_x)_{k,k}^{-1/2}\right)\leq\left(\sqrt{n}+C_x'\sqrt{k}+\sqrt{t}\right)^2.$$

$\mathrm{Tr}\left(X_k(\Sigma_x)_{k,k}^{-1}X_k^\top\right)$ is the sum of $l_2$ norms of $n$ i.i.d. isotropic vectors in $\mathbb{R}^k$, and $\mathrm{Tr}\left(X_{-k}(\Sigma_x)_{-k,-k}X_{-k}^\top\right)$ is the sum of squared norms of $n$ i.i.d. sub-gaussian vectors with covariance $(\Sigma_x)_{-k,-k}^2$. By Lemma 11, it holds that with probability at least $1-4e^{-c_2 t}$,

$$\mathrm{Tr}\left(X_k(\Sigma_x)_{k,k}^{-1}X_k^\top\right)\leq\left(n+t\sigma_x^2\right)k,$$

$$\mathrm{Tr}\left(X_{-k}(\Sigma_x)_{-k,-k}^2 X_{-k}^\top\right)\leq\left(n+t\sigma_x^2\right)\mathrm{Tr}\left((\Sigma_x)_{-k,-k}^2\right).$$

By Lemma 20 in (Tsigler and Bartlett, 2020) it holds that with probability at least $1-6e^{-t/c_3}$,

$$\left\|X_{-k}(\Sigma_x)_{-k,-k}X_{-k}^\top\right\|\leq c_3\sigma_x^2\left((t+n)\mu_1\left((\Sigma_x)_{-k,-k}\right)^2+\mathrm{Tr}\left((\Sigma_x)_{-k,-k}^2\right)\right).$$

$\frac{1}{m}\sum_{i=1}^m\|X_{-k}(\beta_{*i})_{-k}\|^2=\mathrm{Tr}\left(X_{-k}(\Sigma_\beta)_{-k,-k}X_{-k}^\top\right)$ is the sum of $l_2$ norms of $n$ i.i.d. sub-gaussian vectors with covariance $(\Sigma_x)_{-k}^{1/2}(\Sigma_\beta)_{-k,-k}(\Sigma_x)_{-k}^{1/2}$. We can apply Lemma 11 to get that for any $t\in(0,n)$ it holds that with probability at least $1-4e^{-c_2 t}$,

$$\frac{1}{m}\sum_{i=1}^m\|X_{-k}(\beta_{*i})_{-k}\|^2=\left(n+t\sigma_x^2\right)\mathrm{Tr}\left((\Sigma_x)_{-k,-k}(\Sigma_\beta)_{-k,-k}\right).$$

Combining the above bounds gives that for some constant $C_x'$ which only depends on $\sigma_x$, it holds that

$$\begin{aligned}
\frac{B}{C_x'}\leq{}&\mathrm{Tr}\left((\Sigma_x)_{-k,-k}(\Sigma_\beta)_{-k,-k}\right)\left(1+\frac{\mu_1\left(A_k(X,\lambda)^{-1}\right)^2}{\mu_n\left(A_k(X,\lambda)^{-1}\right)^2}\right)\\
&+\mathrm{Tr}\left((\Sigma_x)_{-k,-k}(\Sigma_\beta)_{-k}\right)\mu_1\left(A_k\left(X,\lambda\right)^{-1}\right)^2 n^2\mu_1\left((\Sigma_x)_{-k,-k}\right)^2\\
&+\mathrm{Tr}\left((\Sigma_x)_{-k,-k}(\Sigma_\beta)_{-k}\right)\mu_1\left(A_k\left(X,\lambda\right)^{-1}\right)^2 n\mathrm{Tr}\left((\Sigma_x)_{-k,-k}^2\right)\\
&+\frac{\mathrm{Tr}\left((\Sigma_x)_{k,k}^{-1}(\Sigma_\beta)_{k,k}\right)}{n^2\mu_n\left(A_k(X,\lambda)^{-1}\right)^2}\\
&+\mathrm{Tr}\left((\Sigma_x)_{k,k}^{-1}(\Sigma_\beta)_{k,k}\right)\frac{\mu_1\left(A_k(X,\lambda)^{-1}\right)^2}{\mu_n\left(A_k(X,\lambda)^{-1}\right)^2}\mu_1\left((\Sigma_x)_{-k,-k}\right)^2\\
&+\mathrm{Tr}\left((\Sigma_x)_{k,k}^{-1}(\Sigma_\beta)_{k,k}\right)\frac{\mu_1\left(A_k(X,\lambda)^{-1}\right)^2}{\mu_n\left(A_k(X,\lambda)^{-1}\right)^2}\frac{1}{n}\mathrm{Tr}\left((\Sigma_x)_{-k,-k}^2\right), \quad\quad\text{(E.1)}\\
\frac{V}{C_x'\sigma^2 t}\leq{}&\frac{\mu_1\left(A_k(X,\lambda)^{-1}\right)^2}{\mu_n\left(A_k(X,\lambda)^{-1}\right)^2}\frac{k}{n}+n\mu_1\left(A_k(X,\lambda)^{-1}\right)^2\mathrm{Tr}\left((\Sigma_x)_{-k,-k}^2\right). \quad\quad\text{(E.2)}
\end{aligned}$$

Note that by Lemma 21 in (Tsigler and Bartlett, 2020) there exists a constant $c_4$ such that with probability at least $1 - \delta - 2e^{-c_4 t}$,

$$\frac{n - t\sigma_x^2}{nL}\left(n\lambda + \mathrm{Tr}\left((\Sigma_x)_{-k,-k}\right)\right) \leq \mu_k(A_k(X, \lambda)^{-1})$$

$$\leq \mu_1(A_k(X,\lambda)^{-1}) \leq \frac{\left(n + t\sigma_x^2\right)L}{n}\left(n\lambda + \mathrm{Tr}\left((\Sigma_x)_{-k,-k}\right)\right).$$

Plugging that into (E.1) and (E.2) gives

$$\frac{B}{C_x'L^2} \leq \mathrm{Tr}\left((\Sigma_x)_{-k,-k}(\Sigma_\beta)_{-k,-k}\right)\left(1 + \frac{n^2\mu_1\left((\Sigma_x)_{-k,-k}\right)^2 + n\mathrm{Tr}\left((\Sigma_x)^2_{-k,-k}\right)}{\left(n\lambda + \mathrm{Tr}\left((\Sigma_x)_{-k,-k}\right)\right)^2}\right)$$

$$+ \mathrm{Tr}\left((\Sigma_x)^{-1}_{k,k}(\Sigma_\beta)_{k,k}\right)\left(\frac{n\lambda + \mathrm{Tr}\left((\Sigma_x)_{-k,-k}\right)}{n}\right)^2$$

$$+ \mathrm{Tr}\left((\Sigma_x)^{-1}_{k,k}(\Sigma_\beta)_{k,k}\right)\left(\mu_1\left((\Sigma_x)_{-k,-k}\right)^2 + \frac{1}{n}\mathrm{Tr}\left((\Sigma_x)^2_{-k,-k}\right)\right), \tag{E.3}$$

$$\frac{V}{C_x''\sigma^2 tL^2} \leq \frac{k}{n} + n\frac{\mathrm{Tr}\left((\Sigma_x)^2_{-k,-k}\right)}{\left(n\lambda + \mathrm{Tr}\left((\Sigma_x)_{-k,-k}\right)\right)^2}.$$

For the first assertion, we have $\frac{1}{\mu_1(\Sigma_{-k,-k})}\left(\lambda + \frac{1}{n}\mathrm{Tr}\left(\Sigma_{-k,-k}\right)\right) = \Omega\left(\frac{1}{L}\right)$ by assumption. Therefore, it holds

$$\mathrm{Tr}\left((\Sigma_x)^2_{-k,-k}\right) \leq \mu_1\left(\Sigma_{-k,-k}\right)\mathrm{Tr}\left((\Sigma_x)_{-k,-k}\right) \lesssim \frac{L}{n}\left(n\lambda + \mathrm{Tr}\left(\Sigma_{-k,-k}\right)\right)\mathrm{Tr}\left(\Sigma_{-k,-k}\right).$$

Plugging this inequality and $\frac{1}{\mu_1(\Sigma_{-k,-k})}\left(\lambda + \frac{1}{n}\mathrm{Tr}\left(\Sigma_{-k,-k}\right)\right) = \Omega\left(\frac{1}{L}\right)$ into (E.3), it holds that

$$\frac{B}{C_x'L^2} \lesssim \mathrm{Tr}\left((\Sigma_x)_{-k,-k}(\Sigma_\beta)_{-k,-k}\right)\left(1 + L^2 + \frac{L\mathrm{Tr}\left(\Sigma_{-k,-k}\right)}{n\lambda + \mathrm{Tr}\left((\Sigma_x)_{-k,-k}\right)}\right)$$

$$+ \mathrm{Tr}\left((\Sigma_x)^{-1}_{k,k}(\Sigma_\beta)_{k,k}\right)\left(\frac{n\lambda + \mathrm{Tr}\left((\Sigma_x)_{-k,-k}\right)}{n}\right)^2$$

$$+ \mathrm{Tr}\left((\Sigma_x)^{-1}_{k,k}(\Sigma_\beta)_{k,k}\right)\left(L^2\left(\lambda + \frac{1}{n}\mathrm{Tr}\left(\Sigma_{-k,-k}\right)\right)^2\right)$$

$$+ \frac{L}{n^2}\left(n\lambda + \mathrm{Tr}\left(\Sigma_{-k,-k}\right)\right)\mathrm{Tr}\left(\Sigma_{-k,-k}\right)\right),$$

and, this gives the first assertion.

For the second assertion, when $\lambda \geq 0$, recall that Lemma 21 in (Tsigler and Bartlett, 2020) also says that $\delta < 1 - 4e^{-c_4 s}$ implies

$$\frac{n\lambda + \mathrm{Tr}\left((\Sigma_x)_{-k,-k}\right)}{n\mu_1\left((\Sigma_x)_{-k,-k}\right)} \geq \frac{1}{L} \cdot \frac{1 - s\sigma_x^2/n}{1 + s\sigma_x^2/n}.$$

Moreover, in the same way as the first assertion, it holds

$$\mathrm{Tr}\left((\Sigma_x)^2_{-k,-k}\right) \leq \mu_1\left((\Sigma_x)_{-k,-k}\right)\mathrm{Tr}\left((\Sigma_x)_{-k,-k}\right) \leq \frac{L}{n}\left(n\lambda + \mathrm{Tr}\left((\Sigma_x)_{-k,-k}\right)\right)^2\frac{1 + s\sigma_x^2/n}{1 - s\sigma_x^2/n}.$$

Plugging it into (E.3) gives the upper bound of the bias in the second assertion. In addition, when $\lambda \geq 0$, it holds that

$$\frac{V}{C_x''\sigma^2 tL^2} \leq \frac{k}{n} + n\frac{\mathrm{Tr}\left((\Sigma_x)^2_{-k,-k}\right)}{\left(n\lambda + \mathrm{Tr}\left((\Sigma_x)_{-k,-k}\right)\right)^2} \leq \frac{k}{n} + n\frac{\mathrm{Tr}\left((\Sigma_x)_{-k,-k}\right)}{n\lambda + \mathrm{Tr}\left((\Sigma_x)_{-k,-k}\right)},$$

and this gives the second assertion. $\square$

**Lemma 8** (Bias term). *In the same setting as Proposition 2, it holds*

$$\frac{1}{m}\sum_{i=1}^{m}\left\|\beta_{*i}-\hat{\beta}(\lambda,X\beta_{*i})\right\|_{\Sigma_x}^2$$

$$\lesssim \frac{\mu_1\left(A_k\left(X,\lambda\right)^{-1}\right)^2}{\mu_n\left(A_k\left(X,\lambda\right)^{-1}\right)^2}\frac{\mu_1\left((\Sigma_x)_{k,k}^{-1/2}X_k^\top X_k(\Sigma_x)_{k,k}^{-1/2}\right)}{\mu_k\left((\Sigma_x)_{k,k}^{-1/2}X_k^\top X_k(\Sigma_x)_{k,k}^{-1/2}\right)^2}\left(\frac{1}{m}\sum_{i=1}^{m}\|X_{-k}(\beta_{*i})_{-k}\|^2\right)$$

$$+\frac{\mathrm{Tr}\left((\Sigma_x)_{k,k}^{-1}(\Sigma_\beta)_{k,k}\right)}{\mu_n\left(A_k\left(X,\lambda\right)^{-1}\right)^2\mu_k\left((\Sigma_x)_{k,k}^{-1/2}X_k^\top X_k(\Sigma_x)_{k,k}^{-1/2}\right)^2}$$

$$+\left\|X_{-k}(\Sigma_x)_{-k,-k}X_{-k}^\top\right\|\left(\mu_1\left(A_1\left(X,\lambda\right)^{-1}\right)^2\left(\frac{1}{m}\sum_{i=1}^{m}\|X_{-k}(\beta_{*i})_{-k}\|^2\right)\right.$$

$$+\frac{\mu_1\left(A_k\left(X,\lambda\right)^{-1}\right)^2}{\mu_n\left(A_k\left(X,\lambda\right)^{-1}\right)^2}\frac{\mu_1\left((\Sigma_x)_{k,k}^{-1/2}X_k^\top X_k(\Sigma_x)_{k,k}^{-1/2}\right)}{\mu_k\left((\Sigma_x)_{k,k}^{-1/2}X_k^\top X_k(\Sigma_x)_{k,k}^{-1/2}\right)^2}\mathrm{Tr}\left((\Sigma_x)_{k,k}^{-1}(\Sigma_\beta)_{k,k}\right)\right)$$

$$+\mathrm{Tr}\left((\Sigma_x)_{-k,-k}(\Sigma_\beta)_{-k,-k}\right).$$

*Proof.*

$$\frac{1}{m}\sum_{i=1}^{m}\left\|\beta_{*i}-\hat{\beta}(\lambda,X\beta_{*i})\right\|_{\Sigma_x}^2$$

$$= \frac{1}{m}\sum_{i=1}^{m}\mathbb{E}_x\left[\left(x^\top\beta_{*i}-x^\top X^\top\left(XX^\top+n\lambda I_n\right)X\beta_{*i}\right)^2\right]$$

$$= \frac{1}{m}\sum_{i=1}^{m}\mathbb{E}_x\left[\left(x_k^\top(\beta_{*i})_k+x_{-k}^\top(\beta_{*i})_{-k}-x_k^\top X_k^\top\left(XX^\top+n\lambda I_n\right)X\beta_{*i}\right.\right.$$

$$\left.\left.-x_{-k}^\top X_{-k}^\top\left(XX^\top+n\lambda I_n\right)X\beta_{*i}\right)^2\right]$$

$$\lesssim \frac{1}{m}\sum_{i=1}^{m}\left\|(\beta_{*i})_k-X_k^\top\left(XX^\top+n\lambda I_n\right)X\beta_{*i}\right\|_{(\Sigma_x)_k}^2$$

$$+\frac{1}{m}\sum_{i=1}^{m}\left\|X_{-k}^\top\left(XX^\top+n\lambda I_n\right)X\beta_{*i}\right\|_{(\Sigma_x)_{-k}}^2+\frac{1}{m}\sum_{i=1}^{m}\left\|(\beta_{*i})_{-k}\right\|_{(\Sigma_x)_{-k}}^2.$$

By the same discussion of chapter F.1 and F.2 in (Tsigler and Bartlett, 2020), we obtain the following bound:

$$\frac{1}{m}\sum_{i=1}^{m}\left\|\beta_{*i}-\hat{\beta}(\lambda,X\beta_{*i})\right\|_{\Sigma_x}^2$$

$$\lesssim \frac{\mu_1\left(A_k\left(X,\lambda\right)^{-1}\right)^2}{\mu_n\left(A_k\left(X,\lambda\right)^{-1}\right)^2}\frac{\mu_1\left((\Sigma_x)_{k,k}^{-1/2}X_k^\top X_k(\Sigma_x)_{k,k}^{-1/2}\right)}{\mu_k\left((\Sigma_x)_{k,k}^{-1/2}X_k^\top X_k(\Sigma_x)_{k,k}^{-1/2}\right)^2}\left(\frac{1}{m}\sum_{i=1}^{m}\|X_{-k}(\beta_{*i})_{-k}\|^2\right)$$

$$+\frac{\mathrm{Tr}\left((\Sigma_x)_{k,k}^{-1}(\Sigma_\beta)_{k,k}\right)}{\mu_n\left(A_k\left(X,\lambda\right)^{-1}\right)^2\mu_k\left((\Sigma_x)_{k,k}^{-1/2}X_k^\top X_k(\Sigma_x)_{k,k}^{-1/2}\right)^2}$$

$$+\left\|X_{-k}(\Sigma_x)_{-k,-k}X_{-k}^\top\right\|\left(\mu_1\left(A_1\left(X,\lambda\right)^{-1}\right)^2\left(\frac{1}{m}\sum_{i=1}^{m}\|X_{-k}(\beta_{*i})_{-k}\|^2\right)\right.$$

$$+\frac{\mu_1\left(A_k\left(X,\lambda\right)^{-1}\right)^2}{\mu_n\left(A_k\left(X,\lambda\right)^{-1}\right)^2}\frac{\mu_1\left((\Sigma_x)_{k,k}^{-1/2}X_k^\top X_k(\Sigma_x)_{k,k}^{-1/2}\right)}{\mu_k\left((\Sigma_x)_{k,k}^{-1/2}X_k^\top X_k(\Sigma_x)_{k,k}^{-1/2}\right)^2}\mathrm{Tr}\left((\Sigma_x)_{k,k}^{-1}(\Sigma_\beta)_{k,k}\right)\Bigg)$$

$$+\mathrm{Tr}\left((\Sigma_x)_{-k,-k}\left(\Sigma_\beta\right)_{-k,-k}\right).$$

$\square$

**Lemma 9** (Variance term). *In the same setting as Proposition 2, for all $t > 1$ there exist constants $c, C$ such that with probability at least $1 - 4e^{-ct}$,*

$$\frac{1}{m}\sum_{i=1}^m\left\|X^\top\left(XX^\top+n\lambda I_n\right)^{-1}\epsilon^{(i)}\right\|$$

$$\leq\quad C\sigma^2 t\frac{\mu_1\left(A_k\left(X,\lambda\right)^{-1}\right)^2}{\mu_n\left(A_k\left(X,\lambda\right)^{-1}\right)^2}\frac{\mathrm{Tr}\left(X_k(\Sigma_x)_{k,k}^{-1}X_k^\top\right)}{\mu_k\left((\Sigma_x)_{k,k}^{-\frac{1}{2}}X_k^\top X_k(\Sigma_x)_{k,k}^{-\frac{1}{2}}\right)^2}$$

$$+C\sigma^2 t\mu_1\left(A_k\left(X,\lambda\right)^{-1}\right)^2\mathrm{Tr}\left(X_{-k}(\Sigma_x)_{-k,-k}X_{-k}^\top\right).$$

*Proof.* From Lemma 12 in (Tsigler and Bartlett, 2020), we obtain

$$V\quad=\quad\frac{1}{m}\sum_{i=1}^m\left\|X^\top\left(XX^\top+n\lambda I_n\right)^{-1}\epsilon^{(i)}\right\|_{\Sigma_x}^2$$

$$\leq\quad\frac{1}{m}\sum_{i=1}^m\frac{\epsilon^{(i)^\top}A_k\left(X,\lambda\right)^{-1}X_k(\Sigma_x)_{k,k}^{-1}X_k^\top A_k\left(X,\lambda\right)^{-1}\epsilon^{(i)}}{\mu_n\left(A_k\left(X,\lambda\right)^{-1}\right)^2\mu_k\left((\Sigma_x)_{k,k}^{-\frac{1}{2}}X_k^\top X_k(\Sigma_x)_{k,k}^{-\frac{1}{2}}\right)^2}$$

$$+\frac{1}{m}\sum_{i=1}^m\epsilon^{(i)^\top}A_0\left(X,\lambda\right)^{-1}X_{-k}\Sigma_{-k}X_{-k}^\top A_0\left(X,\lambda\right)^{-1}\epsilon^{(i)}.\qquad\text{(E.4)}$$

From Lemma 13, for all $t > 1$ it holds that with probability at least $1 - 4e^{-ct}$,

$$\frac{1}{m}\sum_{i=1}^m\epsilon^{(i)^\top}A_k\left(X,\lambda\right)^{-1}X_k(\Sigma_x)_{k,k}^{-1}X_k^\top A_k\left(X,\lambda\right)^{-1}\epsilon^{(i)}$$

$$\leq\quad C\sigma^2 t\mathrm{Tr}\left(A_k\left(X,\lambda\right)^{-1}X_k(\Sigma_x)_{k,k}^{-1}X_k^\top A_k\left(X,\lambda\right)^{-1}\right)$$

$$\leq\quad C\sigma^2 t\mu_1\left(A_k\left(X,\lambda\right)^{-1}\right)^2\mathrm{Tr}\left(X_k(\Sigma_x)_{k,k}^{-1}X_k^\top\right),$$

$$\frac{1}{m}\sum_{i=1}^m\epsilon^{(i)^\top}A_0\left(X,\lambda\right)^{-1}X_{-k}(\Sigma_x)_{-k,-k}^{-1}X_{-k}^\top A_0\left(X,\lambda\right)^{-1}\epsilon^{(i)}$$

$$\leq\quad C\sigma^2 t\mathrm{Tr}\left(A_0\left(X,\lambda\right)^{-1}X_{-k}(\Sigma_x)_{-k,-k}X_{-k}^\top A_0\left(X,\lambda\right)^{-1}\right)$$

$$\leq\quad C\sigma^2 t\mu_1\left(A_k\left(X,\lambda\right)^{-1}\right)^2\mathrm{Tr}\left(X_{-k}(\Sigma_x)_{-k,-k}X_{-k}^\top\right),$$

where we used $\mu_1\left(A_k\left(X,\lambda\right)^{-1}\right)\geq\mu_1\left(A_0\left(X,\lambda\right)^{-1}\right)$. Combining these two inequalities and (E.4) gives the result. $\square$

For $W\in\mathbb{R}^{d\times d}$ which corresponds to eigenvalues of $\Sigma_x$, we can obtain Corollary 1 by applying Proposition 2 with $X = XU$.

# F    EQUIVALENCE OF UPPER BOND AND LOWER BOUND

**Proposition 3** (Tsigler and Bartlett (2020)). *Let,*

$$\underline{B} := \sum_{i=1}^{d} \frac{\lambda_i \sigma_i^2}{\left(1 + \frac{\lambda_i}{\lambda_{k+1} \rho_k}\right)^2}, \qquad \overline{B} := \sum_{i=1}^{k} \lambda_i \tilde{\sigma}_i^2 \frac{\rho_k^2 \lambda_{k+1}^2}{\lambda_i^2} + \sum_{i=k+1}^{d} \lambda_i \tilde{\sigma}_i^2,$$

$$\underline{V} := \frac{1}{n} \sum_{i=1}^{d} \min \left\{ 1, \frac{\lambda_i^2}{\sigma_x^4 \lambda_{k+1}^2 (\rho_k + 2)^2} \right\}, \quad \overline{V} = \frac{k}{n} + \frac{1}{n} \sum_{i=k+1}^{d} \frac{\lambda_i^2}{\rho_k^2 \lambda_{k+1}^2}.$$

*Suppose $\rho_k \in (a, b)$ for some $b > a > 0$. Then, it holds that*

$$1 \le \frac{\overline{B}}{\underline{B}} \le \max \left\{ (1+b)^2, \left(1+a^{-1}\right)^2 \right\}, \quad 1 \le \frac{\overline{V}}{\underline{V}} \le \max \left\{ (2+b)^2, \left(1+2a^{-1}\right)^2 \right\}.$$

*Alternatively, if $k = \min\{l : \rho_l > b\}$ and $b > 1/n$, then, it holds that*

$$1 \le \frac{\overline{B}}{\underline{B}} \le \max \left\{ (1+b)^2, \left(1+b^{-1}\right)^2 \right\}, \quad 1 \le \frac{\overline{V}}{\underline{V}} \le \max \left\{ (2+b)^2, \left(1+2b^{-1}\right)^2 \right\}.$$

# G    EQUIVALENCE OF PROPOSED MODEL AND NORMAL RIDGE REGRESSION

**Proposition 4.** *Let* $\tilde{B}_{\text{Norm}} = \sum_{i=1}^{k} \lambda_i \tilde{\sigma}_i^2 \frac{\rho_k^2 \lambda_{k+1}^2}{\lambda_i^2} + \sum_{i=k+1}^{d} \lambda_i \tilde{\sigma}_i^2, \tilde{V}_{\text{Norm}} = \frac{k\sigma^2}{n} +$ $\frac{\sigma^2}{n} \sum_{i=k+1}^{d} \frac{\lambda_i^2}{\rho_k^2 \lambda_{k+1}^2}$. *When for some constant $\alpha > 0$ we can write $\Sigma_\beta = \alpha I_d$, if there exists $k < n$ such that $\mu_{k+1} \le \frac{\sigma^2}{n} \le \lambda_k$, then for some $\lambda \in \mathbb{R}$, it holds that*

$$\tilde{B}_{\text{Norm}} + \tilde{V}_{\text{Norm}} \approx \sum_{i=1}^{d} \min \left\{ \frac{\sigma^2}{n}, \lambda_i \right\} = U_{\text{NN}}.$$

*Proof.* When it holds $\tilde{\sigma}_i^2 = \alpha$, if we choose $k$ such that $\lambda_{k+1} \le \frac{\sigma^2}{n} \le \lambda_k$ and $\lambda \in \mathbb{R}$ such that $\rho_k = \Theta(1)$, then we can evaluate $\tilde{B}_{\text{Norm}}$ and $\tilde{V}_{\text{Norm}}$ as

$$\tilde{B}_{\text{Norm}} = \sum_{i=1}^{k} \lambda_i \tilde{\sigma}_i^2 \frac{\rho_k^2 \lambda_{k+1}^2}{\lambda_i^2} + \sum_{i=k+1}^{d} \lambda_i \tilde{\sigma}_i^2$$

$$\approx \sum_{i=1}^{k} \frac{\lambda_k}{\lambda_i} \frac{\sigma^2}{n} + \sum_{i=k+1}^{d} \lambda_i,$$

$$\tilde{V}_{\text{Norm}} = \frac{k\sigma^2}{n} + \frac{\sigma^2}{n} \sum_{i=k+1}^{d} \frac{\lambda_i^2}{\rho_k^2 \lambda_{k+1}^2}$$

$$\approx \sum_{i=1}^{k} \frac{\sigma^2}{n} + \sum_{i=k+1}^{d} \frac{\lambda_i}{\lambda_k} \lambda_i.$$

Therefore, since $\lambda_{k+1} \le \frac{\sigma^2}{n} \le \lambda_k$ it holds

$$\tilde{B}_{\text{Norm}} + \tilde{V}_{\text{Norm}} \approx \frac{\sigma^2}{n} \sum_{i=1}^{k} \max \left\{ 1, \frac{\lambda_k}{\lambda_i} \right\} + \sum_{i=k+1}^{d} \lambda_i \max \left\{ 1, \frac{\lambda_i}{\lambda_k} \right\}$$

$$= \frac{\sigma^2}{n} \sum_{i=1}^{k} + \sum_{i=k+1}^{d} \lambda_i$$

$$= \min \left\{ \frac{\sigma^2}{n}, \lambda_i \right\}.$$

$\square$

## H   EXAMPLES

### H.1   PROOF OF EXAMPLE 1

*Proof.* In the vanilla ridge regression, for any $k < n$, the bias term can be lower bounded as follows:

$$
\begin{aligned}
B_{\text{Norm}} &\gtrsim \sum_{i=1}^{k} \lambda_i \sigma_i^2 \frac{\rho_k^2 \lambda_{k+1}^2}{\lambda_i^2} + \sum_{i=k+1}^{d} \lambda_i \sigma_i^2 \\
&> \lambda_1 \sigma_1^2 \frac{\rho_k^2 \lambda_{k+1}^2}{\lambda_1^2} \\
&= \rho_k^2 \frac{n}{e \log n}.
\end{aligned}
$$

Therefore, for $B_{\text{Norm}} = o(1)$ it is necessary that $\rho_k^{-2} = \omega\left(\frac{n}{\log n}\right)$. However, in that case, we can evaluate variance $V_{\text{Norm}}$ as

$$
\begin{aligned}
\frac{V_{\text{Norm}}}{L^2 t} &= \frac{k}{n}\sigma^2 + \frac{\sigma^2}{n} \sum_{i=k+1}^{d} \frac{\lambda_i^2}{\rho_k^2 \lambda_{k+1}^2} \\
&> \frac{\sigma^2}{n} \sum_{i=n}^{d} \frac{\lambda_i^2}{\rho_k^2 \lambda_{k+1}^2} \\
&\gtrsim \frac{n}{n \log n} \sum_{i=n}^{d} \frac{1}{i+1-n} \\
&= \frac{1}{\log n} \sum_{i=1}^{d+1-n} \frac{1}{i} \\
&> \frac{1}{\log n} \log(d+1-n) \\
&> \frac{\log(n+1)}{\log n} \\
&> 1.
\end{aligned}
$$

On the other hand, the obtained two-layer neural network achieves a predictive risk converging to zero:

$$
\begin{aligned}
U_{\text{NN}} &= \sum_{i=1}^{d} \min\left\{\lambda_i \sigma_i^2, \frac{\sigma'^2}{n}\right\} \\
&= \sum_{i=1}^{n} \min\left\{\lambda_i \sigma_i^2, \frac{\sigma'^2}{n}\right\} + \sum_{i=n+1}^{d} \min\left\{\lambda_i \sigma_i^2, \frac{\sigma'^2}{n}\right\} \\
&= \sum_{i=1}^{n} \min\left\{\frac{n}{e^i \log n}, \frac{1}{n}\right\} + \sum_{i=n+1}^{d} \min\left\{\frac{n}{(i+1-n)e^i \log n}, \frac{1}{n}\right\} \\
&< \sum_{i=1}^{n} \min\left\{\frac{n}{e^i}, \frac{1}{n}\right\} + \sum_{i=n+1}^{d} \min\left\{\frac{n}{e^i}, \frac{1}{n}\right\} \\
&= \sum_{i=1}^{\lfloor 2\log n \rfloor} \frac{1}{n} + \sum_{i=\lceil 2\log n \rceil}^{d} \frac{n}{e^i} \\
&\leq \frac{2\log n}{n} + \frac{n}{n^2} \frac{1 - e^{-d-1+\lceil 2\log n \rceil}}{1 - \frac{1}{e}} \\
&\lesssim \frac{\log n}{n}.
\end{aligned}
$$

$\square$

### H.2   PROOF OF EXAMPLE 2

*Proof.* In the vanilla ridge regression, for any $k < n$, the bias term can be lower bounded as follows:

$$
\begin{aligned}
\frac{B_{\mathrm{Norm}}}{L^4} &= \sum_{i=1}^{k} \lambda_i \sigma_i^2 \frac{\rho_k^2 \lambda_{k+1}^2}{\lambda_i^2} + \sum_{i=k+1}^{d} \lambda_i \sigma_i^2 \\
&> \lambda_d \sigma_d^2 \\
&= e^{d-d} \\
&= 1.
\end{aligned}
$$

Therefore, the bias term does not vanish as $n$ goes to $\infty$. On the other hand, the obtained two-layer neural network achieves a predictive risk converging to zero:

$$
\begin{aligned}
U_{\mathrm{NN}} &= \sum_{i=1}^{d} \min\left\{ \lambda_i \sigma_i^2, \frac{\sigma'^2}{n} \right\} \\
&= \sum_{i=1}^{d} \min\left\{ e^{i-d}, \frac{1}{n} \right\} \\
&= \frac{1}{e^d} \sum_{i=1}^{\lfloor d - \log n \rfloor} e^i + \sum_{i=\lceil d - \log n \rceil}^{d} \frac{1}{n} \\
&\leq \frac{1}{e^{d-1}(e-1)} \left( \frac{e^d}{n} - 1 \right) + \frac{\log n}{n} \\
&\lesssim \frac{1}{n} + \frac{\log n}{n} \\
&\lesssim \frac{\log n}{n}.
\end{aligned}
$$

$\square$

### H.3   PROOF OF EXAMPLE 3

*Proof.* In the vanilla ridge regression, it holds that

$$
\begin{aligned}
\frac{V_{\mathrm{Norm}}}{L^2 t} &= \frac{k}{n} \sigma^2 + \frac{\sigma^2}{n} \sum_{i=k+1}^{d} \frac{\lambda_i^2}{\rho_k^2 \lambda_{k+1}^2} \\
&> \frac{\sigma^2}{n \rho_k^2 \lambda_{k+1}^2} \sum_{i=n+1}^{d} \lambda_i^2 \\
&= \frac{\sigma^2(d-n)}{n^3 \rho_k^2 \lambda_{k+1}^2}.
\end{aligned}
$$

Therefore, $\rho_k^2 \lambda_{k+1}^2 = \omega(\frac{\sigma^2(d-n)}{n^3})$ is necessary for the variance term go to zero. However, in that situation, it holds that

$$
\begin{aligned}
\frac{B_{\mathrm{Norm}}}{L^4} &= \sum_{i=1}^{k} \lambda_i \sigma_i^2 \frac{\rho_k^2 \lambda_{k+1}^2}{\lambda_i^2} + \sum_{i=k+1}^{d} \lambda_i \sigma_i^2 \\
&= \rho_k^2 \lambda_{k+1}^2 \sum_{i=1}^{k} \frac{\sigma_i^2}{\lambda_i} + \sum_{i=k+1}^{d} \lambda_i \sigma_i^2 \\
&> \frac{\sigma^2(d-n)}{n^3} \frac{\sigma_1^2}{\lambda_1} \\
&= 1 - \frac{1}{n^2}
\end{aligned}
$$

$$\gtrsim \quad 1.$$

Therefore, the bias term doesn't vanish. On the contrary, in obtained two-layer neural network, it holds that

$$
\begin{aligned}
U_{\text{NN}} &= \sum_{i=1}^{d} \min\left\{\lambda_i \sigma_i^2, \frac{\sigma'^2}{n}\right\} \\
&= \sum_{i=1}^{n} \min\left\{i^{-3}, \frac{1}{n}\right\} + \sum_{i=n+1}^{d} \min\left\{n^{-1}i^{-2}, \frac{1}{n}\right\} \\
&= \sum_{i=1}^{\lfloor n^{\frac{1}{3}} \rfloor} \frac{1}{n} + \sum_{i=\lceil n^{\frac{1}{3}} \rceil}^{n} i^{-3} + \sum_{i=n+1}^{d} \frac{1}{ni^2} \\
&\leq \frac{n^{\frac{1}{3}}}{n} + \int_{\lfloor n^{-\frac{1}{3}} \rfloor}^{n} x^{-3} dx + \frac{1}{n} \int_{n}^{d} x^{-2} dx \\
&= n^{-\frac{2}{3}} + \frac{1}{2}\left(\lfloor n^{\frac{1}{3}} \rfloor\right)^2 - \frac{1}{2}n^{-2} + \frac{1}{n^2} - \frac{1}{nd} \\
&\lesssim \frac{3}{2n^{\frac{2}{3}}} - \frac{1}{nd} \\
&\lesssim n^{-\frac{2}{3}}.
\end{aligned}
$$

$\square$

## H.4    PROOF OF EXAMPLE 4

*Proof.* For the vanilla ridge regression, if $k < n^{\frac{2}{3}}$, it holds

$$
\begin{aligned}
\frac{B_{\text{Norm}}}{L^4} &= \sum_{i=1}^{k} \lambda_i \tilde{\sigma}_i^2 \frac{\rho_k^2 \lambda_{k+1}^2}{\lambda_i^2} + \sum_{i=k+1}^{d} \lambda_i \tilde{\sigma}_i^2 \\
&\geq \lambda_{n^{\frac{2}{3}}} \sigma_{n^{\frac{2}{3}}}^2 \\
&= 1.
\end{aligned}
$$

On the other hand, if $n^{\frac{2}{3}} \leq k$, it holds

$$
\begin{aligned}
\frac{B_{\text{Norm}}}{L^4} &= \sum_{i=1}^{k} \lambda_i \sigma_i^2 \frac{\rho_k^2 \lambda_{k+1}^2}{\lambda_i^2} + \sum_{i=k+1}^{d} \lambda_i \sigma_i^2 \\
&= \sum_{i=1}^{n^{\frac{2}{3}}} \lambda_i \sigma_i^2 \frac{\rho_k^2 \lambda_{k+1}^2}{\lambda_i^2} + \sum_{i=n^{\frac{2}{3}}+1}^{k} \lambda_i \sigma_i^2 \frac{\rho_k^2 \lambda_{k+1}^2}{\lambda_i^2} + \sum_{i=k+1}^{d} \lambda_i \sigma_i^2 \\
&> \sum_{i=1}^{n^{\frac{2}{3}}} \lambda_i \sigma_i^2 \frac{\rho_k^2 \lambda_{k+1}^2}{\lambda_i^2} \\
&= \rho_k^2 \lambda_{k+1}^2 \sum_{i=1}^{n^{\frac{2}{3}}} \frac{\sigma_i^2}{\lambda_i} \\
&= \rho_k^2 \lambda_{k+1}^2 n^{\frac{3}{2}} \sum_{i=1}^{n^{\frac{2}{3}}} i \\
&\geq \frac{\rho_k^2 \lambda_{k+1}^2}{2} n^{\frac{5}{2}}.
\end{aligned}
$$

$\rho_k^2 \lambda_{k+1}^2 = o\left(n^{-\frac{5}{2}}\right)$ is necessary for bias term to vanish. However, in this situation, it holds

$$
\begin{aligned}
\frac{V_{Norm}}{L^2 t} &= \frac{k}{n}\sigma^2 + \frac{\sigma^2}{n}\sum_{i=k+1}^{d}\frac{\lambda_i^2}{\rho_k^2\lambda_{k+1}^2} \\
&> n^{\frac{3}{2}}\sum_{i=n+1}^{d} i^{-2} \\
&> n^{\frac{3}{2}}\int_{n}^{d} x^{-2}dx \\
&= n^{\frac{3}{2}}\left(\frac{1}{n} - \frac{1}{d}\right).
\end{aligned}
$$

In the obtained two-layer neural network, it holds

$$
\begin{aligned}
U_{NN} &= \sum_{i=1}^{d}\min\left\{\lambda_i\sigma_i^2, \frac{\sigma'^2}{n}\right\} \\
&= \sum_{i=1}^{n^{\frac{2}{3}}}\min\left\{n^{\frac{3}{2}}i^{-1}, \frac{1}{n}\right\} + \sum_{i=n^{\frac{2}{3}}}^{d}\min\left\{i^{-2}, \frac{1}{n}\right\} \\
&< \frac{n^{\frac{2}{3}}}{n} + \int_{n^{\frac{2}{3}}}^{d} x^{-2}dx \\
&= n^{-\frac{1}{3}} + n^{-\frac{2}{3}} - \frac{1}{d} \\
&\lesssim \frac{1}{n^{1/3}} - \frac{1}{d} \\
&< n^{-\frac{1}{3}}.
\end{aligned}
$$

$\square$

## I   AUXILIARY LEMMAS

**Lemma 10.** *Suppose $Z \in \mathbb{R}^d$ is a matrix with independent isotropic sub-gaussian rows with $\|Z\|_{\psi_2} \leq \sigma$. Consider $\Sigma \in \mathbb{R}^{d \times d}$ is a positive semidefinite matrix. Then it holds that $\left\|(Z\Sigma)_{k,k}\Sigma_{k,k}^{-1}\right\|_{\psi_2} \leq \sigma$ and $\left\|(Z\Sigma)_{-k,-k}\Sigma_{-k,-k}^{-1}\right\|_{\psi_2} \leq \sigma$.*

*Proof.*

$$
\begin{aligned}
\sup_{\|s_k\|=1}\langle s_k, (Z\Sigma)_{k,k}\Sigma_{k,k}^{-1}\rangle &= \sup_{\|s_k\|=1}\left\langle \begin{pmatrix} s_k \\ 0 \end{pmatrix}, Z\Sigma\begin{pmatrix} \Sigma_{k,k}^{-1} & 0 \\ 0 & 0 \end{pmatrix}\right\rangle \\
&= \sup_{\|s_k\|=1}\left\langle \begin{pmatrix} s_k \\ 0 \end{pmatrix}, Z\begin{pmatrix} I_k & 0 \\ 0 & 0 \end{pmatrix}\right\rangle \\
&= \sup_{\|s_k\|=1}\left\langle \begin{pmatrix} s_k \\ 0 \end{pmatrix}, \begin{pmatrix} Z_k \\ 0 \end{pmatrix}\right\rangle \\
&= \sup_{\|s_k\|=1}\left\langle \begin{pmatrix} s_k \\ 0 \end{pmatrix}, Z\right\rangle \\
&\leq \sup_{\|s\|=1}\langle s, Z\rangle.
\end{aligned}
$$

Therefore, it holds $\left\|(Z\Sigma)_{k,k}\Sigma_{k,k}^{-1}\right\|_{\psi_2} \leq \sigma$. We can show the second assertion, in the same way as the first assertion. $\square$

**Lemma 11.** *Suppose $Z \in \mathbb{R}^{n \times d}$ is a matrix with independent isotropic sub-gaussian rows with $\|Z\|_{\psi_2} \leq \sigma$. Consider $\Sigma \in \mathbb{R}^{d \times d}$ is a positive semidefinite matrix. Then for some absolute constant $c$ for any $t \in (0, n)$ and $0 < k < d$, it holds that with probability at least $1 - 2e^{-ct}$,*

$$\left(n - t\sigma^2\right) \operatorname{Tr}\left(\Sigma_{-k,-k}\right) \leq \sum_{i=1}^{n} \left\|\left(\Sigma^{1/2} Z_i^\top\right)_{-k}\right\|^2 \leq \left(n + t\sigma^2\right) \operatorname{Tr}\left(\Sigma_{-k,-k}\right).$$

*Proof.* Let eigenvectors of $\Sigma_{-k,-k}$ as $U_{-k} \in \mathbb{R}^{d \times d}$. Then,

$$\left|\sum_{i=1}^{n} \left\|\left(\Sigma^{1/2} Z_i^\top\right)_{-k}\right\|^2 - \operatorname{Tr}\left(\Sigma_{-k,-k}\right)\right| = \left|\sum_{i=1}^{n} \left\|U_{-k}^\top \left(\Sigma^{1/2} Z_i^\top\right)_{-k}\right\|^2 - \sum_{i=1}^{d} \mu_i\left(\Sigma\right)\right|.$$

From Lemma 10, it holds that $\left\|(Z\Sigma)_{-k,-k} \Sigma_{-k,-k}^{-1}\right\|_{\psi_2} \leq \sigma$. Therefore, applying Lemma 17 in (Tsigler and Bartlett, 2020) gives the result. $\square$

**Lemma 12.** *Let $x_1, \ldots, x_n$ be i.i.d. centered random vectors in $\mathbb{R}^l$ with covariance matrix $\Sigma$. If $x$ is sub-Gaussian with $\|x\|_{\psi_2} \leq \sigma_x$, then there exist constants $C$ and $c$ which only depends on $\sigma_x$ such that it holds that with probability at least $1 - 2\exp(-ct^2)$,*

$$\left(1 - C\sqrt{\frac{l}{n}} - t\right) \Sigma \preccurlyeq \hat{\Sigma} \preccurlyeq \left(1 + C\sqrt{\frac{l}{n}} + t\right) \Sigma.$$

*Proof.* By Theorem 5.39 in (Vershynin, 2012), there exist $C'$ and $c'$ which only depends on $\sigma_x$ such that it holds that with probability at least $1 - 2e^{-c't^2}$,

$$\|I_l - \Sigma^{-\frac{1}{2}} \hat{\Sigma} \Sigma^{-\frac{\top}{2}}\|_{\text{op}} \leq 2C\sqrt{\frac{l}{n}} + 2t,$$

$$\therefore \quad -\left(2C'\sqrt{\frac{l}{n}} + 2t\right) I_l \preccurlyeq I_l - \Sigma^{-\frac{1}{2}} \hat{\Sigma} \Sigma^{-\frac{\top}{2}} \preccurlyeq \left(2C'\sqrt{\frac{l}{n}} + 2t\right) I_l,$$

$$\therefore \quad \left(1 - 2C'\sqrt{\frac{l}{n}} - 2t\right) \Sigma \preccurlyeq \hat{\Sigma} \preccurlyeq \left(2C'\sqrt{\frac{l}{n}} - 2t\right) \Sigma.$$

$\square$

**Lemma 13.** *Suppose $A \in \mathbb{R}^{n \times n}$ is a positive semidefinite matrix and $\epsilon^{(i)}$ $(i = 1, \ldots, m)$ are centered vectors whose components are independent and have sub-gaussian norm at most $\sigma$. Then for some constants $c, C$, for any $t > 1$ it holds that with probability at least $1 - 2e^{-t/c}$,*

$$\frac{1}{m} \sum_{i=1}^{m} \epsilon^{(i)\top} A \epsilon^{(i)} \leq C\sigma^2 t \operatorname{Tr}(A).$$

*Proof.* From Lemma 18 in Tsigler and Bartlett (2020) for any $i$ there exists constants $C_\epsilon, c_i$ it holds that with probability at least $1 - 2e^{-t/c_i}$,

$$\epsilon^{(i)\top} A \epsilon^{(i)} \leq C_\epsilon \sigma^2 t \operatorname{Tr}(A).$$

Notice that $C_\epsilon$ doesn't depend on the choice of $i$. Let $c := \min_i c_i$ and $J := \operatorname{argmax}_i \epsilon^{(i)\top} A \epsilon^{(i)}$, and we have

$$\Pr\left(\epsilon^{(J)\top} A \epsilon^{(J)} \leq C_\epsilon \sigma^2 t \operatorname{Tr}(A)\right) > 1 - 2e^{-t/c_J}$$
$$> 1 - 2e^{-t/c}.$$

Therefore it holds that with probability at least $1 - 2e^{-t/c}$,

$$\frac{1}{m} \sum_{i=1}^{m} \epsilon^{(i)\top} A \epsilon^{(i)} \leq \left(\epsilon^{(J)}\right)^\top A \epsilon^{(J)} \leq C_\epsilon \sigma^2 t \operatorname{Tr}(A).$$

$\square$