# OpenReview forum: "Optimal criterion for feature learning of two-layer linear neural network in high dimensional interpolation regime"
_ICLR.cc/2024/Conference — ICLR 2024 poster_

### Official Review · Reviewer_8UMM · 2023-10-17

**Soundness:** 3 good
**Presentation:** 2 fair
**Contribution:** 2 fair
**Rating:** 5
**Confidence:** 5

**Summary:**

The paper considers solving multi-output linear regression with two-layer linear networks. First, the authors construct $R(W)$ which serves as an (asymptotic) upper bound of the predictive risk. Next, they further bound $R(W)$ by a quantity $U_\text{NN}$ which depends on the eigenvalues of $\Sigma_\beta^\frac{1}{2} \Sigma_x \Sigma_\beta^\frac{1}{2}$. Then, it is argued that the Bayes risk can be lower bounded by the same expression. Under certain conditions, it is shown that the two-layer linear neworks obtained by minimizing $R(W)$ can outperform standard ridge regression. Results are supported by an experiment on synthetic data.

**Strengths:**

- Results are sound and fully justified.

- Implication of the main results are well explained.

- Comprehensive case study is provided.

**Weaknesses:**

- There are many additional assumptions stated inside the theorems: "for all $W$ which satisfies $R(W) - \sigma^2 \leq$..."(Theorem 1), "there exists $k\leq n$ such that .." (Theorem 2), "the rows of $X W_B^\top U_B$ are independent" (Theorem 3), "the condition number of $A_k$.." (Corollary 1). To me, the assumptions looked very artificial, and it was hard to see why it is reasonable to assume those. It would strengthen the paper if the authors could state all necessary assumptions separately before stating the theorems. Without this, it is hard to judge how non-trivial the results are.

- Although the authors have provided Example 1--4 to show the advantage of using $R(W)$ over ridge regression, the settings seem very artificial and restrictive. I wouldn't be too surprised even if there exists a pathological setting where standard ridge regression does not generalize well and there is a quick fix for it.

- This happens again in the experiment; the authors choose a very specific parameters and argue that the proposed method improves over ridge regression. I would be interested to see further justification for considering such setting.


**Minor points**

- Theorems are hard to parse. It would be better if the authors can simplify the statement using $o(\cdot), O(\cdot)$ notation. For example, in Theorem 1, many unnecessary constants are displayed, making it hard to see why it holds with a vanishing probability.

- The authors should used a different notation for $V$ in Theorem 2 as $V$ is already used for the variance term.

**Questions:**

- What happens in the experiment if we use other stochastic gradient descent algorithms (SGD, Adam, etc.) instead of the Langevin dynamics? Other algorithms are also capable of escaping local minima.

- What is the justification for using $\lambda = \frac{1}{n}$ in the experiment? What happens if we optimally tune the ridge parameter?

- As mentioned above, it is hard to judge how restrictive the assumptions are and how often the suggested method improves over ridge regression. Therefore, I would like to see some experiments with real-world data.

---

> ### Author Response · Authors · 2023-11-19
> **Reply to reviewer 8UMM (1)**
>
> Thank you very much for your insightful comments.
>
> __Q__. There are many additional assumptions stated inside the theorems: "for all  which satisfies ..."(Theorem 1), "there exists  such that .." (Theorem 2), "the rows of
> are independent" (Theorem 3), "the condition number of .." (Corollary 1). To me, the assumptions looked very artificial, and it was hard to see why it is reasonable to assume those. It would strengthen the paper if the authors could state all necessary assumptions separately before stating the theorems. Without this, it is hard to judge how non-trivial the results are.
> __A__. First, we would like to emphasize that the assumptions we put in this study are rather standard.
> Although they would look complicated,  each assumption can be well interpreted as follows:
>
>  - The phrase ``for all $W$ which satisfies ...'' in Theorem 1
>  indicates that we discuss $W$ with a small value of our criterion.
> This is quite natural because, when we discuss a model selection criterion, we should choose $W$ that minimizes the criterion $R(W)$ so that we obtain the minimum predictive error.
> This is exactly analogous to other model selection criteria such as AIC and BIC in which we choose a model that minimizes those model selection criteria.
> Therefore, we restricted the argument on the parameter range of $W$ where $W$ achieves almost minimum value of $R(W)$.
>
>  - The assumption ``there exists $k\leq n$ such that ...'' in Theorem 2 is a relaxation of Assumption 3 and implies that the decay of $\Sigma_{\beta}^{1/2}\Sigma_x\Sigma_{\beta}^{1/2}$'s eigenvalues is not too slow. This is met in the following two situations:
>     - (1) The first situation is that the decay of $\Sigma_{\beta}$'s eigenvalues is not too slow. This happens when $m$ signals are correlated as stated in the top of p.5 in this paper, which is a very natural assumption seen when there is correlation between each component in a regression problem of multi-dimensional vectors.
> The assumption about the decay speed of $\Sigma_{\beta}$'s eigenvalues is also widely used in the analysis of high-dimensional regression problems such as Xu and Hsu (2019).
> Otherwise, we don't gain any benefit from the neural network model because the signals are almost independent across the outputs.
>     - (2) The second situation is that the ``alignment matrix''  $\Sigma_{\beta}^{\frac{1}{2}}\Sigma_x\Sigma_{\beta}^{\frac{1}{2}}$ between $\Sigma_x$ and $\Sigma_{\beta}$ has fast decay of eigenvalues as referred in p.6 of this paper, which corresponds to a situation where the principle components of both $\Sigma_x$ and $\Sigma_{\beta}$ are well aligned.
> In this case, larger benefits of feature learning can be obtained by finding informative direction of input at the first layer. Please refer to the description in the paper for explanations of alignment and misalignment. This assumption about the alignment of $\Sigma_x$ and $\Sigma_{\beta}$ is often used in the theoretical analysis of feature learning, and is also used in Wu and Xu (2020), for example.
>
> - The assumption in Corollary 1 that ``the rows of $XW_BU_B^{\top}$ are independent'' is the one that follows the assumption used in Tsigler and Bartlett (2020) to derive the lower bound of the predictive error in high dimensional regression.
>
>
> As we have seen above, these assumptions are natural from a statistical theory perspective, and please note that another reviewer is actually convinced by this point.
>
> J. Xu and D. J. Hsu. On the number of variables to use in principal component regression. Advances in Neural Information Processing Systems, 32, 2019.
>
> D. Wu and J. Xu. On the optimal weighted l2 regularization in overparameterized linear regression. In H. Larochelle, M. Ranzato, R. Hadsell, M. Balcan, and H. Lin, editors, Advances in Neural Information Processing Systems, pages 10112–10123. Curran Associates, Inc.
>
> A. Tsigler and P. L. Bartlett. Benign overfitting in ridge regression. arXiv preprint arXiv:2009.14286, 2020.

---

> ### Author Response · Authors · 2023-11-19
> **Reply to reviewer 8UMM (2)**
>
> (This is a continuation of "Reply to Reviewer 8UMM (1)". We are sorry for the long reply.)
>
> __Q__. Although the authors have provided Example 1--4 to show the advantage of using  $R(W)$ over ridge regression, the settings seem very artificial and restrictive.
> __A__. Although you mention that our four examples look artificial, we believe that it is quite important to presents synthetic examples in which we can understand the general theory in an intuitive way.
> Our theorems on the property of the proposed criterion $R(W)$ are presented in a considerably general form, which prevents the readers to see its benefit immediately.
> Therefore, we presented concrete ``easy'' but essential examples to demonstrate how our theory can prove the benefit of feature learning.
> Naturally, an analogous consequence can be obtained in a slightly different setting.  Of course, it is also an important future work to find more general but interpretable situations where the ridge regression fails but feature learning works.
>
> __Q__.  I wouldn't be too surprised even if there exists a pathological setting where standard ridge regression does not generalize well and there is a quick fix for it.
> __A__. It is not trivial how to fix the failure of normal ridge regression in a general setting.
> If we know the true signal $\beta_*$, we may construct a quick fix of ridge regression in a case-by-case way. However, this does not work in practical situations and does not scale for variety of situations. On the other hand, our analysis shows that just minimizing the proposed criterion can resolve these problems in a unifying manner because it minimizes the predictive error.
> Moreover, the aim of this study is to purely investigate the benefits of feature learning, and thus a case-by-case improvement of ridge regression is not our research target.
>
> __Q__.  This happens again in the experiment; the authors choose a very specific parameters and argue that the proposed method improves over ridge regression. I would be interested to see further justification for considering such setting.
> __A__. As we have mentioned above, we consider that it is important to present a simple synthetic examples to understand our general results even in the numerical evaluations so that the readers have better understandings of our study.
>
> __Q__. What is the justification for using $\lambda=1/n$ in the experiment? What happens if we optimally tune the ridge parameter?
> __A__. We agree with you that the comparison was not fair unless we chose the optimal tuning parameters for both our method and the naive ridge regression.
> Hence, we carefully re-examined the experimental settings and conducted the experiments again with optimally tuned regularization parameters. The regularization parameter for ridge regression was selected from $1.0, 0.5, 0.1, 0.01, 0.001, 0.0001$ to minimize the predictive
> risk calculated using additional data. Even under these settings, we observed significant superiority of the proposed method in the situation where feature learning is effective.

---

> > ### Comment · Reviewer_8UMM · 2023-11-19
> >
> > Thanks for the response. I updated my score.

---

### Official Review · Reviewer_Uw19 · 2023-10-28

**Soundness:** 3 good
**Presentation:** 3 good
**Contribution:** 3 good
**Rating:** 8
**Confidence:** 4

**Summary:**

The authors consider a two-layer linear network for multivariate regression. They introduce an upper bound on the standard MMSE loss, which they treat as a surrogate minimization objective for minimizing the predictive loss. They provide a thorough analysis of this bound and show that if the true regression coefficients are misaligned with the data, then a two-layer network trained using the upper bound as a loss function, outperforms ridge regression and closely matches they Bayes optimal estimator. They also provide experiments corroborating their theoretical results.

**Strengths:**

- Introduction of an upper bound obtained by adding a regularization term (degrees of freedom), which has previously been used in the setting where the number of samples is much larger than the dimension.
- Theoretical analysis of the proposed upper bound and the neural network trained on it, which provides insight as to how/why it works
- Characterization of a regime in which the network outperforms ridge regression and closely matches Bayes-optimal performance
- Clear presentation of the results

**Weaknesses:**

- Since the bounds in the theorems only hold up to a constant, they only give quantitative results if the misalignment is strong enough for the corresponding bound to be of vanishing order

**Questions:**

- The theoretical results give sufficient conditions for the superior performance of the network over ridge regression. Do you believe these conditions are also necessary and did you try any numerical experiments in that direction?

---

> ### Author Response · Authors · 2023-11-19
> **Reply to Reviewer Uw19**
>
> Thank you very much for understanding our research correctly and appreciating the contributions.
>
> __Q__. Since the bounds in the theorems only hold up to a constant, they only give quantitative results if the misalignment is strong enough for the corresponding bound to be of vanishing order.
> __A__. Our main theorems (Theorems 1, 2, and 3) are proven by combining various concentration inequalities so that we obtain a uniform bound over the choice of $W$.
> Thus, it is not easy to derive the constants accurately as in several previous studies in high dimensional statistics.
> However, we also think that such a constant factor freedom could be absorbed into the choice of the regularization parameter in practice.
> As an important future work, we would like to clarify their constants for precise understanding of the generalization of neural network.
>
> __Q__. The theoretical results give sufficient conditions for the superior performance of the network over ridge regression. Do you believe these conditions are also necessary and did you try any numerical experiments in that direction?
> __A__. The four examples in this paper are obtained by considering situations where the upper bound of Proposition 1 does not generalize, and since this upper bound also achieves the lower bound, we believe that these examples essentially cover almost all situations where ridge regression cannot generalize and our proposed method can generalize.

---

### Official Review · Reviewer_hzqz · 2023-10-30

**Soundness:** 3 good
**Presentation:** 2 fair
**Contribution:** 3 good
**Rating:** 6
**Confidence:** 3

**Summary:**

In this paper, the primary objective of the authors is to understand how and why a linear two-layer neural network outperforms classical ridge regression in a multivariate output regression problem, thereby illustrating how, why and when proper feature feature learning can benefit the learning. To that end, the authors propose an optimal regularization for training a two-layer linear neural network. This loss incorporates a regularization term inspired by the Widely Applicable Information Criterion (WAIC). The authors show that this loss represents an upper bound for the predictive regression error. Moreover, the authors show that the upper bound can reach a lower bound using Bayes optimal risk. The authors compare ridge regression with the proposed optimal regularization, highlighting a set of cases and examples where optimal regularization yields better results, showing when good regularization leads to greater learning.

**Strengths:**

1- The purpose of the paper is quite important. Explaining theoretically why a neural network is able to outperform classical linear models through efficient feature learning is an interesting problem even in the simple framework of a linear two-layer network.

2- The proof that the proposed regularization allows us to obtain the lower bound given by the optimal Bayes is also interesting since it justifies the optimality of the approach showing that the approach allows us to obtain the best performance in this class of functions of linear two-layer networks. In addition, the comparison with ridge regression and the illustration of specific cases where feature learning can dominate a non feature learning approach is also interesting.

3- Even though few experiments have been carried out, they tend to suggest that regularization does in fact enable good performance to be achieved compared with conventional ridge regression.

**Weaknesses:**

1- The methodology and design of regularization in neural networks seem arbitrary. They suggest that the regularization used in previous cases is applied to neural networks, and by chance, it provides an upper bound of the true risk and constitutes a lower bound. The author postulates that this type of regularization appears implicitly in neural networks. The question is why this regularization is exploited by neural networks and not another that would also allow reaching the Bayes risk. For example, in recursive feature machines where authors apply an optimal feature learning step and a ridge regression step, one could expect results that reach the lower bound. One could say that it is possible to look for regularization that already exists in the literature or create new ones that will also show that it is an upper bound of the predictive loss and lead to a lower bound of the Bayes risk. But how can we ensure that the regularization proposed by the authors is unique and actually the one implicitly used by linear two-layer Neural Network? Otherwise, it is possible to write a lot of studies with different regularizations. The only thing we seem to be sure of is that ridge regression is not optimal.

2- Along the same lines, one could ask whether this regularization is not optimal in the restricted setting considered by the authors (hypothesis on the data, generative model considered for the data, zero mean data, etc.). In a neural network study, one would expect to study the learning dynamics that naturally bring out the learning model and the implicit regularization that could also depend on the type of data and not be generic. And probably not the reverse: propose an arbitrary regularization and showing by its properties (lower bound of Bayes risk, upper bound of predictive risk) that it is the actual regularization achieved by linear neural network.

3- Even though the theoretical contributions are important, it would have been interesting to illustrate the theoretical conclusions a bit more. In particular, it would have been interesting to compare the W learned by a linear NN network (without the regularization proposed (by a gradient descent method, for example)) and compare it to that obtained by regularization. This could represent an empirical argument that the actual learning process follows that of the proposed regularization. It would also be interesting to test the robustness of the conclusions on real data to see how dependent they are on the model considered and not obsolete for real data.

**Questions:**

1- The sentence: "This is because there is a difficulty in feature learning ..." in the introduction seems not to be understandable, at least to me.

2- I think the main question in the introduction is a bit confusing: "Can we design an optimal regularisation...". I think this question is interesting, but maybe it's not related to the learning process in real NN. At least, even if implicit regularisation doesn't explain all the phenomena, it is implicitly given by the learning process and the dynamics of learning. Also, this question seems a bit challenging and too broad as I would expect this regularisation term to depend on the data modelling under consideration.

3- I think some explanations would have been welcome for assumption 1. Why do you consider putting the sub-Gaussianity assumption on the rows of $X\Sigma_x^{-1/2}$ and not on the rows of $X$?

4- The explanation at the end of page 4 and the beginning of page 5 is confusing to me. A lot of information is given, sometimes without any link, and some terms are not explained. In particular, several topics are mentioned in the same paragraph which are not related to each other (decay of eigenvalues for benign overfitting, high dimensional regime considered for $d, n$, ...). Furthermore, the kernel regime is not defined.

5- In numerical experiments, it would have been fairer to tune the regularization parameter for ridge regression and take the best one. What if the poor performance of ridge is only due to a bad choice of the regularization parameter?  It would have been interesting to add the important baseline that would train a linear neural network as usual on the synthetic data and also some real data, at least to show the robustness of the approach with respect to the data distribution and the model considered.

---

> ### Author Response · Authors · 2023-11-19
> **Reply to reviewer hzqz (1)**
>
> Thank you very much for your insightful comments.
>
> __Q__. The methodology and design of regularization in neural networks seem arbitrary. They suggest that the regularization used in previous cases is applied to neural networks, and by chance, it provides an upper bound of the true risk and constitutes a lower bound.
> __A__. As mentioned in the paper, the criterion we propose is introduced as an exact estimator of the predictive risk of the two-layer linear neural network, and it is inevitable that regularization of this form achieves the lower bound as in Theorem 3 (please refer to the proof of Theorem 1 for more details). The correct understanding is that as a result of constructing an exact estimator of the predictive risk, the regularization term that should appear must essentially coincide with the degrees of freedom that appears in WAIC and Mallows' Cp.
>
> __Q__. The author postulates that this type of regularization appears implicitly in neural networks. The question is why this regularization is exploited by neural networks and not another that would also allow reaching the Bayes risk.
> __A__.  As emphasized in the introduction, the goal of this study is to reveal the maximum potential of feature learning for a theoretical understanding of the generalization of neural networks. Therefore, the regularization proposed in this study was not introduced to ``explain'' the usual regularization or implicit regularization that has been used conventionally. Regarding the uniqueness of this regularization, if $W$ is fixed and $d \leq n$, the unbiased estimator of the predictive risk (with an appropriate regularization parameter) takes the same form. Thus, it is unique.
>
> __Q__. For example, in recursive feature machines where authors apply an optimal feature learning step and a ridge regression step, one could expect results that reach the lower bound.
> __A__.  Of course, as you have illustrated, it is possible to add something that becomes zero when taking the expectation or something that is $o(1)$, but we see no point in unnecessarily increasing the variance by introducing such an additional term.
>
> __Q__. One could say that it is possible to look for regularization that already exists in the literature or create new ones that will also show that it is an upper bound of the predictive loss and lead to a lower bound of the Bayes risk. But how can we ensure that the regularization proposed by the authors is unique and actually the one implicitly used by linear two-layer Neural Network? Otherwise, it is possible to write a lot of studies with different regularizations. The only thing we seem to be sure of is that ridge regression is not optimal.
> __A__. Although we haven't been able to demonstrate uniqueness in a completely strict sense, we believe that bringing in another regularization in the above sense would essentially result in almost the same consequence. Furthermore, it is entirely nontrivial that the regularization term stemming from this unbiased estimator indeed achieves both the upper and lower bounds of the predictive risk up to a constant factor even when the input is high-dimensional and $W$ can move freely. This is a remarkable novelty of our research.

---

> ### Author Response · Authors · 2023-11-19
> **Reply to reviewer hzqz (2)**
>
> (This is a continuation of "Reply to Reviewer hzqz (1)". We are sorry for the long reply.)
>
> __Q__. Along the same lines, one could ask whether this regularization is not optimal in the restricted setting considered by the authors (hypothesis on the data, generative model considered for the data, zero mean data, etc.). In a neural network study, one would expect to study the learning dynamics that naturally bring out the learning model and the implicit regularization that could also depend on the type of data and not be generic. And probably not the reverse: propose an arbitrary regularization and showing by its properties (lower bound of Bayes risk, upper bound of predictive risk) that it is the actual regularization achieved by linear neural network.
> __A__. Reveling the role and mechanism of the implicit regularization and the training dynamics of the usual neural network and its affect on the generalization is actually one of the most important researche topics in the field of deep learning theory.
> However, we strongly disagree the opinion that one should not acknowledge theoretical studies which are not aimed to such a typical research direction.
> As we emphasized in the introduction, this research aims to elucidate the full potential of feature learning, which can be performed by neural networks.
> Although this is not directly intended to ``understand'' the real neural network behavior,
> it is important to reveal the potential benefit of neural network from the theoretical view point,
> which is also extremely meaningful in actual feature learning scenarios. If the optimization methods and neural network architectures used so far realize implicit regularization that reduces the criteria proposed in this research or something similar, such methods can be characterized as ones that maximize the potential of feature learning. Moreover, even if existing methods cannot realize implicit regularization that reduces the criteria of this research, studying optimization methods and architectures that realize it can lead to the development of higher-performing methods that are theoretically substantiated.
>
> __Q__. Even though the theoretical contributions are important, it would have been interesting to illustrate the theoretical conclusions a bit more. In particular, it would have been interesting to compare the W learned by a linear NN network (without the regularization proposed (by a gradient descent method, for example)) and compare it to that obtained by regularization. This could represent an empirical argument that the actual learning process follows that of the proposed regularization. It would also be interesting to test the robustness of the conclusions on real data to see how dependent they are on the model considered and not obsolete for real data.
> __A__. As you properly pointed out, investigating the relationship between the implicit regularization in practical deep learning and the method proposed in this study is indeed an important issue. We may closely compare the implicit regularization induced by the usual stochastic gradient descent and the optimal regularization proposed in this study. That would bring an interesting insight into the practical applications.
> As an important future work, we would like to explore a situation where the implicit regularization performed by the real neural networks yields the Bayes optimal risk.
>
> __Q__.  The sentence: "This is because there is a difficulty in feature learning ..." in the introduction seems not to be understandable, at least to me.
> __A__. The feature mapping requires additional parameters, and naively minimizing the empirical loss in high-dimensional settings, which are already prone to overfitting, could further promote overfitting if we increase the number of parameters. Hence, we advocate for the necessity of regularization, including implicit regularization.
>
> __Q__.  I think the main question in the introduction is a bit confusing: "Can we design an optimal regularisation...". I think this question is interesting, but maybe it's not related to the learning process in real NN. At least, even if implicit regularisation doesn't explain all the phenomena, it is implicitly given by the learning process and the dynamics of learning.
> __A__.  As we emphasized in the introduction, the ultimate goal of this paper is to understand the generalization phenomena of neural networks, not to provide theoretical guarantees for conventional methods. However, as mentioned above, this study is also highly meaningful for real neural networks in the sense that it leads to providing theoretical backing for feature learning of traditional methods, and the development of new methods using the results of this study.

---

> ### Author Response · Authors · 2023-11-19
> **Reply to reviewer hzqz (3)**
>
> (This is a continuation of "Reply to Reviewer hzqz (2)". We are sorry for the long reply.)
>
> __Q__. Also, this question seems a bit challenging and too broad as I would expect this regularisation term to depend on the data modelling under consideration.
> __A__. As for the broadness of this goal, in this study, assumptions such as multi-output are made when designing optimal regularization, but it is natural to impose appropriate assumptions and conditions when deriving theoretical results. Also, please note that the multi-output regression problem considered in this study is a natural problem setting that includes widely used problem settings such as multi-class classification problems.
>
> __Q__. I think some explanations would have been welcome for assumption 1. Why do you consider putting the sub-Gaussianity assumption on the rows of $X\Sigma_x$ and not on the rows of $X$?
> __A__. This is one of the standard assumptions required to utilize the concentration inequality such as Bernstein's inequality. It is a natural assumption in statistical learning theory, and has been used in several existing research such as Chattterji et al. (2022).
>
> N. S. Chatterji, P. M. Long, and P. L. Bartlett. The interplay between implicit bias and benign overfitting in two-layer linear networks. Journal of Machine Learning Research, 23(263):1–48, 2022.
>
> __Q__.  Furthermore, the kernel regime is not defined.
> __A__. The kernel regime is a framework for performing regression in a fixed reproducing kernel Hilbert space, a framework widely used in statistical learning theory.
>
> __Q__. A lot of information is given, sometimes without any link, and some terms are not explained. In particular, several topics are mentioned in the same paragraph which are not related to each other (decay of eigenvalues for benign overfitting, high dimensional regime considered for , ...). Furthermore, the kernel regime is not defined.
> __A__. This paragraph emphasizes the benefits of converting the model generalization framework from benign overfitting to a kernel method, with the optimal reproducing kernel Hilbert space in terms of predictive risk by feature learning. With the addition of feature learning, generalization ability of the models is enhanced, surpassing what can be achieved with benign overfitting.
>
> __Q__.   In numerical experiments, it would have been fairer to tune the regularization parameter for ridge regression and take the best one. What if the poor performance of ridge is only due to a bad choice of the regularization parameter? It would have been interesting to add the important baseline that would train a linear neural network as usual on the synthetic data and also some real data, at least to show the robustness of the approach with respect to the data distribution and the model considered.
> __A__. Regarding the numerical experiments, we agree with you that the comparison was not fair unless we chose the optimal tuning parameters for both our method and the naive ridge regression as you pointed.
> Hence, we carefully re-examined the experimental settings and conducted the experiments again with optimally tuned regularization parameters. The regularization parameter for ridge regression was selected from $1.0, 0.5, 0.1, 0.01, 0.001, 0.0001$ to minimize the predictive
> risk calculated using additional data. Even under these settings, we demonstrated the significant superiority of the proposed method in the situation where feature learning is effective.

---

> > ### Comment · Reviewer_hzqz · 2023-11-23
> > **Answer to rebuttal**
> >
> > I think the main point of contention is the assumption that WAIC regularization must be introduced to explicitly understand the principle of feature learning in a neural network. Although I understand that regularization has important properties, the fact that it explains the process of feature learning seems to me to be too strong a conclusion. There is no theoretical justification that this is actually what happens. In fact, the article would have proposed a new algorithm which, thanks to WAIC regularisation, achieves the performance of a neural network learnt with feature learning process, which would have been understandable. But to postulate that this would explain feature learning seems to me to be a very strong conclusion.
> >
> > I think that even if a hypothesis is classic in a field, it should not be used as an argument to explain why a particular hypothesis about the data is being considered. But this remark is very slight and does not influence my judgement of the article.

---

### Official Review · Reviewer_pQvZ · 2023-11-01

**Soundness:** 4 excellent
**Presentation:** 4 excellent
**Contribution:** 3 good
**Rating:** 6
**Confidence:** 4

**Summary:**

This paper studies two layer linear neural networks with multiple outputs. The authors propose a penalty term (regularization) and add it to their least squares optimization problem. This penalty term is effective because it enables "feature learning" They analyze the estimator in the high-dimensional regime and show that in some scenarios, it can outperform ridge regression.

**Strengths:**

- Feature learning is a very important problem and is attracting a lot of attention from the deep learning theory community. The problem is very well motivated.

- The penalty term introduced is related to the classic Millow's $C_p$ and WAIC. The properties of these criteria were already studied in the classic statistical setting where dimension is fixed and the number of samples is large. The authors analyze these in the high-dimensional regime. This is interesting on its own.

- The paper is solid, the proofs seem to be correct, and the paper is well written in general.

**Weaknesses:**

* The authors argue that the proposed method can beat ridge regression. However, how does it compare to ridge regression with optimally tuned regularization? For example, in figure 1, "Normal Ridge" corresponds to ridge regression with $\lambda = 1/n$ which is not the optimally tuned. More importantly, the authors consider a case where the $\beta$s are not isotropic. In that case, one might try to look at the optimization problem $\hat\beta_{\Omega} = \min_{\beta}||y - X \beta||_2^2 + ||\Omega \beta||_2^2$ where $\Omega$ is a $d\times d$ matrix. Then, try to tune $\Omega$ optimally (similar to Wu and Xu 2020). One can also use the min-norm interpolation version of this that was studied in (Sun et al., 2022), section 2.2.


Yue Sun, Adhyyan Narang, Halil Ibrahim Gulluk, Samet Oymak, Maryam Fazel. Towards Sample-efficient Overparameterized Meta-learning, NeurIPS 2022.

 For example, setting $\Omega \asymp \Sigma_\beta^{-1}$ can exactly give the Bayes optimal in Proposition 1. It is also doing the feature learning because it regularizes the dimensions with small signal power more that the directions with strong signal. What is the benefit of the method proposed by the authors to this? How do they compare? What are the benefits?


* Right before section 2, it is mentioned that (Ba et al., 2022) studies the problem in the setting where $n\geq d$. Is this true? They consider the regime where $d$ and $n$ are in the same order but $d$ can be large or smaller than $n$. How does the results in this paper compare to the results of (Ba et al., 2022) with $O(\sqrt{n})$ gradient step size?

* How does the proposed method compare to sketched linear regression? See e.g. (Chen et al., 2023).

Xin Chen, Yicheng Zeng, Siyue Yang, Qiang Sun. Sketched ridgeless linear regression: The role of downsampling, 2023.

**Questions:**

Please see above.

---

> ### Author Response · Authors · 2023-11-19
> **Reply to reviewer pQvZ (1)**
>
> Thank you very much for your insightful comments.
>
>  __Q__. The authors argue that the proposed method can beat ridge regression. However, how does it compare to ridge regression with optimally tuned regularization? For example, in figure 1, "Normal Ridge" corresponds to ridge regression with  which is not the optimally tuned.
> __A__. We would like to emphasize that Proposition 1 holds for any regularization parameter $\lambda$. Therefore, the statement applies to the optimally tuned $\lambda$ as a special case. Consequently, the comparison between the proposed method and ridge regression in our four examples is valid for any choice of $\lambda$. Thus, the analyses there state that, even if we choose the optimal $\lambda$ for the ridge regression as well as our proposed approach, our method can significantly outperform the ridge regression in those situations.  Regarding the numerical experiments, we agree that the comparison is not fair unless we choose the optimal tuning parameters for both methods.  Hence, following your suggestion, we carefully re-examined the experimental conditions and conducted the experiments again with optimally tuned regularization parameters. The regularization parameter for ridge regression was selected from $1.0, 0.5, 0.1, 0.01, 0.001, 0.0001$ to minimize the predictive risk calculated using additional data. Even under these settings, we demonstrated the significant superiority of the proposed method in the situation where feature learning is effective.
>
> __Q__.  More importantly, the authors consider a case where the s are not isotropic. In that case, one might try to look at the optimization problem $\hat{\beta}\_\Omega= \min\_{\beta}  \Vert y-X\beta \Vert^2+ \Vert \Omega \beta  \Vert\_2^2$ where $\Omega$ is a $d \times d$ matrix. Then, try to tune  optimally (similar to Wu and Xu 2020). One can also use the min-norm interpolation version of this that was studied in (Sun et al., 2022), section 2.2.
> __A__. As for the relationship between our research and generalized ridge regression, the first layer's coordinate transformation corresponds to selecting the metric for the norm employed in the $l_2$ regularization.
> In that sense, the two-layer linear neural network in our paper is equivalent to the generalized ridge regression you are referring to. In fact, for a fixed $W$, we can write $\min\_{\beta}\left \Vert y-XW^{\top}\beta \right \Vert^2 + \lambda \left\Vert\beta\right\Vert\_2^2 = \min\_{\beta'}\left \Vert y-X\beta' \right \Vert^2 + \lambda \left\Vert W^{-\top}\beta\right\Vert\_2^2$, and the same discussion can be applied as the chapter on Bayes risk optimality in this paper. Therefore, selecting the optimal first-layer parameters by our proposed method is equivalent to selecting the optimal metric for the regularization of the generalized ridge regression. Therefore, the feature learning based on the selection of $\Omega$ you mentioned is actually the same framework as this research. As you say, theoretically, it is optimal to adopt a metric in $l_2$ norm that is proportional to $\Sigma_{\beta}^{-1}$, but $\Sigma_{\beta}^{-1}$ is a matrix calculated from signal information, which is unknown and impossible to use in training. The contribution of this work lies in that our proposed method can extract the information of such $\Sigma_{\beta}^{-1}$ from the training data through feature learning and can actually obtain nearly optimal $l_2$ norm metrics even in the high dimensional setting. This is quite important to obtain better predictive accuracy as we have presented in the main text.
> In terms of comparing with Wu and Xu (2020), they theoretically analyzed the optimal $l_2$ regularization metric, that is, the first layer parameters in the context of a two-layer linear neural network. While we perform the analysis in general situations, they limit it to fairly restricted conditions where $\Sigma_x$ and $\Sigma_{\beta}$ can be simultaneously diagonalized, giving us an advantage in that respect. Please note that the results of both coincide in the case that $\Sigma_x$ and $\Sigma_{\beta}$ can be simultaneously diagonalized, as stated in the chapter on Bayes risk optimality in this paper. Additionally, regarding the choice of the optimal metric, while they optimize in a further limited parameter space, our method optimizes in a broader, more general parameter space, giving us an advantage.

---

> ### Author Response · Authors · 2023-11-19
> **Reply to reviewer pQvZ (2)**
>
> (This is a continuation of "Reply to Reviewer pQvZ (1)". We are sorry for the long reply.)
>
> __Q__. Right before section 2, it is mentioned that (Ba et al., 2022) studies the problem in the setting where $n\geq d$. Is this true? They consider the regime where $d$ and $n$ are in the same order but $d$ can be large or smaller than $n$. How does the results in this paper compare to the results of (Ba et al., 2022) with $O(\sqrt{n})$ gradient step size?
> __A__. Certainly, while the entire paper by Ba et al. (2022) deals with the situation where $d \geq n$ too, Theorem 11 in their paper, which demonstrates the superiority of one gradient step feature learning with a step size $\sqrt{n}$ over the conjugate kernel, requires $d < n$ for the model to generalize because the upper bound of the model after feature learning is given as $O(\frac{d}{n})$. The remarkable point in our study which is different from theirs is that we can show superiority over models without feature learning even in situations where $d\sim n$ or $d \gg n$.
>
> __Q__.  How does the proposed method compare to sketched linear regression? See e.g. (Chen et al., 2023).
> __A__.  Since the sketched linear regression performs random down sampling, it achieves better computational complexity over our proposed method (or more generally, neural network approaches).
> In terms of generalization, since sketching matrices are random projection matrices, they are expected to reduce variance without greatly increasing bias by appropriately reducing the model dimension.
> However, the construction of sketching matrices does not utilize any information of the input-output relation because it is a completely random procedure, which does not yield any improvement on the bias.
> This is especially problematic when the bias is large, for example, when $\Sigma_x$ and $\Sigma_{\beta}$ are misaligned. In that setting, the performance improvement by sketching is limited (i.e., it can reduce only the variance) while our proposed approach achieves significant performance improvement.
> From the viewpoint of whether or not the bias can be efficiently reduced (in other words, whether they utilize the input-output relation or not), the proposed method and the sketched linear regression are entirely different.

---

> > ### Comment · Reviewer_pQvZ · 2023-11-20
> >
> > I thank the authors for the detailed response.
> >
> > 1. I thank the authors for providing simulation for ridge regularization with varying $\lambda$. This resolves my concern.
> >
> > 2. How is it possible to extract information from $\Sigma_\beta$ if we only have one instance of $\beta$? Doesn't it all boil down to the fact that some alignment is assumed between $\Sigma_\beta$ and $\Sigma_X$?
> >
> > 3. I agree with the authors on the assessment of sketched least squares, but still, my question is when (if) there is no common structure shared between $\Sigma_\beta$ and $\Sigma_X$, how can your approach achieve what you describe in your answer for my third question?

---

> ### Author Response · Authors · 2023-11-20
> **Reply to Reviewer pQvZ (3)**
>
> Thank you for your meaningful discussion.
>
> __Q__. How is it possible to extract information from $\Sigma_{\beta}$ if we only have one instance of $\beta$? Doesn't it all boil down to the fact that some alignment is assumed between $\Sigma_x$ and $\Sigma_{\beta}$?
> __A__. Theorems 1 and 2 hold for any $m$, so even if there is only one signal, it holds for $\Sigma_{\beta}=\beta\beta^{\top}$ and it is possible to extract the signal information. However, in the case of single-output, improvements cannot be expected as much as in the case of multiple-output in principle. At least, in Theorem 3 we assume that there are enough signals and that $\Sigma_{\beta}$ has an inverse matrix, so it cannot satisfy Bayes risk optimality. Although the proposed method can be applied to single-output cases as shown in Theorems 1 and 2, it should be noted that the main focus of this paper is on multi-output problem settings.
>
> __Q__. I agree with the authors on the assessment of sketched least squares, but still, my question is when (if) there is no common structure shared between $\Sigma_\beta$ and $\Sigma_X$, how can your approach achieve what you describe in your answer for my third question?
> __A__. As mentioned in this paper, our proposed criterion is constructed as an exact estimator of the predictive risk of a two-layer linear neural network (please refer to the proof of Theorem 1 for more details). Since it uses information about the input-output relation and predictive risk, it can extract information on the principle directions of $\Sigma_\beta$ from the data by making full use of the multi-output situation. This enables our proposed method to realize feature learning that is essentially different from methods based on the principal components of $\Sigma_x$ such as sketched linear regression. Of course, if $\Sigma_x$ and $\Sigma_{\beta}$ are aligned, the model is easier to generalize, but as the second example of comparison with ridge regression shows, the model can still generalize even when $\Sigma_x$ and $\Sigma_{\beta}$ are misaligned to some extent. This is because our proposed method realizes feature learning that extracts signal information even in such misaligned situations.

---

> > ### Comment · Reviewer_pQvZ · 2023-11-21
> >
> > I thank the authors for the response. The authors do indeed resolve some of my concerns regarding the paper. I am willing to raise my score to 6 if the following concern is resolved:
> >
> > In this work, the main benefits come from the fact that there is some alignment between $\Sigma_x$ and $\Sigma_\beta$. Even in the misaligned example, these matrices are still aligned to some extent. I'm still not sure how this method compares to other methods that are solely based on $\Sigma_x$. It might as well be the case that there is a benefit; but I cannot see that from the paper.
> >
> > Also, the authors are gaining information about $\Sigma_\beta$ through their multi-output model. In these scenarios, normal ridge seems like an easy opponent to beat.

---

> > > ### Author Response · Authors · 2023-11-21
> > > **Reply to Reviewer pQvZ (4)**
> > >
> > > Thank you very much for your careful reading and suggestive discussion.
> > >
> > > __Q__. In this work, the main benefits come from the fact that there is some alignment between $\Sigma_x$ and $\Sigma_{\beta}$. Even in the misaligned example, these matrices are still aligned to some extent. I'm still not sure how this method compares to other methods that are solely based on $\Sigma_x$. It might as well be the case that there is a benefit; but I cannot see that from the paper.
> > > __A__. As you pointed out, the current examples do not properly demonstrate the effectiveness of the proposed method when $\Sigma_x$ and $\Sigma_{\beta}$ are misaligned. To demonstrate the effectiveness of the proposed method under such situations, we have created a new example where the proposed method can outperform ridge regression even when $\Sigma_x$ and $\Sigma_{\beta}$ are completely misaligned. In this example, since $\Sigma_x$ and $\Sigma_{\beta}$ are completely misaligned, no method that only uses the information of $\Sigma_x$ can be expected to generalize.
> > >
> > > __Q__. Also, the authors are gaining information about $\Sigma_{\beta}$ through their multi-output model. In these scenarios, normal ridge seems like an easy opponent to beat.
> > > __A__. Our purpose of the comparison is purely to clarify the impact of feature learning, so we compared the feature learning method with the standard ridge regression for any $\lambda$. Also, note that we can construct examples where even in the case of a single output, feature learning with our proposed method outperforms ridge regression, and a method that only uses the information of $\Sigma_x$ other than ridge regression does not necessarily improve performance, especially in multi-output problem settings.

---

> > > > ### Comment · Reviewer_pQvZ · 2023-11-21
> > > >
> > > > 1-  Where is this example? I cannot find it. It would be great if the authors can use a different color for the revisions in the manuscript.
> > > >
> > > > 2- I agree and I am satisfied with the answers that the  authors provide; but I think the authors should be more transparent about this in the paper.

---

> > > > > ### Author Response · Authors · 2023-11-21
> > > > > **Reply to Reviewer pQvZ (5)**
> > > > >
> > > > > I apologize for the unclear change. The Example 2 in the latest version of the paper now submitted is replaced with the case where $\Sigma_x$ and $\Sigma_{\beta}$ are completely misaligned.

---

> > > > > > ### Comment · Reviewer_pQvZ · 2023-11-22
> > > > > >
> > > > > > I thank the author for the revisions. I raise my score to 6.

---

### Meta-Review · Area_Chair_ujin · 2023-12-19

**Metareview:**

The paper aims to theoretically analyze the statistical properties of the benefits of feature learning in a two-layer linear neural network with multiple outputs (in a high-dimensional limit). The authors propose a new criterion that allows feature learning of a two-layer linear neural network, and accordingly, they show that models with smaller values of the criterion generalize even in situations where normal ridge regression fails to generalize.

Most of the reviewers (including myself) are in favor of accepting the paper. I thank the authors for their thorough responses -- there was a good amount of discussion between the authors and the reviewers. I encourage the authors to incorporate all the great comments from the reviewers (who are very knowledgeable) in the revised version (e.g. comments on the methodology, comparison with related work, assumptions, adding examples, etc).

**Justification For Why Not Higher Score:**

--

**Justification For Why Not Lower Score:**

Based on my own reading, and communication with the authors, I think accepting this paper would be a reasonable decision.

---

### Decision · Program_Chairs · 2024-01-16

Accept (poster)